



# Acidification of the Nordic Seas

**Filippa Fransner[1], Friederike Fröb[1,2], Jerry Tjiputra[3], Nadine Goris[3], Siv K. Lauvset[3], Ingunn Skjelvan[3], Emil Jeansson[3], Abdirahman Omar[3], Melissa Chierici[4], Elizabeth Jones[4], Agneta Fransson[5], Sólveig R. Ólafsdóttir[6], Truls Johannessen[1], and Are Olsen[1]**

[1]Geophysical Institute, University of Bergen, and Bjerknes Centre for Climate Research, Bergen, Norway
[2]Max Planck Institute for Meteorology, Hamburg, Germany
[3]NORCE Norwegian Research Centre, Bjerknes Centre for Climate Research, Bergen, Norway
[4]Institute of Marine Research, Fram Centre, Tromsø, Norway
[5]Norwegian Polar Institute, Tromsø, Norway
[6]Marine and Freshwater Research Institute, Reykjavík, Iceland

**Correspondence:** Filippa Fransner (filippa.fransner@uib.no)

**Abstract.** CE1 Due to low calcium carbonate saturation states, and winter mixing that brings anthropogenic carbon to the deep ocean, the Nordic Seas and their cold-water corals are vulnerable to ocean acidification. Here, we present a detailed investigation of the changes in pH and aragonite saturation in the Nordic Seas from preindustrial times to 2100, by using in situ observations, gridded climatological data, and projections for three different future scenarios with the Norwegian Earth System Model (NorESM1-ME).

During the period of regular ocean biogeochemistry observations from 1981–2019, the pH decreased with rates of $2$–$3 \times 10^{-3}$ yr$^{-1}$ in the upper 200 m of the Nordic Seas. In some regions, the pH decrease can be detected down to 2000 m depth. This resulted in a decrease in the aragonite saturation state, which is now close to undersaturation in the depth layer of 1000–2000 m. The model simulations suggest that the pH of the Nordic Seas will decrease at an overall faster rate than the global ocean from the preindustrial era to 2100, bringing the Nordic Seas' pH closer to the global average. In the esmRCP8.5 scenario, the whole water column is projected to be undersaturated with respect to aragonite at the end of the 21st century, thereby endangering all cold-water corals of the Nordic Seas. In the esmRCP4.5 scenario, the deepest cold-water coral reefs are projected to be exposed to undersaturation. Exposure of all cold-water corals to corrosive waters can only be avoided with marginal under the esmRCP2.6 scenario.

Over all timescales, the main driver of the pH drop is the increase in dissolved inorganic carbon ($C_T$) caused by the oceanic uptake of anthropogenic $CO_2$, followed by the temperature increase. Thermodynamic salinity effects are of secondary importance. We find substantial changes in total alkalinity ($A_T$) and $C_T$ as a result of the salinification, or decreased freshwater content, of the Atlantic water during all time periods, and as a result of an increased freshwater export in polar waters in past and future scenarios. However, the net impact of this decrease (increase) in freshwater content on pH is negligible, as the effects of a concentration (dilution) of $C_T$ and $A_T$ are canceling.

## 1 Introduction

Since 1850, human activities have released $650 \pm 65$ Gt of carbon to the atmosphere, of which about 25 % have been taken up by the oceans (Friedlingstein et al., 2020), where it has been added to the $C_T$ pool. The increasing $C_T$ has resulted in a surface seawater pH decline of approximately 0.1 in the global ocean from the preindustrial era to the present day, which corresponds to an approximately 30 % increase in hydrogen ion ($H^+$) concentration (e.g., Doney et al., 2009; Gattuso and Hansson, 2011; Jiang et al., 2019). Furthermore, the decreasing pH also causes a reduction in the calcium carbonate ($CaCO_3$) saturation state ($\Omega$). It, hence, poses a serious threat to marine organisms that have shells or structures

consisting of $CaCO_3$, such as pteropods and corals (Guinotte et al., 2006; Turley et al., 2007; Manno et al., 2017; Doney et al., 2020; Doo et al., 2020). Depending on the $CO_2$ concentration pathway, future projections suggest further reductions in the surface ocean pH of 0.1–0.3 from the 1990s until the end of the 21st century (Bopp et al., 2013). While global average acidification rates for surface waters, both from preindustrial times to the present day and as projected for the future, are investigated in several studies (e.g., Caldeira and Wickett, 2003; Raven et al., 2005; Kwiatkowski et al., 2020), less is known about acidification rates on regional scales, especially below the surface.

The Nordic Seas, comprised of the Greenland, Iceland, and Norwegian seas (Fig. 1) and bounded by the Fram Strait in the north, the Barents Sea Opening to the northeast, and the Greenland–Scotland ridge in the south, are of particular interest when it comes to ocean acidification due to their specific physical, biogeochemical, and ecosystem characteristics. The surface circulation pattern of the Nordic Seas (e.g., Blindheim and Østerhus, 2013; Våge et al., 2013) is dominated by the relatively warm, saline Atlantic waters that flow northward as the Norwegian Atlantic Current in the east, mainly constrained to the Norwegian Sea, and relatively cold and fresh waters of Arctic origin flowing southward as the East Greenland Current in the west. In the Greenland and Iceland seas, deep and intermediate water masses are formed through open-ocean convection (Våge et al., 2015; Brakstad et al., 2019). Some of these water masses ultimately overflow the Greenland–Scotland ridge and feed into the North Atlantic Deep Water, helping to sustain the lower limb of the Atlantic Meridional Overturning Circulation (AMOC; Dickson and Brown, 1994; Våge et al., 2015; Chafik and Rossby, 2019). The surface water $pCO_2$ is generally lower than that of the atmosphere, making the Nordic Seas important sinks for atmospheric $CO_2$. This undersaturation results from several processes, including primary production, cooling of northward flowing Atlantic waters, and the inflow of $pCO_2$ undersaturated waters from the Arctic Ocean (Anderson and Olsen, 2002; Takahashi et al., 2002; Ólafsson et al., 2020b). Although the Nordic Seas are an overall sink for atmospheric $CO_2$, the direct uptake of anthropogenic $CO_2$ through air–sea $CO_2$ exchange is limited. Instead, there is a large advective supply of excess anthropogenic $CO_2$ from the south (Anderson and Olsen, 2002; Olsen et al., 2006; Jeansson et al., 2011) that contributes to the acidification. Part of the anthropogenic $CO_2$ that enters the Nordic Seas' surface waters is brought to deep waters through the deep water formation, from where it is slowly advected to the North Atlantic Ocean (Tjiputra et al., 2010; Perez et al., 2018). The deep-reaching anthropogenic $CO_2$, in combination with the prevailing low temperatures that give low saturation states of $CaCO_3$ (Ólafsson et al., 2009; Skjelvan et al., 2014), make the cold-water coral reefs of the Nordic Seas particularly exposed to ocean acidification (Kutti et al., 2014).

There has been extensive research on changes in the carbonate system and pH in the Nordic Seas facilitated by the many research and monitoring cruises in the area (e.g., Olsen et al., 2006; Ólafsson et al., 2009; Skjelvan et al., 2008; Chierici et al., 2012; Skjelvan et al., 2014; Jones et al., 2020; Skjelvan et al., 2021; Pérez et al., 2021). Between the 1980s and 2010s, the pH has been shown to decrease with rates of $-0.0023$ to $-0.0041\,\text{yr}^{-1}$ in surface waters, which is greater than expected from the increase in atmospheric $CO_2$ alone (Ólafsson et al., 2009; Skjelvan et al., 2014). This is consistent with the observations that have indicated a weakening of the $pCO_2$ undersaturation of the Nordic Seas surface waters, i.e., that surface ocean $pCO_2$ has risen faster than the atmospheric $pCO_2$ (Olsen et al., 2006; Skjelvan et al., 2008; Ólafsson et al., 2009), over the past few decades. The future pH of the Nordic Seas has been assessed with different modeling approaches (Bellerby et al., 2005; Skogen et al., 2014, 2018). Bellerby et al. (2005) investigated the impact of climate change on the Nordic Seas $CO_2$ system under a doubling of the atmospheric $CO_2$ to a value of 735 ppm (parts per million). It was done by combining the observed relationships between the inorganic $CO_2$ system and temperature and salinity with the output of ocean physics from the Bergen Climate Model. They found the pH to decrease by about 0.3, with the largest decrease taking place in the polar waters of the western Nordic Seas. For the future scenario A1B (see Meehl et al., 2007), which assumes approximately 700 ppm atmospheric $CO_2$ by the year 2100, Skogen et al. (2014) found that the pH of the Nordic Seas' surface waters decreases by 0.19 between 2000 and 2065 and that the aragonite saturation horizon shoals by 1200 m. They estimated $C_T$ to be the overall driver of this acidification. Skogen et al. (2018) looked into future changes in the Nordic Seas biogeochemistry under the Representative Concentration Pathway 4.5 (RCP4.5) scenario, a stabilization future scenario used within Climate Model Intercomparison Project Phase 5 (CMIP5; Taylor et al., 2012), and found the surface pH to drop by 0.18 between 1995 and 2070.

All the studies mentioned above have been focusing on selected periods of time and scenarios, using specific datasets. There is, to our knowledge, no work assessing pH changes and their drivers from the preindustrial era until the end of the 21st century, under different scenarios, using both observational and modeling data, and that provides a detailed regional perspective on the various drivers. In this study, we fill this gap by examining past, present-day, and projected future changes in pH and aragonite saturation in the Nordic Seas, over the full water column and in different regions, by using the best available information for the various time periods. This includes a combination of in situ observations, gridded climatological data, and Earth System Model (ESM) projections for different future scenarios.

**Table 1.** Direction of direct effects of an increase in temperature, salinity, $C_T$, and $A_T$ on pH and $\Omega$.

| Driver | pH | $\Omega$ |
|---|---|---|
| Temperature | − | + |
| Salinity | − | − |
| $C_T$ | − | − |
| $A_T$ | + | + |

## 2   Drivers of pH and saturation states – theoretical background

The rising atmospheric $CO_2$ concentration results in a flux of $CO_2$ from the atmosphere into the ocean. In the ocean $CO_2$ reacts with water to form carbonic acid ($H_2CO_3$), which then dissociates into bicarbonate ($HCO_3^-$) and hydrogen ions ($H^+$). A large part of the resulting $H^+$ is neutralized by carbonate ions ($CO_3^{2-}$) that have been supplied to the ocean by the weathering of carbonate and silicious minerals. Together, this forms the following equilibria TS1:

$$CO_2 + H_2O \rightleftharpoons H_2CO_3 \tag{R1}$$
$$H_2CO_3 \rightleftharpoons HCO_3^- + H^+ \tag{R2}$$
$$CO_3^{-2} + H^+ \rightleftharpoons HCO_3^-. \tag{R3}$$

Combined, the concentrations of $CO_2$, $H_2CO_3$, $HCO_3^-$, and $CO_3^{2-}$ constitute the concentration of dissolved inorganic carbon ($C_T$). In seawater, approximately 90 % of $C_T$ exists in the form of $HCO_3^-$, 9 % as $CO_3^{2-}$, and 1 % as $CO_2$.

As seen from Reactions (R1)–(R3), the dissolution of $CO_2$ in seawater results in an increase in $H^+$ concentration, which leads to a decrease in pH. On total scale, pH is defined as follows:

$$pH = -\log_{10}([H^+] + [HSO_4^-]), \tag{1}$$

where $HSO_4^-$ is sulfate. Apart from $C_T$, pH is influenced by temperature, salinity, and $A_T$. $A_T$ is mostly determined by the concentration of $HCO_3^-$ and $CO_3^{2-}$ (carbonate alkalinity). Temperature and salinity affect pH by altering the dissociation constants and, thus, the partitioning of $C_T$ between its different constituents. The relation between $C_T$ and $A_T$ influences pH by affecting the buffer capacity of seawater. The qualitative, direct effects of an increase in each property are shown in Table 1. Note that this table does not consider the indirect effects on pH, for example, from the change in air–sea fluxes that will follow from, e.g., a temperature-driven $pCO_2$ change (e.g., Jiang et al., 2019; Wu et al., 2019).

Reactions (R1)–(R3) show that an increase in anthropogenic $CO_2$ and $C_T$ results in a reduction in $CO_3^{2-}$. This affects the saturation state of $CaCO_3$ ($\Omega$), which is defined as follows:

$$\Omega = \frac{[Ca^{2+}][CO_3^{2-}]}{K_{sp}}, \tag{2}$$

where $K_{sp}$ is the solubility product. When $\Omega$ is less than one, the water becomes corrosive, and $CaCO_3$ starts to dissolve. In seawater, the two most abundant forms of $CaCO_3$ are calcite and aragonite. The saturation state of aragonite ($\Omega_{Ar}$) is lower than that of calcite ($\Omega_{Ca}$), as aragonite is more soluble than calcite, equating to a higher $K_{sp}$.

The impact of $C_T$ on the saturation state is seen in the spatial distribution of $\Omega$ in the surface ocean, which broadly follows temperature gradients (e.g., Orr, 2011; Jiang et al., 2019). The reason behind this temperature dependency is the higher $CO_2$ solubility of colder waters that results in higher $C_T$ concentrations. Consequently, cold waters have a relatively low $\Omega_{Ar}$ and $\Omega_{Ca}$ and are more vulnerable to acidification. $\Omega$ is also influenced by $A_T$, temperature, and salinity, as shown in Table 1.

The sensitivity of pH and $\Omega$ to an anthropogenic $CO_2$ increase is dependent on the buffer capacity of the seawater (e.g., Sarmiento and Gruber, 2006; Orr, 2011). Waters with a higher buffer capacity, i.e., higher concentrations of $CO_3^{2-}$, have the capability of converting a larger fraction of the absorbed $CO_2$ into bicarbonate. A smaller fraction remains as dissolved $CO_2$, implying a smaller increase in the seawater $pCO_2$. These waters, therefore, have the capability of absorbing more $CO_2$ for any given increase in atmospheric $pCO_2$ (assuming a uniform increase in $pCO_2$ between water masses), which also implies a larger decline in $CaCO_3$ saturation state. pH is, on the contrary, decreasing more in waters with a lower buffer capacity as they are less effective at neutralizing carbonic acid.

## 3   Data

### 3.1   Observational data

As observational data, we used $C_T$, $A_T$, temperature, salinity, phosphate, and silicate data collected between 1981 and 2019 during dedicated research cruises, at two time series stations, and in the framework of the Norwegian program "Monitoring ocean acidification in Norwegian waters". Sampling locations are shown in Fig. 1.

Data from 28 research cruises (Brewer et al., 2010; Anderson et al., 2013a, b; Anderson, 2013a, b; Bellerby and Smethie, 2013; Johannessen and Golmen, 2013; Johannessen, 2013a, b; Johannessen and Simonsen, 2013; Johannessen and Olsen, 2013; Johannessen et al., 2013a, b, c; Jones et al., 2013; Olsen et al., 2013; Olsen and Omar, 2013; Omar and Olsen, 2013; Omar and Skogseth, 2013; Omar, 2013; Pegler et al., 2013; Skjelvan et al., 2013; Wallace and Deming, 2014; Lauvset et al., 2016; Tanhua, 2017; Jeansson et al., 2018; Marcussen, 2018; Schauer et al., 2018) in the Nordic Seas were extracted from the GLODAPv2.2019 data product, which provides bias-corrected, cruise-based, interior ocean data (Olsen et al., 2019). The GLODAPv2 data product is considered to be consistent among cruises within 0.005 for

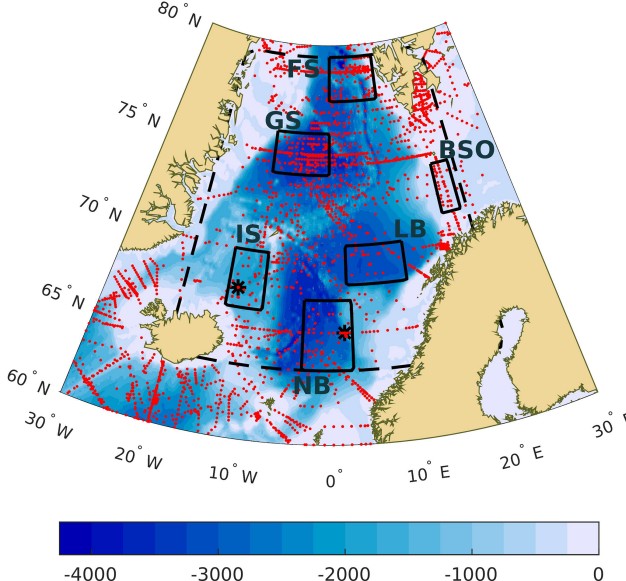

**Figure 1.** Map of the Nordic Seas with sampling locations (red). Also shown are the locations of the six regions where trends have been analyzed separately (rectangles), that is BSO (Barents Sea Opening), FS (eastern Fram Strait), GS (Greenland Sea), IS (Iceland Sea), LB (Lofoten Basin), and NB (Norwegian Basin). The dashed line marks the borders of the area that we define as the Nordic Seas. The asterisk markers in the Norwegian Basin and the Iceland Sea show the positions of Ocean Weather Station M and the Iceland Sea time series station, respectively. The filled contours illustrate the bathymetry at 250 m intervals. TS2

salinity, 2 % for silicate , 2 % for phosphate, and 4 µmol kg⁻¹ for both $C_T$ and $A_T$ (Olsen et al., 2019).

Time series data are from Ocean Weather Station M in the Norwegian Sea and from the Iceland Sea. The data from the Ocean Weather Station M, located at 66° N and 2° E, have been described in Skjelvan et al. (2008). At this station, sampling at 12 depth levels between the surface and seabed (2100 m) was carried out each month between 2002 and 2009 and 4–6 times each year between 2010 and 2019. For these data, the uncertainty related to the measurements is 0.001 for salinity, 0.7 µmol kg⁻¹ for silicate, 0.06 µmol kg⁻¹ for phosphate, and 2 µmol kg⁻¹ for $C_T$ and $A_T$. The time series station in the Iceland Sea, covering the period of 1985–2019, is situated at 68° N and 12.67° W. It is visited approximately 4 times a year, and samples are taken at 10–20 depth levels between surface and seabed (1900 m). The uncertainty related to the measurements at this station is 0.005 for salinity, 2 % for silicate, 2 % for phosphate, and 4 µmol kg⁻¹ for both $C_T$ and $A_T$ TS3. These data have been described in Ólafsson et al. (2009).

The data from the program "Monitoring ocean acidification in Norwegian waters" were sampled in the full water column along repeated sections in the Nordic Seas in the period 2011–2019 (Chierici et al.,

2012, 2013, 2014, 2015, 2016, 2017; Jones et al., 2018, 2019, 2020). The uncertainties related to the sampled data are 0.005 for salinity, 0.1 µmol kg⁻¹ for silicate, 0.06 µmol kg⁻¹ for phosphate, and 2 µmol kg⁻¹ for both $C_T$ and $A_T$.

Data from the eastern Fram Strait were collected on cruises with research vessel (R/V) *Helmer Hanssen*, within the CarbonBridge project, and on cruises with R/V *Lance* (Chierici et al., 2019c), as organized by the Norwegian Polar Institute.

Analytical methods for $C_T$ and $A_T$ in all datasets described above (for the Global Data Analysis Project, GLODAP, after the mid-1990s) follow Dickson et al. (2007), and the accuracy and precision is controlled by certified reference materials (CRMs) and by participation in international intercomparison studies (e.g., Bockmon and Dickson, 2015).

For estimates of atmospheric $CO_2$ change, we used the annual mean atmospheric $CO_2$ mole fraction ($x$CO₂) from the Mauna Loa updated records downloaded from http://www.esrl.noaa.gov/gmd/ccgg/trends/ (last access: 24 August 2020). Although the absolute value of atmospheric $x$CO₂ varies with latitude, the growth rates are the same across the globe.

### 3.2 Gridded climatological data

Climatological distributions of pH and $\Omega_{Ar}$ were calculated from $C_T$, $A_T$, temperature, salinity, phosphate, and silicate in the mapped GLODAPv2 data product (Lauvset et al., 2016). Preindustrial pH was determined by subtracting the GLODAPv2 estimate of anthropogenic carbon from the mapped climatology of the present (i.e., 2002) $C_T$ (Lauvset et al., 2016). We assumed that the changes in the temperature, salinity, and $A_T$ of the Nordic Seas are of minor importance for the changes in pH between preindustrial times and the present day. The GLODAPv2 estimate of anthropogenic carbon has been calculated using the transit time distribution (TTD) approach. We use the GLODAPv2 estimate of preindustrial pH only for comparison with the ESM data, specifically in Fig. 4 (Sect. 5.2).

### 3.3 Earth System Model data

For the estimates of past and future ocean acidification and saturation states under various climate scenarios, we primarily used the output from the fully coupled Norwegian Earth System Model with interactive atmospheric $CO_2$ (NorESM1-ME; Bentsen et al., 2013; Tjiputra et al., 2013, 2016). NorESM1-ME includes a dynamical isopycnic vertical coordinate ocean model, which originates from the Miami Isopycnic Coordinate Ocean Model (MICOM; Bleck and Smith, 1990), and the Hamburg Oceanic Carbon Cycle model (HAMOCC5; Maier-Reimer et al., 2005), adapted to the isopycnic ocean model framework. HAMOCC5 simulates lower trophic ecosystem processes up to the zooplank-

ton level, including primary production, remineralization, and predation, and full water column inorganic carbon chemistry. For our assessment, we utilized emission-driven historical simulations for the period from 1850 to 2005 and future scenario simulations for the period from 2006 to 2100, with a focus on RCPs 2.6, 4.5, and 8.5 (RCP2.6, RCP4.5, and RCP8.5; Meinshausen et al., 2011; van Vuuren et al., 2011a). RCP2.6 represents a mitigation scenario, RCP4.5 a stabilization scenario, and RCP8.5 a high-emission scenario. For the emission-driven runs used here, the corresponding scenarios are named esmRCP2.6, esmRCP4.5, and esmRCP8.5. Because the emission-driven scenarios prognostically simulate the atmospheric $CO_2$ concentration, it normally deviates from the prescribed concentrations in the concentration-driven scenarios (e.g., Friedlingstein et al., 2014). This is most critical for the historical scenario, where the prescribed atmospheric $CO_2$ follows the observed evolution. Here, deviations in the simulated atmospheric $CO_2$ might result in pH changes that differ from the actual pH change. The deviation in the simulated atmospheric $CO_2$ concentration in the emission-driven NorESM1-ME scenarios, from the prescribed one in the concentration-driven scenarios, and its effect on pH, is shown in Table S1 in the Supplement. Between 1850 and 2005, the model simulates an increase in the atmospheric $CO_2$ that is 14 ppm too strong, which results in a pH drop that exceeds the expected one by 0.01. This deviation is, however, 1 order of magnitude smaller than the actual pH change between 1850 and 2005 and has the same order of magnitude as the estimated uncertainty in both the observational data (Table 2) and the GLODAPv2 preindustrial pH estimate in the Nordic Seas (Sect. 4.4). The impact of the historical atmospheric $CO_2$ deviations between emission-driven and concentration-driven scenarios on pH change in our results is, therefore, negligible. Prior to the experiments, NorESM1-ME has undergone an extended spin-up procedure (> 1000 years). The changes in pH, in all considered depth layers, are minor (more than 1 order of magnitude less) in the preindustrial control simulation compared to the historical run and the future scenarios, indicating that the impact of model drift on our results is insignificant.

As a means of uncertainty assessment, we use the outputs from an ensemble of emission-driven ESMs that participated in CMIP5 (Taylor et al., 2012). We chose emission-driven rather than concentration-driven scenarios, as they include the feedback between the carbon cycle and the physical climate (Booth et al., 2013) and, thus, give a more comprehensive estimate of the effect of model-related uncertainties on climate projections and, in particular, on atmospheric $CO_2$, ocean carbon uptake, and ocean acidification. It is well known that the intermodel spread is larger in emission-driven simulations than in concentration-driven ones (Booth et al., 2013; Friedlingstein et al., 2014). While NorESM1-ME outputs are available for low to high future emission scenarios, the CMIP5 data portals only contains emission-driven ESM outputs for the high future emission scenario (esmRCP8.5).

Our ESM ensemble consists of all ESMs that participated in the experiment esmRCP8.5 and RCP8.5 and whose output is publicly available in one of the CMIP5 data portals and contains all variables needed for our analysis. This results in an ensemble of seven ESMs, namely (1) CESM1(BGC) (The Community Earth System Model, version 1 – Biogeochemistry; Long et al., 2013), (2) CanESM2 (second-generation Canadian Earth System Model; Arora et al., 2011), (3) GFDL-ESM2G (Geophysical Fluid Dynamics Laboratory Earth System Model with Modular Ocean Model, version 4 component; Dunne et al., 2013a, b), (4) GFDL-ESM2M (Geophysical Fluid Dynamics Laboratory Earth System Model with Generalized Ocean Layer Dynamics (GOLD) component; Dunne et al., 2013a, b), (5) IPSL-CM5A-LR (L'Institut Pierre-Simon Laplace Coupled Model, version 5A, low resolution; Dufresne et al., 2013), (6) MPI-ESM-LR (Max Planck Institute Earth System Model, low resolution; Giorgetta et al., 2013), and (7) MRI-ESM1 (Meteorological Research Institute Earth System Model v1; Yukimoto et al., 2011). Both for NorESM1-ME and our model ensemble, we only investigate one realization of each scenario.

### 3.4 Cold-water coral positions

To estimate the potential impact of the Nordic Seas acidification on cold-water corals, we used habitat positions in longitude and latitude from the European Marine Observation and Data Network (EMODnet) Seabed Habitats (https://www.emodnet-seabedhabitats.eu; last access: 19 May 2020) together with information on depth from ETOPO1 (NOAA National Geophysical Data Center, 2020).

## 4 Methods

### 4.1 Spatial drivers of pH and $\Omega_{Ar}$

To identify the drivers of the observed spatial variability in surface pH and $\Omega_{Ar}$, we calculated pH and $\Omega_{Ar}$ by using spatially varying GLODAPv2 climatologies of specific drivers in Table 1, while keeping all other drivers constant (set to the spatial mean value of the Nordic Seas' surface waters). Because the relation between $C_T$ and $A_T$ is a proxy for the buffer capacity, we decided to look at their combined effect on pH, meaning that both changes in $C_T$ and $A_T$ are included in the calculations. Their combined effect we, from now on, refer to as $C_T + A_T$. First, pH and $\Omega_{Ar}$ were calculated with temperature being the only spatially varying climatology (pH($T$) and $\Omega_{Ar}(T)$). Thereafter, we used spatially varying temperature, $C_T$, and $A_T$ climatologies to calculate pH($T, C_T, A_T$) and $\Omega_{Ar}(T, C_T, A_T)$. Finally, the salinity variability was added to estimate pH($T, C_T, A_T, S$) and $\Omega_{Ar}(T, C_T, A_T, S)$. To estimate the contribution of each driver, the pH and $\Omega_{Ar}$ fields calculated with the different

spatially varying drivers were correlated with the actual pH and $\Omega_{\text{Ar}}$ of the Nordic Seas.

## 4.2   Temporal drivers of pH change

### 4.2.1   Present-day observational change

Measurements of temperature, salinity, $C_T$, and $A_T$ (Figs. S1–S4 in the Supplement), phosphate, and silicate from the datasets described in Sect. 3.1 were used to calculate pH and $\Omega_{\text{Ar}}$, using CO2SYS for MATLAB (Lewis and Wallace, 1998; van Heuven et al., 2011). pH was calculated on total scale at in situ pressure and temperature. Wherever nutrient data were missing, silicate and phosphate concentrations were set to 5 and $1\,\mu\text{mol}\,\text{kg}^{-1}$, respectively, which are representative values for the Nordic Seas. For the CO2SYS calculations, the dissociation constants of Lueker et al. (2000), the bisulfate dissociation constant of Dickson (1990), and the borate-to-salinity ratio of Uppström (1974) were used. This ratio has recently been shown to be suitable for the western Nordic Seas (Ólafsson et al., 2020a).

Present-day trends (1981–2019) in pH and $\Omega_{\text{Ar}}$ were determined for six different regions in the Nordic Seas, i.e., the Norwegian Basin (NB), the Lofoten Basin (LB), the Barents Sea Opening (BSO), the eastern Fram Strait (FS), the Greenland Sea (GS), and the Iceland Sea (IS; Fig. 1). These regions were chosen based on the data availability and were centered around stations and sections where repeated measurements are taken, but also to obtain a representation of the main surface water masses of the Nordic Seas. In the surface, the Norwegian Basin, Lofoten Basin, and Barents Sea Opening are influenced by relatively warm and salty northward-flowing Atlantic water, while the Greenland and Iceland Seas are influenced by relatively cold and fresh southward-flowing polar waters. As the Fram Strait surface is influenced by Atlantic and polar waters, we constrain the Fram Strait box to the east (hereinafter referred to as eastern Fram Strait) to ensure that it mostly represents Atlantic waters. The geographical range of each regional box is kept small so that the aliasing effects of latitudinal and longitudinal gradients are minimized.

Regional trends were computed from the annual means for five different depth intervals (0–200, 200–500, 500–1000, 1000–2000, and 2000–4000 m), using linear regression. The depth of 200 m sets the approximate lower limit for the impact of seasonal variations (Skjelvan et al., 2008). It was therefore chosen as the lower bound of the upper layer to keep all depths influenced by the seasonal cycle in one layer, that is, to minimize the number of layers where the trends may be affected by seasonal undersampling. The significance of the trends (at 95 % confidence level) were determined from the $p$ value of the $t$ statistic (as implemented in MATLAB's fitlm function). For the comparison of trends, 95 % confidence intervals of the slopes were determined by the use of the Wald method (as implemented in MATLAB's fitlm and coefCI functions).

The observed long-term changes in pH were decomposed into contributions from changes in temperature ($T$), salinity ($S$), $C_T$, and $A_T$ (Figs. S1–S4 and Tables S2–S5), following the procedure of Lauvset et al. (2015). First, the effect of each of these processes on the $CO_2$ fugacity ($f\!CO_2$) change is determined following Takahashi et al. (1993):

$$\frac{\mathrm{d}f\!CO_2}{\mathrm{d}t} = \frac{\partial f\!CO_2}{\partial T}\frac{\mathrm{d}T}{\mathrm{d}t} + \frac{\partial f\!CO_2}{\partial S}\frac{\mathrm{d}S}{\mathrm{d}t} + \frac{\partial f\!CO_2}{\partial C_T}\frac{\mathrm{d}C_T}{\mathrm{d}t}$$
$$+ \frac{\partial f\!CO_2}{\partial A_T}\frac{\mathrm{d}A_T}{\mathrm{d}t}. \tag{3}$$

The long-term mean values for the sensitivities (the $f\!CO_2$ partial derivatives) were approximated as in Fröb et al. (2019). Changes in $A_T$ and $C_T$ are driven by biogeochemical processes, transport, mixing, and dilution or concentration by freshwater fluxes, which is in direct proportion to the dilution or concentration of salinity. The freshwater effect can be separated by introducing salinity-normalized $C_T$ ($sC_T$) and $A_T$ ($sA_T$), as follows (Keeling et al., 2004; Lovenduski et al., 2007):

$$sC_T = \frac{S_0}{S}(C_T - C_0) + C_0; \quad sA_T = \frac{S_0}{S}(A_T - A_0) + A_0. \tag{4}$$

Here we set $S_0$ to 35 (Friis et al., 2003) and used the intercepts of Eqs. (6) and (7) in Nondal et al. (2009) as the non-zero freshwater end member ($A_0$ and $C_0$). Substituting $A_T$ and $C_T$ in Eq. (3) by Eq. (4) yields the following:

$$\frac{\partial f\!CO_2}{\partial C_T}\frac{\mathrm{d}C_T}{\mathrm{d}t} = \frac{sC_T - C_0}{S_0}\frac{\partial f\!CO_2}{\partial C_T}\frac{\mathrm{d}S}{\mathrm{d}t}$$
$$+ \frac{S}{S_0}\frac{\partial f\!CO_2}{\partial C_T}\frac{\mathrm{d}sC_T}{\mathrm{d}t} \tag{5}$$

$$\frac{\partial f\!CO_2}{\partial A_T}\frac{\mathrm{d}A_T}{\mathrm{d}t} = \frac{sA_T - A_0}{S_0}\frac{\partial f\!CO_2}{\partial A_T}\frac{\mathrm{d}S}{\mathrm{d}t}$$
$$+ \frac{S}{S_0}\frac{\partial f\!CO_2}{\partial A_T}\frac{\mathrm{d}sA_T}{\mathrm{d}t}. \tag{6}$$

Subsequently, the magnitude of each $f\!CO_2$ driver is converted to $[\text{H}^+]$ by using Henry's law ($[CO_2] = k_0 \times f\!CO_2$) and the expression for $\mathrm{d}[\text{H}^+]/\mathrm{d}[CO_2]$ from Eq. (1.5.87) in Zeebe and Wolf-Gladrow (2001), resulting in the following expression for $[\text{H}^+]$ change:

$$\frac{\mathrm{d}[\text{H}^+]}{\mathrm{d}t} = \frac{\mathrm{d}[\text{H}^+]}{\mathrm{d}[CO_2]}\frac{k_0 \times \mathrm{d}f\!CO_2}{\mathrm{d}t}. \tag{7}$$

Finally, $\text{H}^+$ in Eq. (7) was converted to pH by acknowledging that $\mathrm{dpH} = -([\text{H}^+]\ln(10))^{-1}\mathrm{d}[\text{H}^+]$. Here we consider the sulfate in Eq. (1) to have negligible impacts on the trends and did, therefore, not include it.

To control whether the observed pH changes are consistent with the changes in atmospheric $CO_2$, we additionally determined the pH change expected in seawater where the $p\text{CO}_2$

perfectly tracks the atmospheric $p\mathrm{CO}_2$ ($\mathrm{pH}_{\mathrm{perf}}$) for each region. This was achieved by adding the observed change in atmospheric $x\mathrm{CO}_2$ to the local mean $p\mathrm{CO}_2$ for the first year with observations and then calculating the pH with CO2SYS with the local temperature, salinity, $A_{\mathrm{T}}$, phosphate, and silicate and their respective changes as inputs. We applied no corrections for water vapor and atmospheric pressure, as the rates of change for $x\mathrm{CO}_2$ and $p\mathrm{CO}_2$ are proportional. Any deviation between observed pH change and $\mathrm{pH}_{\mathrm{perf}}$ is a consequence of changes in seawater $p\mathrm{CO}_2$ that are smaller/larger than in the atmosphere, i.e., a change in the air–sea $p\mathrm{CO}_2$ difference.

### 4.2.2 Model- and observation-based past and future changes

As described in Sect. 2, the total change in pH and saturation states does not only depend on local changes in $C_{\mathrm{T}}$, $A_{\mathrm{T}}$, temperature, salinity, and nutrients but also on the initial buffer capacity of the seawater. For the calculation of past and future pH changes, we use ESM data, which are usually biased, i.e., there is an offset between modeled fields and reality, and this also holds for the buffer capacity. In particular, NorESM1-ME has high $A_{\mathrm{T}}$ and low $C_{\mathrm{T}}$ relative to observations in deep waters, leading to biased high pH (Fig. S5) and saturation states (not shown). To alleviate this bias in our analysis of past and future pH and $\Omega_{\mathrm{Ar}}$, we applied the modeled change of temperature, salinity, $C_{\mathrm{T}}$, $A_{\mathrm{T}}$, phosphate, and silicate to the gridded GLODAPv2 climatology. Here, the modeled change between preindustrial era, present day, and future were calculated as the differences between 10-year means; i.e., 1850–1859, 1996–2005, and 2090–2099, respectively. We note that we could not center our present-day 10-year mean around the year 2002, to which the GLODAPv2 climatology is normalized, as the future scenarios start in 2006. After we obtained past and future states of the properties listed above, we calculated past and future pH, $\Omega_{\mathrm{Ar}}$, and $\Omega_{\mathrm{Ca}}$ in CO2SYS. Similar procedures have been used by Orr et al. (2005) and Jiang et al. (2019) to calculate future pH. Additionally, we used these data to calculate the drivers of past-to-present and present-day-to-future pH changes, following the methodology described in the previous section. Here we used a value of zero for the freshwater end members $A_0$ and $C_0$, as NorESM1-ME does not include any riverine input of $A_{\mathrm{T}}$ and $C_{\mathrm{T}}$.

To estimate the impact of acidification on the cold-water corals of the Nordic Seas, we calculated the mean saturation state in our region east of $0°\,\mathrm{E}$ and south of $64°\,\mathrm{N}$ for the preindustrial (PI) era, the present day, and for the future under the esmRCP2.6, esmRCP4.5, and esmRCP8.5 scenarios. The exclusion of the western and northern parts was done to constrain the mean to the Atlantic water where the cold-water corals are located. The saturation horizon was defined as the deepest vertical grid cell, where $\Omega_{\mathrm{Ar}} > 1$.

In order to facilitate a comparison with other model-based acidification studies, we have chosen to present the past and future changes for the surface ocean (i.e., $0\,\mathrm{m}$) in Sect. 5.3 and 5.5. However, in Sect. 5.2, where the observed changes of the upper $200\,\mathrm{m}$ are put into perspective with respect to past and future changes, we have calculated and presented the model mean over the upper $200\,\mathrm{m}$.

### 4.3 pH or H$^+$ change?

In a recent publication, Fassbender et al. (2021) recommend analyzing changes in H$^+$ concentrations, in addition to changes in pH, when comparing pH trends across water masses with different initial pH. The underlying reason is that a change in pH represents a relative change, and that it is possible to obtain the same pH changes across water masses with different change in H$^+$ concentration. We estimated the sensitivity of our results to the choice between pH and H$^+$ by plotting the change in H$^+$ concentration against the change in pH for a given change in $C_{\mathrm{T}}$ at various initial pH (Fig. 2). The different initial pH were obtained by varying the $C_{\mathrm{T}}$ over $A_{\mathrm{T}}$ ratio, and the calculations were done with a temperature and salinity of $5\,°\mathrm{C}$ and 35, respectively. For a given increase in $C_{\mathrm{T}}$ below $200\,\mu\mathrm{mol\,kg}^{-1}$, we see that the relationship between the H$^+$ and pH change is approximately linear in the Nordic Seas. The maximum $C_{\mathrm{T}}$ change in this study amounts to $170\,\mu\mathrm{mol\,kg}^{-1}$ in the surface waters under the esmRCP8.5 scenario. The choice between pH or H$^+$, therefore, has little impact on our results. The linear relationship breaks down with an increasing $C_{\mathrm{T}}$ over $A_{\mathrm{T}}$ ratio. The maximum pH change takes place at the buffer minimum, which is close to where $C_{\mathrm{T}} = A_{\mathrm{T}}$, approximately at $(pK_1 + pK_2)/2$ (Frankignoulle, 1994; Fassbender et al., 2017; Middelburg et al., 2020), which, in our example, is at pH 7.6. The linear relationship between the H$^+$ and pH change does, therefore, not hold for pH ranges where relatively low initial pH values are included, as is the case for the examples in Fassbender et al. (2021) and for larger $C_{\mathrm{T}}$ changes. In these cases, it is more appropriate to use H$^+$ for diagnosing ocean acidification.

### 4.4 Uncertainty analysis

There are several sources of uncertainties ($\sigma$) involved in our calculations of pH and $\Omega$, including measurement uncertainty ($\sigma_{\mathrm{mes}}$), mapping uncertainty ($\sigma_{\mathrm{map}}$) for the gridded product, and uncertainties related to dissociation constants ($\sigma_{K_x}$) used in the CO2SYS calculations. To estimate the total uncertainties in our calculations of pH and $\Omega$, we used the error propagation routine in the MATLAB version of CO2SYS (Orr et al., 2018). The uncertainties in the input parameters ($A_{\mathrm{T}}$, $C_{\mathrm{T}}$, temperature, salinity, phosphate, and silicate) were set to $\sigma_{\mathrm{mes}}$ for the single measurements and $\sqrt{\sigma_{\mathrm{mes}}^2 + \sigma_{\mathrm{map}}^2}$ for the mapped product, as well as for past and future estimates. As $\sigma_{\mathrm{mes}}$ and $\sigma_{\mathrm{map}}$, the product consistency from Olsen et al.

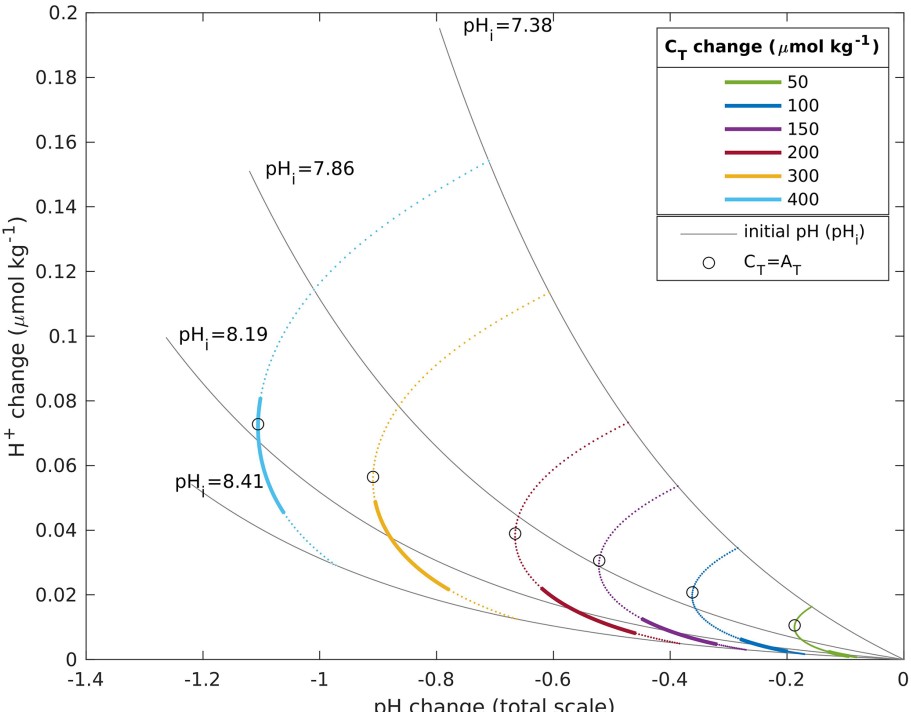

**Figure 2.** $H^+$ change plotted against pH change for six different increases in $C_T$ (colored lines) for a range of initial pH. The upper and lower ends of the colored lines represent an initial pH of 7.38 and 8.41, respectively. The bold part of the lines represents the pH range in the Nordic Seas' surface water in the GLODAPv2 climatology. The circles are plotted at the initial pH, where the initial and final $C_T$ over the $A_T$ ratio are centered around 1.

(2019) and the mapping error (3D field) from Lauvset et al. (2016) were used, respectively. The correlation between uncertainties in $A_T$ and $C_T$ were set to 0. This is a reasonable assumption, given that $C_T$ and $A_T$ are measured on different instruments using different analytical methodologies. In addition, including a positive correlation term would decrease the overall uncertainty, and we prefer a potential overestimation. For the dissociation constants, the default uncertainties in the errors.m function were used. From here on, the calculated uncertainties will be presented as $\sigma_{point}$, for discrete data, when $\sigma_{K_x}$ and $\sigma_{mes}$ are included, and $\sigma_{field}$, for 3D data, when $\sigma_{K_x}$, $\sigma_{mes}$, and $\sigma_{map}$ are included.

For the observations described in Sect. 3.1, the mean, maximum, and minimum uncertainties ($\sigma_{point}$) for our calculations of pH, $\Omega_{Ar}$, $\Omega_{Ca}$, and $pCO_2$ are listed in Table 2. Variations in the uncertainties arise from variations in temperature and salinity, which impact the uncertainty of the dissociation constants. While systematic uncertainties would tend to cancel out when calculating trends (i.e., comparing measurements from the same location but from different times), random uncertainties would not (Orr et al., 2018). Therefore, to estimate to what extent these uncertainties could impact our trend estimates, we further investigated whether there is any trend in the uncertainties (Figs. S6 and S7). This is discussed in Sect. 5.4.

**Table 2.** Uncertainties ($\sigma_{point}$, mean, max, and min) in pH, $\Omega_{Ar}$, $\Omega_{Ca}$, and $pCO_2$ (µatm), as calculated from the individual observations described in Sect. 3.1.

|                | Mean   | Max    | Min   |
|----------------|--------|--------|-------|
| pH             | 0.017  | 0.022  | 0.014 |
| $\Omega_{Ar}$  | 0.085  | 0.174  | 0.037 |
| $\Omega_{Ca}$  | 0.134  | 0.271  | 0.058 |
| $pCO_2$        | 14.387 | 53.608 | 5.901 |

For the GLODAPv2 estimate of preindustrial $C_T$, there is an additional uncertainty coming from the TTD method that was used to calculate the anthropogenic $CO_2$. He et al. (2018) published a thorough analysis of the different sources of uncertainty in this method and concluded that the overall uncertainty is 7.8 %–13.6 %. Combining this with the mapping errors, Lauvset et al. (2020) estimate that the global ocean anthropogenic carbon inventory calculated from the mapped fields is $167 \pm 29$ PgC. This results in an uncertainty of 0.02 in the preindustrial Nordic Seas upper layer pH.

In the trends of the uppermost layer (0–200 m), there is also an uncertainty related to seasonal undersampling. Most samples (about 60 % in total) from the datasets described in Sect. 3.1 were collected during spring and summer (April–September; Figs. S8–S13). The uneven sampling frequency

of different seasons introduces uncertainty in the annual means of the uppermost ocean layer and can lead to biases in our trend estimates. Unfortunately, there are not enough data to allow for de-seasonalization in order to remove such potential biases. Therefore, to estimate the effect of seasonal undersampling, we additionally calculated trends by using annual means containing samples from the productive season only, both for a longer period (April–September), to include both the spring bloom and the summer production, and for a shorter period (June–August), to include only the summer season.

Modeled future projections are uncertain due to incomplete understanding or parameterization of fundamental processes and different and unknown future carbon emission scenarios (Frölicher et al., 2016). Because this study primarily focuses on process understanding and the driving factors behind pH change, we do not consider model uncertainty in Sect. 5.3, 5.5, and 5.7.2, where the drivers of pH changes in the model projections are analyzed. However, in Sect. 5.6, where the future aragonite saturation horizon is presented, we do account for model uncertainty. The model-dependent uncertainty, here defined as the model spread, of the future saturation horizon under the esmRCP8.5 scenario, was estimated by adding the modeled change in $C_T$ and $A_T$ for each model of our ESM ensemble to the GLODAPv2 climatologies. Model differences in changes of temperature, salinity, phosphate, and silicate are neglected because they are minor in comparison to the effect of the changes in $C_T$ and $A_T$ (this is further discussed in Sect. 5.7.2). Internal climate variability is an additional source of model uncertainty that we do not explicitly account for in this study. However, a large part of this variability is eliminated because we use 10-year means for the future and past estimates of pH.

## 5 Results and discussion

This section is organized as follows: we will start to describe the present spatial distribution of pH and $\Omega_{Ar}$ and its drivers (Sect. 5.1). In Sect. 5.2, we give an overview of pH changes from the preindustrial era to 2100. Thereafter, we describe regional changes from the preindustrial era to the present day (Sect. 5.3), present-day changes (Sect. 5.4), and changes from the present day to the future (Sect. 5.5) and assess its impacts on cold-water corals (Sect. 5.6). In Sect. 5.7, we analyze the drivers of pH change in the different time periods.

### 5.1 Present-day spatial distribution of pH and $\Omega_{Ar}$

Due to the contrasting properties of Atlantic waters, here defined as waters with salinity > 34.5 (Malmberg and Désert, 1999; Nondal et al., 2009), and polar waters (defined as the waters with salinity < 34.5 detached from the Norwegian coast) that meet and mix in the Nordic Seas, there are large spatial gradients in surface (0 m) temperature, salinity, and

**Table 3.** Spatial correlation ($r$) and explained variance ($r^2$; in parenthesis) between pH and pH($T$), pH($T, C_T, A_T$), and pH($T, C_T, A_T, S$) and between $\Omega_{Ar}$ and $\Omega_{Ar}(C_T, A_T)$, $\Omega_{Ar}(C_T, A_T, T)$, and $\Omega_{Ar}(C_T, A_T, T, S)$ in the Nordic Seas' surface (0 m) waters. Numbers in bold indicate the significant correlation.

| Drivers | $(T)$ | $(T, C_T, A_T)$ | $(T, C_T, A_T, S)$ |
|---|---|---|---|
| pH | **0.58** (0.34) | **0.94** (0.89) | **1.00** (1.00) |
| $\Omega_{Ar}$ | **0.85** (0.73) | **1.00** (1.00) | **1.00** (1.00) |

chemical properties (Figs. 3 and S14). The Atlantic water, located in the eastern part of the Nordic Seas, is characterized by higher temperature, salinity, and $A_T$, while polar waters are colder and fresher with lower $A_T$. This results in a decrease in temperature, salinity, and $A_T$ from southeast to northwest. Within the Atlantic water, there is a tendency of increasing $C_T$ with decreasing temperature. This is largely as a consequence of the increased $CO_2$ solubility in colder water, i.e., a cooling of a water mass results in an increase in $C_T$ due to an uptake of $CO_2$ from the atmosphere. In polar waters, $C_T$ is lower than in Atlantic waters due to the lower $pCO_2$ (Fig. S14), and also as a result of the large freshwater export from the Arctic Ocean that dilutes not only $C_T$ but also $A_T$ and salinity.

The surface pH in the Nordic Seas increases from the Atlantic waters to the polar waters (Fig. 3). The correlation between the pH and the pH calculated with spatially varying temperature only (pH($T$)), keeping all other drivers constant, is 0.58. This means that temperature-induced variations (through the thermodynamic effect) are able to explain 34 % of the spatial variability in pH (Fig. 3 and Table 3). Adding $C_T + A_T$ and salinity contributions explains an additional 55 % and 11 %, respectively, of the spatial variability in pH. The effect of salinity is largest in the low-salinity regions, i.e., in polar waters and the Norwegian coastal waters. In contrast to what is suggested by directly correlating pH and $C_T + A_T$ (Table S9), the results in Table 3 show that $C_T + A_T$ are important contributors to the spatial variations in pH. This indicates that the influence of $C_T$ and $A_T$ on pH is masked out by temperature variations in Table S9 and Fig. S15, which can be explained by the two canceling effects that temperature has on pH (Jiang et al., 2019). For example, while the instantaneous thermodynamic effect of a drop in temperature leads to a pH increase, it also results in a drop in $pCO_2$, which subsequently leads to an anomalous $CO_2$ uptake from the atmosphere. This increases the $C_T/A_T$ ratio, which, in turn, causes a drop in pH that counteracts the initial thermodynamic affect.

The saturation state $\Omega_{Ar}$ shows an opposite pattern to pH, with low saturation states in polar waters and high saturation states in Atlantic water. From Fig. 3f, it becomes clear that the temperature effect on the solubility of $\Omega_{Ar}$ ($\Omega_{Ar}(T)$)

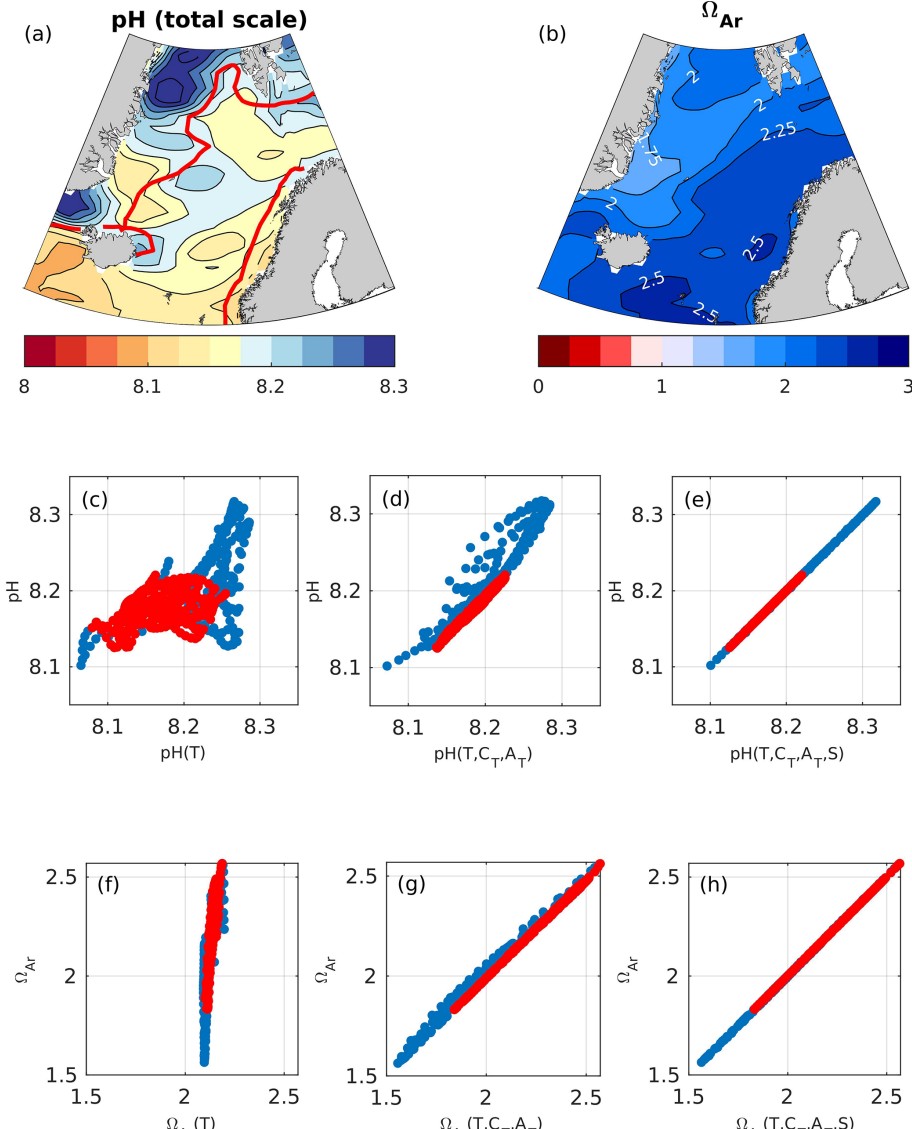

**Figure 3.** Maps of present-day surface (0 m) pH **(a)** and $\Omega_{Ar}$ **(b)**. The solid red line in panel **(a)** marks the border between Atlantic water (salinity > 34.5) and low-salinity waters (salinity < 34.5). The low-salinity waters include Norwegian coastal waters (constrained to the Norwegian coast) and polar waters (constrained to the northwestern part of the domain). pH and $\Omega_{Ar}$ plotted against variations induced by temperature **(c, f)**, temperature and $C_T + A_T$ **(d, g)**, and temperature, $C_T + A_T$, and salinity **(e, h)** in pH and $\Omega_{Ar}$, calculated as described in Sect. 4.1 in Atlantic water (red) and low-salinity waters (blue). Each circle represents a value from a single grid cell.

only explains 11 % of the observed $\Omega_{Ar}$ range, although it is able to explain 98 % TS4 of the variability. When adding $C_T + A_T$ contributions, the observed range in $\Omega_{Ar}$ is reproduced, and 100 % of the variability is explained. $C_T + A_T$ strongly influences $\Omega_{Ar}$ because, with an increasing $C_T$ to $A_T$ ratio, the $CO_3^{2-}$ concentration decreases. The $C_T$ to $A_T$ ratio itself strongly correlates with temperature as the $CO_2$ solubility increases with decreasing temperature and vice versa (Table S9). The strong correlation between $\Omega_{Ar}$ and temperature (Table S9) is, therefore, largely a result of the temperature effect on $C_T + A_T$ and, as such, the $CO_3^{2-}$ concentration

(Sect. 2 and Orr, 2011; Jiang et al., 2019). Thermodynamic-salinity-induced variations only have a minor contribution to the spatial variations in $\Omega_{Ar}$ (less than 1 %), and as for pH, the effect of salinity is more prominent in the low-salinity regions.

## 5.2 Overview of modeled and observed pH changes from the preindustrial era to the end of the 21st century

Here we give an overview of upper layer, taken to be the upper 200 m for both model and observations, pH changes

in the Nordic Seas from 1850 to 2100 (Fig. 4). Note that, in this section, we use the actual modeled pH data, and not the modeled change applied to observational data, and use this as an opportunity to evaluate the model's performance. The preindustrial average Nordic Seas surface pH estimated from GLODAPv2, corresponding to an atmospheric $CO_2$ of 280 ppm, and from NorESM1-ME, using year 1850 with an atmospheric $CO_2$ of 284 ppm, are in good agreement, with mean values of $8.21 \pm 0.02$ and $8.22 \pm 0.02$, respectively. From 1850 to 1980, the emission-driven NorESM1-ME simulates an average pH decline of 0.06 in the Nordic Seas, while the concentration-driven run simulates a drop of 0.05 (Fig. S5). The difference is caused by the slight deviation in atmospheric $CO_2$ between the emission-driven historical run and historical data (see Sect. 3.3).

For the period between 1981 and 2019, the modeled pH largely encompasses the observed one (within the spatial standard deviations), showing that the pH of the Nordic Seas' surface water is reasonably well simulated. The pH trend estimated from the observations for this period, $-2.64 \pm 0.31 \times 10^{-3}$ yr$^{-1}$, is not significantly different (at the 95 % confidence level) from the modeled pH trend, $-2.21 \pm 0.04 \times 10^{-3}$ yr$^{-1}$. Because the pH trend calculated from the observational data is based on discrete samples with a limited spatial and temporal coverage, its representativeness for the entire Nordic Seas is questionable, and we do not expect an exact agreement with the model. For example, the stronger trend obtained from the observational data might be a result of the samples at the beginning of the period being biased to regions with higher pH.

The future evolution of upper layer pH in the Nordic Seas depends strongly on the $CO_2$ emission scenario (Fig. 4). In the esmRCP2.6 scenario, where the $CO_2$ emissions are kept within what is needed to limit global warming to 2 °C (van Vuuren et al., 2011b), pH drops by 0.04 from 2020 to 2099 and reaches a value of $8.03 \pm 0.01$. Note that, in this scenario, there is a peak and decline, related to the overshoot profile of the atmospheric $CO_2$ concentration, with a minimum pH value in mid-century. For the esmRCP4.5 scenario, which corresponds roughly to the currently pledged $CO_2$ emission reductions under the Paris Agreement, the surface pH is simulated to drop by about 0.15, reaching an average value of $7.93 \pm 0.01$ by the end of the 21st century. Under the high-$CO_2$ esmRCP8.5 scenario, NorESM1-ME simulates the pH to decrease by 0.40 between 2020 and 2099 to an average value of $7.67 \pm 0.02$. This equals a pH decline of approximately $-5.00 \times 10^{-3}$ yr$^{-1}$. The model-related uncertainty in the esmRCP8.5 scenario, measured as the intermodel spread of pH in 2099, displays a pH range of 7.59–7.79 in the surface layer (Figs. 4, S5). This spread is larger than that observed in the concentration-driven simulations with the same models, 7.69–7.75, as expected from the increased degrees of freedom brought about by the interactive atmospheric $CO_2$. Within the emission-driven model ensemble, the pH decline from preindustrial era to the end of the 21st century, as simu-

lated by NorESM1-ME, is among the strongest, which most likely is a result of a simulated stronger increase in atmospheric $CO_2$. A full analysis of the reasons behind the intermodel spread is beyond the scope of this paper.

The simulated Nordic Seas average upper layer pH is 0.11 higher than the global average in 1850, which is related to the undersaturation of $CO_2$ in the surface waters of the Nordic Seas (Jiang et al., 2019). Our global average pH is about 0.1 lower than that estimated by, e.g., Jiang et al. (2019) for the surface ocean due to our consideration of a 200 m thick upper layer. The difference between the simulated upper layer pH of the global ocean and the Nordic Seas is decreasing with time. By the end of the 21st century, the Nordic Seas upper layer pH is 0.03, 0.07, and 0.08 higher than the global average for the esmRCP8.5, esmRCP4.5, and esmRCP2.6 scenarios, respectively. This is partially a result of the colder waters of the Nordic Seas, which gives them a lower buffer capacity. Additionally, in esmRCP8.5, there is an increase in the $p\mathrm{CO_2}$ undersaturation of the global ocean that increases the global average pH (Fig. S16). Other factors driving the decreasing pH difference between the global ocean and the Nordic Seas can be differential heating. A quantitative assessment of the drivers is beyond the scope of this paper.

## 5.3 Modeled pH and $\Omega_{\mathrm{Ar}}$ changes from the preindustrial era to the present day

In this and the following sections, we present temporal changes in pH and $\Omega_{\mathrm{Ar}}$. Note that results for the modeled changes refer to the 0 m surface, unlike the 0–200 m depth range that we use for the upper layer in Sect. 5.2 and 5.4.

From the preindustrial era to the present, the spatial pattern of changes in surface pH and $\Omega_{\mathrm{Ar}}$ are similar (Fig. 5). The strongest decreases, reaching $-0.12$ and $-0.55$, respectively, are found in Atlantic water along the Norwegian coast for both pH and $\Omega_{\mathrm{Ar}}$. The smallest change is found in polar waters (see a more in-depth discussion in Sect. 5.7.2). The corresponding maps for $H^+$ (Fig. S17) show a similar spatial distribution as for pH. Due to the longer ventilation timescales of deeper waters, the pH decrease weakens with depth. As shown in the section across 70° N (Fig. 6), waters below 2500 m are nearly unaffected. While the entire water column remains saturated with respect to calcite, the saturation horizon ($\Omega = 1$) of aragonite shoaled from a mean depth of 2200 m (uncertainty range of 2100–2400 m) during the preindustrial era to a present-day mean depth of 2000 m (uncertainty range of 1700–2300 m) across this specific section. Note that these depths were obtained from the contour interpolation when creating Fig. 6, which has a finer vertical resolution than the GLODAPv2 climatology.

## 5.4 Observed present-day changes in pH and $\Omega_{\mathrm{Ar}}$

Regional trends in observed seawater pH between 1981 and 2019 for five different depth intervals are presented in Fig. 7

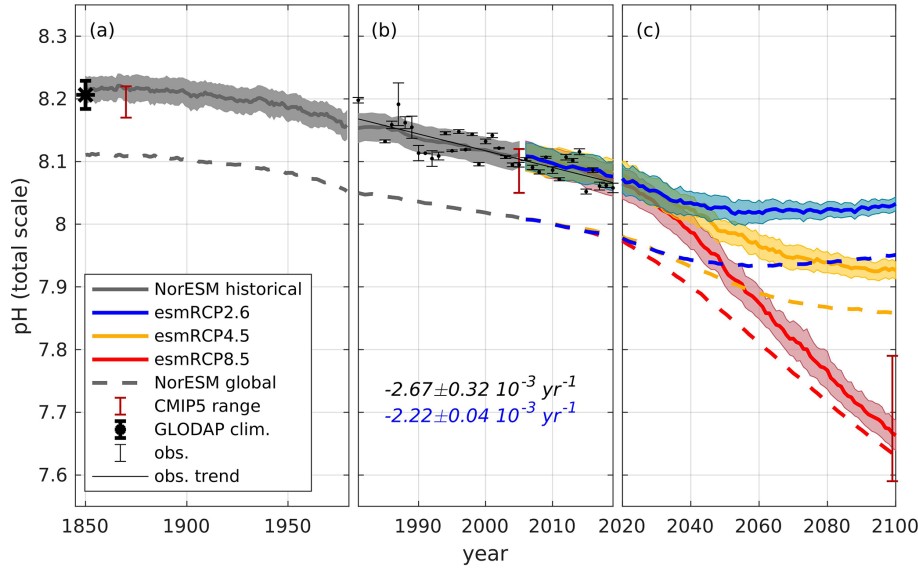

**Figure 4.** pH evolution, averaged over the Nordic Seas' surface waters (0–200 m), from 1850 to 2100, separated into **(a)** past (1850–1980), **(b)** present day (1981–2019), and **(c)** future (2020–2100). Black dots with error bars show the observed annual mean pH, with standard deviations (due to spatial/seasonal variations) determined from all available observations in the Nordic Seas, as shown in Fig. 1. The solid black line shows the trend calculated from these observations. The gray, red, yellow, and blue solid lines show NorESM1-ME output for emission-driven historical and future (esmRCP8.5, esmRCP4.5, and esmRCP2.6) simulations, respectively, where the shading depicts the spatial variation (standard deviation). Note that the atmospheric $CO_2$ increase, as simulated by NorESM1-ME for 1850 to 2005, deviates by 14 ppm from the actual measured increase, which results in a simulated pH decrease that is 0.01 stronger than expected (see Sect. 3.3). The red vertical bars display the pH range in the CMIP5 model ensemble for the historical and esmRCP8.5 simulations. The figure illustrates the actual modeled pH data and not the modeled change applied to observational data. The dashed lines show the evolution of global surface ocean pH from the same simulations. The black asterisks (1850) with error bars show an estimate of the preindustrial mean pH, with the spatial standard deviation derived from the GLODAPv2 mapped product, as described in Sect. 3.2. The numbers in black and blue show the calculated and significant linear trend, with standard errors from the observations and the model, respectively, for the period of 1981–2019.

and Table 4. The corresponding trends in $H^+$ are shown in Fig. S18 and Table S10. In the upper layer (0–200 m), significant pH trends of $2$–$3 \times 10^{-3}$ $yr^{-1}$ are found in all basins, except for the Barents Sea Opening. The uncertainties (standard errors) of these trends are between $\pm 0.2 \times 10^{-3}$ and $\pm 0.8 \times 10^{-3}$ $yr^{-1}$. Due to the difference in sampled years, we cannot robustly compare the magnitude of trends between the basins. Skjelvan et al. (2014) also found significant trends in the upper 200 m pH of the Norwegian and Lofoten basins and of the Greenland Sea for the period of 1981–2013. Our estimated trend in the Norwegian Basin of $-3.04 \pm 0.32 \times 10^{-3}$ $yr^{-1}$ is weaker than their $-4.1 \times 10^{-3}$ $yr^{-1}$ trend, which can be a result of different sampling period and slightly different definition of regions. However, our trend estimates in the Greenland Sea and Lofoten Basin of $-2.19 \pm 0.37 \times 10^{-3}$ and $-2.40 \pm 0.23 \times 10^{-3}$ $yr^{-1}$, respectively, agrees well with the trend of $-2.3 \times 10^{-3}$ $yr^{-1}$ that they calculated for both regions. The nonsignificant trend we find in the Barents Sea Opening is also in agreement with the results of Skjelvan et al. (2014). In contrast to their results, we obtained a significant trend in the eastern Fram Strait, which may be a result of the larger time span of our dataset. As expected from the generally longer ventilation timescales of deep waters,

the trends in pH decline with depth. Significant trends are detected down to 2000 m layer TS5 in the Greenland Sea, in agreement with Skjelvan et al. (2014), and also in the Iceland Sea and in the Norwegian Basin. In the Lofoten Basin and eastern Fram Strait, the decrease in pH is significant down to the 1000 and 500 m TS6 layers, respectively. As for the upper layer, no significant trend is found in the 200–500 m layer in the shallow Barents Sea Opening.

Trends of aragonite saturation states are shown in Fig. 8 and Table 5. As for pH, the rate of change is strongest in the upper layer. For $\Omega_{Ar}$, the decline is of CE2 the order of $10^{-2}$ $yr^{-1}$ and significant in all regions, except for the Greenland Sea. The weak decline in the Greenland Sea surface layer is a result of a smaller increase in $C_T$ in combination with relatively strong increases in $A_T$ and temperature, which counteracts the effect of $C_T$ on the saturation states (while the temperature increase amplifies the pH decline; see Sect. 2). The reduction in $\Omega_{Ar}$ is significant down to 2000 m TS7 layer in the Norwegian Basin and the Greenland and Iceland seas. In the other regions, no significant decline has occurred below the surface layer. In the depth layers considered, aragonite undersaturation occurs in the 2000–4000 m layer. The waters in the depth range 1000–2000 m are close to the limit

**Biogeosciences, 19, 1–34, 2022**  https://doi.org/10.5194/bg-19-1-2022

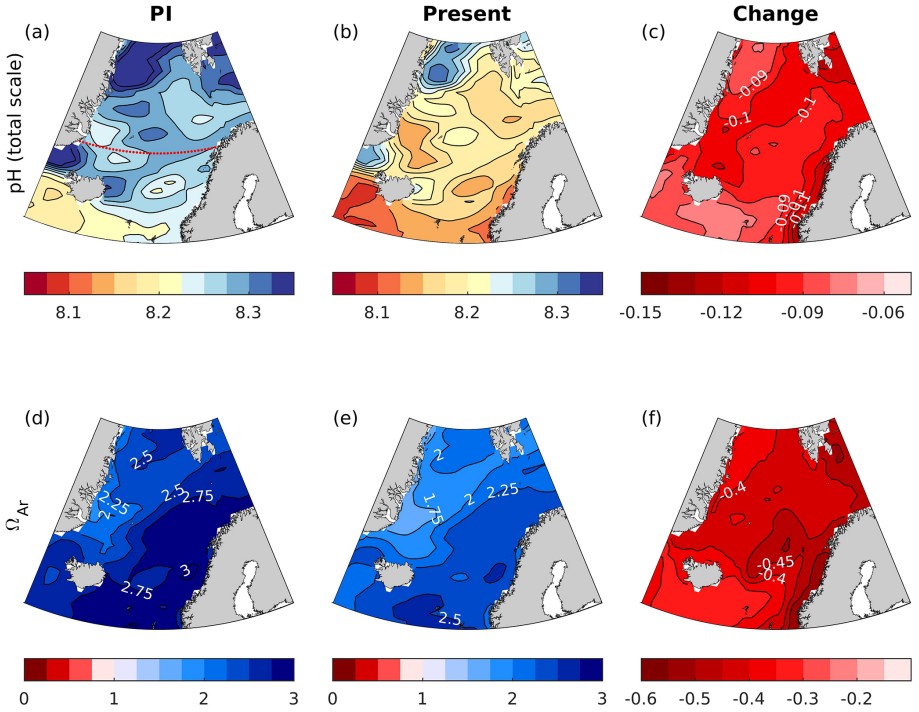

**Figure 5.** Maps of surface water (0 m) pH and $\Omega_{Ar}$ for the preindustrial (PI) era (1850–1859), the present day (1996–2005), and the change between the two periods. The maps were calculated from the GLODAPv2 gridded climatologies (Lauvset et al., 2016), applying the simulated changes by the emission-driven NorESM1-ME, as explained in Sect. 4.2. Note that the increase in atmospheric $CO_2$ in NorESM1-ME is 13 % higher than the observed record between 1850 and 2005, resulting in simulated a decrease in surface pH that is approximately 0.01 too strong (see Sect. 3.3). The dotted red line in panel **(a)** shows the location of the cross section presented in Fig. 6.

**Table 4.** pH trends $\pm$ standard error ($10^{-3}$ yr$^{-1}$) calculated from the data presented in Fig. 7 in the Norwegian Basin (NB), Lofoten Basin (LB), Barents Sea Opening (BSO), Fram Strait (FS), Greenland Sea (GS), and Iceland Sea (IS; Fig. 1). Bold numbers indicate that the trends are significantly different from zero.

| Depth (m) | NB | LB | BSO | FS | GS | IS |
|---|---|---|---|---|---|---|
| 0–200 | **−3.04 ± 0.32** | **−2.40 ± 0.23** | −1.67 ± 0.77 | **−2.53 ± 0.74** | **−2.19 ± 0.37** | **−3.10 ± 0.30** |
| 200–500 | **−2.22 ± 0.32** | **−1.89 ± 0.31** | −1.05 ± 0.82 | **−1.49 ± 0.42** | **−1.61 ± 0.22** | **−2.51 ± 0.27** |
| 500–1000 | **−1.17 ± 0.27** | **−2.27 ± 0.46** | | −1.09 ± 0.52 | **−1.52 ± 0.18** | **−1.84 ± 0.29** |
| 1000–2000 | **−0.65 ± 0.22** | −0.80 ± 0.40 | | −0.55 ± 0.81 | **−1.36 ± 0.15** | **−1.3 ± 0.21** |
| 2000–4000 | 0.46 ± 0.55 | −0.22 ± 0.51 | | −0.03 ± 0.69 | −0.31 ± 0.23 | |

of undersaturation. The smallest values in this layer are 1.05, 1.07, 0.99, 1.02, and 1.01 for the Norwegian Basin, Lofoten Basin, eastern Fram Strait, Greenland Sea, and Iceland Sea, respectively. Considering the associated uncertainties of 0.06 (Table 2), this is indistinguishable from undersaturation in all regions, except for the Lofoten Basin. In contrast to Skjelvan et al. (2014), who only found a significant negative trend in the upper 200 m layer of the Norwegian Basin, we are now, with the longer time series, able to state that there is a significant decrease in $\Omega_{Ar}$ in several regions and at several depth layers.

During the period 1981–2019, we detect trends in the uncertainties of pH and $\Omega_{Ar}$ (Figs. S6 and S7), reaching

$-0.04 \times 10^{-3}$ yr$^{-1}$ and $0.53 \times 10^{-3}$ yr$^{-1}$, respectively. These are, however, about 2 orders of magnitude smaller than the trends in pH and $\Omega_{Ar}$, and they do, therefore, not significantly impact the interpretation of our results.

## 5.5 Modeled pH and $\Omega_{Ar}$ changes from the present day to future

In this section, we go into regional details of future pH and $\Omega_{Ar}$ changes under the esmRCP2.6 and the esmRCP8.5 scenarios. The results are presented for the surface (0 m) and not for the upper layer 0–200 m, as in Sect. 5.2 and 5.4.

In esmRCP2.6, a pH decline of 0.06–0.11 in the surface waters is simulated between the present day (1996–2005) and

**Table 5.** $\Omega_{Ar}$ trends $\pm$ standard error ($10^{-3}$ yr$^{-1}$), calculated from the data presented in Fig. 8, in the Norwegian Basin (NB), Lofoten Basin (LB), Barents Sea Opening (BSO), Fram Strait (FS), Greenland Sea (GS), and Iceland Sea (IS; Fig. 1). Bold numbers indicate that the trends are significantly different from zero.

| Depth (m) | NB | LB | BSO | FS | GS | IS |
|---|---|---|---|---|---|---|
| 0–200 | **−11.97±3.25** | **−8.45±1.18** | **−8.29±3.54** | **−11.61±3.13** | −4.05±3.21 | **−11.20±2.22** |
| 200–500 | **−5.57±2.51** | −1.76±2.17 | 3.94±3.01 | −2.06±1.60 | **−3.19±0.61** | **−6.37±0.74** |
| 500–1000 | **−4.28±1.25** | −5.55±3.38 | | −1.11±1.46 | **−2.98±0.52** | **−4.52±0.71** |
| 1000–2000 | **−3.49±1.24** | 0.03±1.76 | | 0.65±3.08 | **−2.98±0.59** | **−2.57±0.50** |
| 2000–4000 | 3.67±1.82 | 0.33±1.57 | | 1.13±1.53 | 0.53±0.80 | |

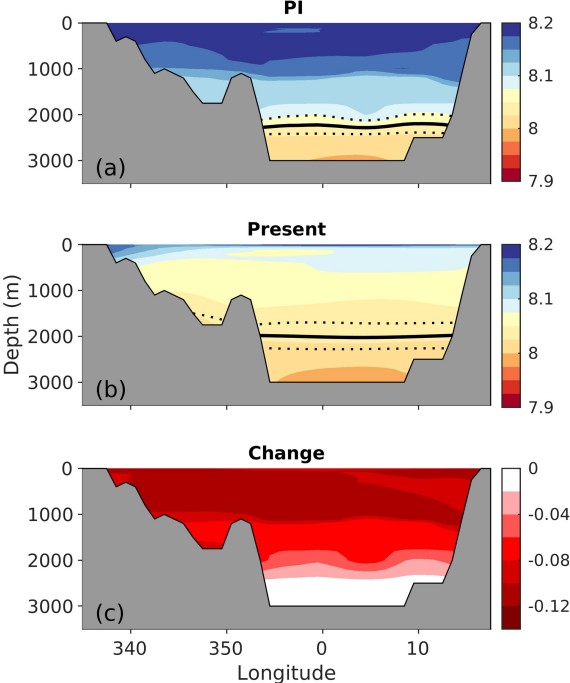

**Figure 6.** Zonal cross sections (at 70° N) of the preindustrial (1850–1859) and the present (1996–2005) pH and the change between the two periods. Note that the simulated increase in atmospheric $CO_2$ of NorESM1-ME is 13 % higher than the observed record between 1850 and 2005, resulting in a simulated decrease in surface pH that is approximately 0.01 too strong (see Sect. 3.3). The solid black line shows the saturation horizon of aragonite ($\Omega_{Ar} = 1$). The dashed lines show the associated uncertainties ($\sigma_{field}$).

the future (2090–2099; Fig. 9c). The largest pH decreases are found in polar waters, leading to a weakening of the present-day zonal pH gradient. Surface $\Omega_{Ar}$ is projected to decrease by about 0.2–0.5 under esmRCP2.6, with the largest drops taking place in polar waters. Surface waters remain supersaturated with respect to both calcite and aragonite. Interestingly, the strongest ocean acidification occurs at depths of 1000–2000 m in this scenario (Fig. 10d), which leads to a shoaling of the aragonite saturation horizon to a depth of

1100 m (uncertainty range of 800–1200 m). This is discussed in more detail in Sect. 5.7.2.

Under the esmRCP8.5 scenario, surface pH drops by about 0.4–0.5 between the present day and the future (Fig. 11), with the largest decreases in polar waters. Surface $\Omega_{Ar}$ drops by around 1.1–1.3. In contrast to esmRCP2.6, the largest decline in $\Omega_{Ar}$ take place in Atlantic water. The reason behind this is discussed in Sect. 5.7.2. The strong ocean acidification in this scenario leads to a reversal of the pH depth dependency, so that pH increases from surface to depth by the end of the 21st century (Fig. 10c). Here, the anthropogenic carbon input at the surface overrides the effect of pressure and organic matter remineralization on the vertical pH gradient. The change in $\Omega_{Ar}$ is large enough to bring the entire water column, and, consequently, also the entire seafloor, to aragonite undersaturation. The only exception is a thin surface layer (above $30 \pm 10$ m) in the Atlantic water region. For all emission scenarios, the spatial distribution of $H^+$ and its change (shown in Figs. S19 and S20) are similar to that of pH.

## 5.6 Implications for cold-water corals

Cold-water corals build their structures out of aragonite, which is the more soluble form of calcium carbonate. These corals can, to some degree, compensate for aragonite undersaturation in seawater by increasing their internal pH by 0.3–0.6 (McCulloch et al., 2012; Allison et al., 2014). For some time, they can, therefore, continue to calcify in waters with $\Omega_{Ar} < 1$. However, the calcification rates and breaking strength of the structures of the most abundant coral organism, *Lophelia pertusa*, is reduced under such conditions (Hennige et al., 2015). Furthermore, dead coral structures, which compose the major part of the reefs, cannot resist corrosive waters and experience increased dissolution rates at $\Omega_{Ar} < 1$. Cold-water coral reefs, along with their ecosystems, are consequently likely to collapse if the water they live in becomes undersaturated with respect to aragonite. It has been estimated that about 70 % of the cold-water corals globally will be below the aragonite saturation horizon by the end of the century under high emission scenarios (Guinotte et al., 2006; Zheng and Cao, 2014).

Most of the reef sites that have been identified in the Nordic Seas (321 out of the 324 within the region defined

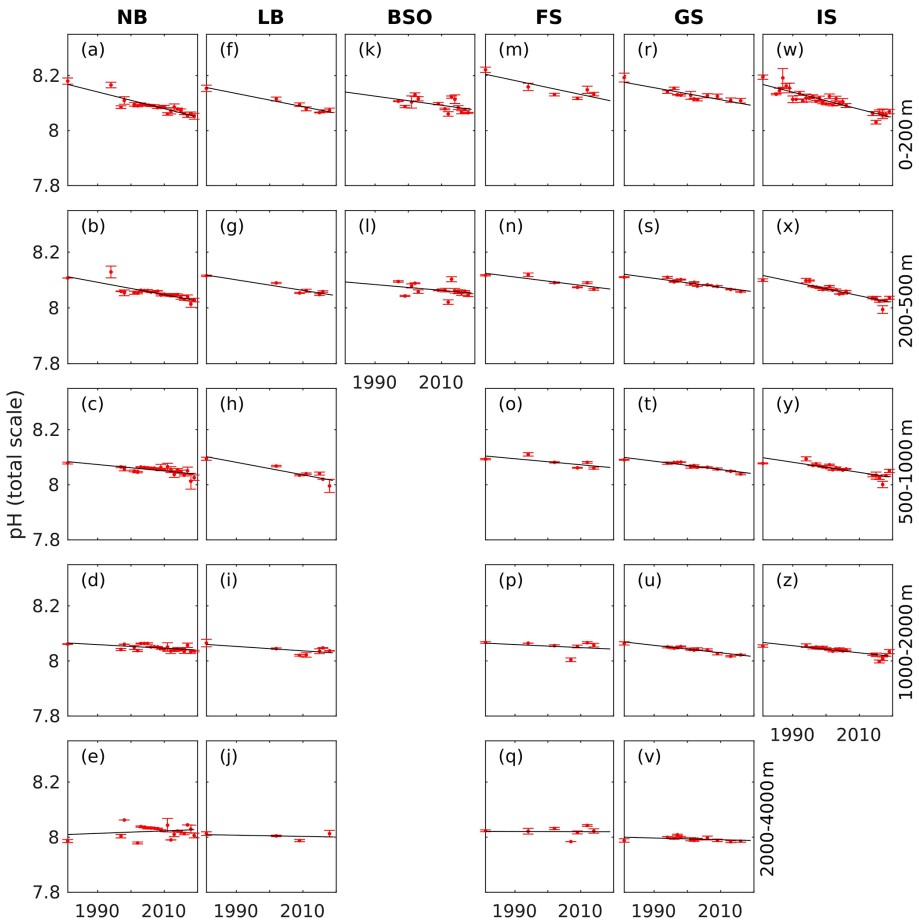

**Figure 7.** Annual mean pH (red dots) with standard deviation (error bars) at five different depth intervals in the Norwegian Basin (NB), Lofoten Basin (LB), Barents Sea Opening (BSO), Fram Strait (FS), Greenland Sea (GS), and Iceland Sea (IS; Fig. 1), which is calculated as described in Sect. 4.2. The solid black line show the trend estimate from the linear regression.

in Fig. 1) are at depths of 0–500 m (Fig. 12; see also Buhl-Mortensen et al., 2015). The aragonite saturation horizon estimated from the GLODAPv2 climatology for the present climate is at 2000 m, with an uncertainty range of 1750–2500 m. Note that the uncertainty range of the depth of the saturation horizon is not equally distributed around the mean because the uncertainty analysis is done for the saturation state from which the depth distribution is calculated. From the discrete measurements, we also see that the waters in the depth range 1000–2000 m are close to being undersaturated with respect to aragonite (Sect. 5.4). For the time being, the saturation horizon is, thus, well below the majority of the cold-water corals in the Nordic Seas.

In the esmRCP2.6 scenario, NorESM1-ME projects that the aragonite saturation horizon will shoal to 900 m (uncertainty of 800–1100 m), while, in the esmRCP4.5 scenario, the saturation horizon is projected to shoal to 600 m depth (uncertainty of 400–700 m) by the end of this century. This implies that the deepest observed reefs will be exposed to corrosive waters and, thus, experience elevated costs of calci-

fication and dissolution of dead structures. The majority (315 out of 324) of the coral sites in the Nordic Seas are, however, found at shallower depths than the projected saturation horizon with uncertainty, although the margins are small. Also, Gehlen et al. (2014) and García-Ibáñez et al. (2021) suggested that cold-water corals in the subpolar North Atlantic will be exposed to corrosive waters if the 2 °C goal (which is the aim of RCP2.6) is not met. In the esmRCP8.5 scenario, NorESM1-ME projects the whole water column below 20 m (uncertainty of 10–20 m) to be undersaturated with respect to aragonite at the end of this century, such that all cold-water coral reefs in the Nordic Seas will be exposed to corrosive waters. For esmRCP8.5, the NorESM1-ME results are consistent with our CMIP5 model ensemble that suggests that the future saturation horizon lies in the range of 0 and 100 m. Comparison with the CMIP5 ensemble is not possible for esmRCP2.6 and esmRCP4.5 because very few of the models have performed emission-driven runs under these scenarios. However, NorESM1-ME simulates one of the stronger pH declines in all depth layers considered in Fig. S5 (Table S6),

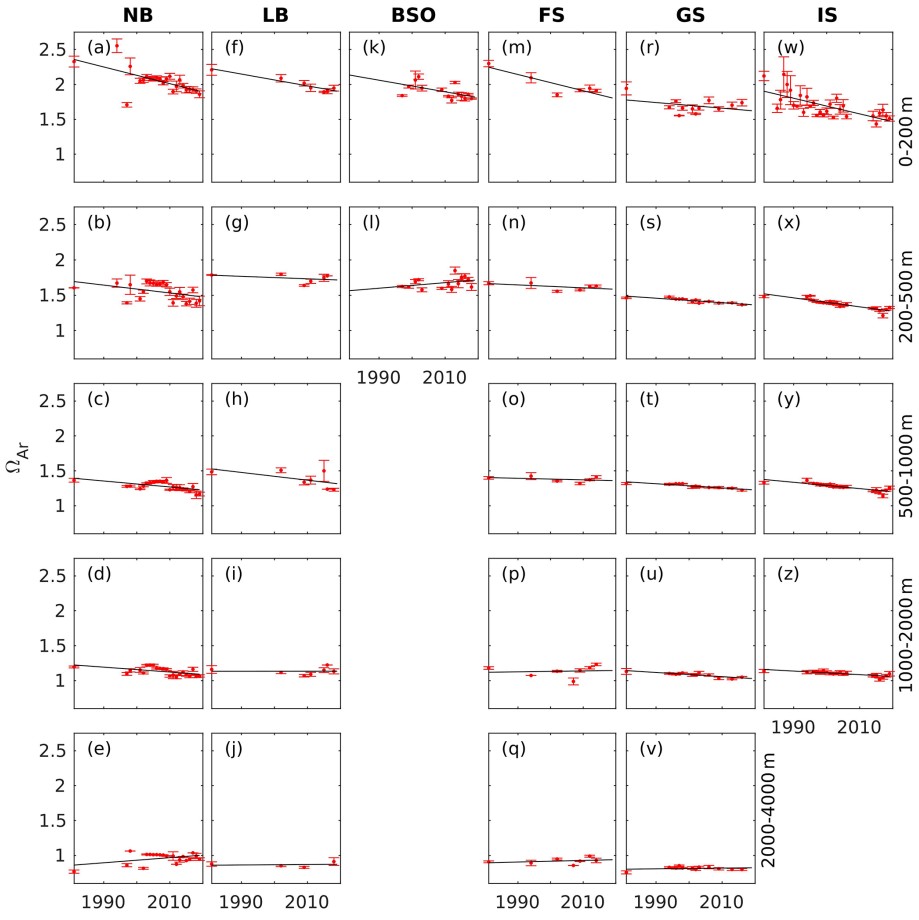

**Figure 8.** Annual mean $\Omega_{Ar}$ (red dots) with standard deviation (error bars) at five different depth intervals in the Norwegian Basin (NB), Lofoten Basin (LB), Barents Sea Opening (BSO), Fram Strait (FS), Greenland Sea (GS), and Iceland Sea (IS; Fig. 1), which is calculated as described in Sect. 4.2. The solid black line show the trend estimate from the linear regression.

and has also been shown to be on the upper end of the absorption of anthropogenic carbon in the Arctic Ocean (Terhaar et al., 2020a), suggesting that our estimates of the future saturation horizon lie in the shallower end of possible future states.

### 5.7 Drivers of ocean acidification

#### 5.7.1 Present-day drivers

To understand the causes behind the observed pH changes presented in Sect. 5.4, we decompose the trends into their different drivers as described in Sect. 4.2 (Fig. 13). In the upper layer (i.e., 0–200 m), the pH decrease in the period 1981–2019 is in agreement (within 95 % confidence) with the pH change expected from the increase in atmospheric $CO_2$, except for in the Norwegian Basin and the Iceland Sea, where the trends are stronger. This is related to a faster increase in the seawater $pCO_2$ compared with that of the atmosphere (Fig. S21 and Table S11), meaning that the $pCO_2$ undersaturation of the Norwegian Basin and the Iceland Sea

has decreased. We note that this diminishing undersaturation is sensitive to seasons. In the Norwegian Basin, there is no significant decrease in the undersaturation if using data from only April to September or June to August. In the Iceland Sea, the decreasing undersaturation is absent for April–September, but it becomes stronger than the annual mean if using data only from June–August. The sensitivity to the choice of seasons indicates that the strong positive trend in the air–sea $pCO_2$ difference, as seen in our dataset, can be a result of seasonal undersampling, and that this should be verified with a larger dataset. This information notwithstanding, diminishing $pCO_2$ undersaturation has been observed in earlier studies of the North Atlantic (Lefèvre et al., 2004; Olsen et al., 2006; Ólafsson et al., 2009; Metzl et al., 2010; Skjelvan et al., 2014) and could be a result of a change in any of the mechanisms underlying the $pCO_2$ undersaturation in surface waters of the Nordic Seas (see Sect. 1), including the cooling of northward flowing Atlantic waters, primary production, and the outflow of $pCO_2$ undersaturated waters from the Arctic Ocean. One other possible mechanism was suggested in Olsen et al. (2006) and Anderson and Olsen

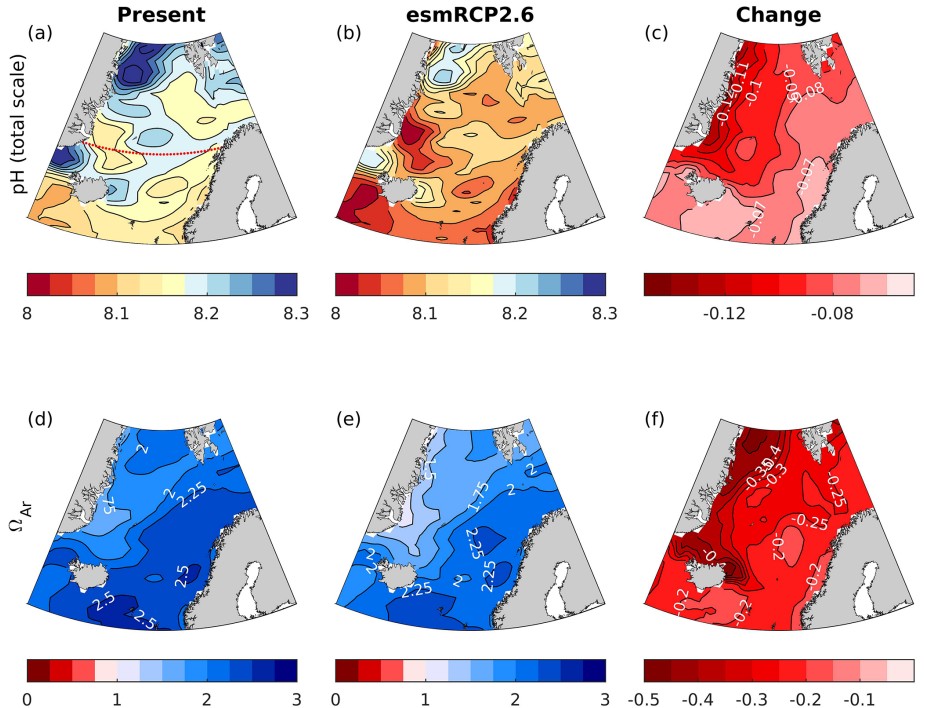

**Figure 9.** Maps of surface water (0 m) pH and $\Omega_{Ar}$ for the present day (1996–2005) and the esmRCP2.6 future (2090–2099), as well as the changes between the periods. The data input of the maps is based on GLODAPv2 gridded climatologies combined with the change from the NorESM1-ME. The dotted red line in panel **(a)** shows the location of the cross section presented in Fig. 10.

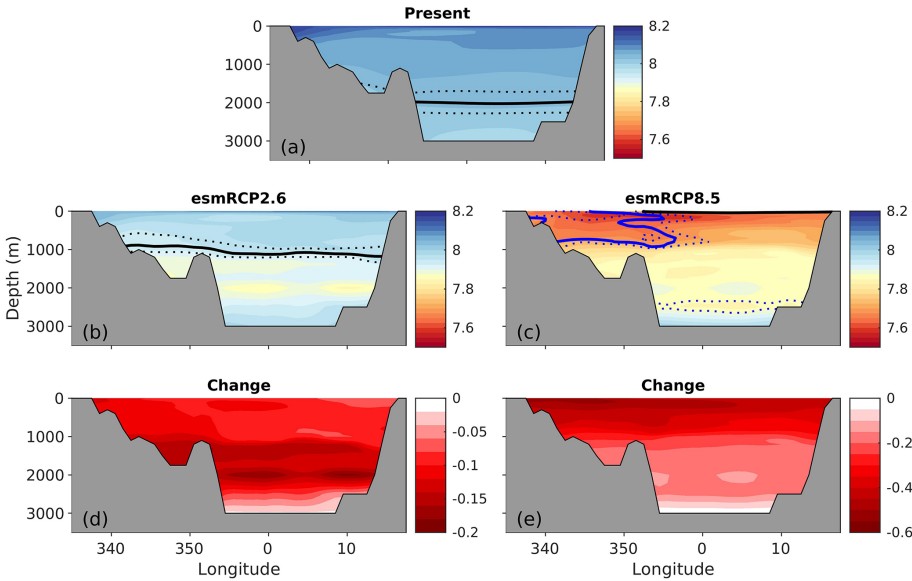

**Figure 10.** Zonal cross sections (at 70° N) of present (1996–2005) and future (2090–2099) pH under the emission-driven esmRCP2.6 and esmRCP8.5 scenarios, along with the change between the periods. The solid and dotted black lines show the saturation horizon of aragonite ($\Omega_{Ar} = 1$) with uncertainty ($\sigma_{field}$). The solid and dotted blue lines show the corresponding for calcite ($\Omega_{Ca} = 1$).

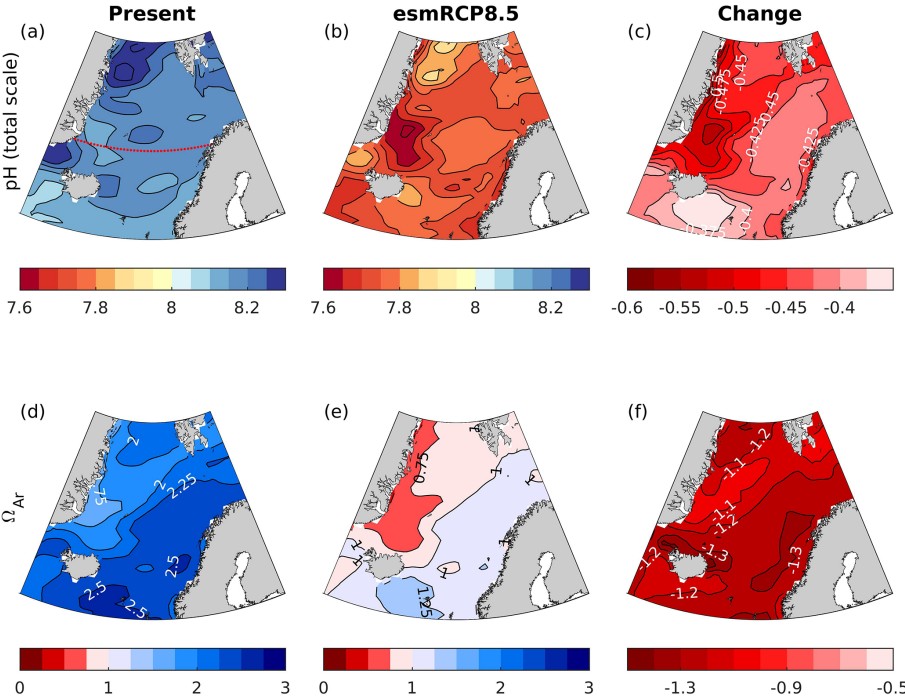

**Figure 11.** Maps of surface water (0 m) pH and $\Omega_{Ar}$ for the present (1996–2005) and the esmRCP8.5 future (2090–2099), as well as the changes between the periods. The data input of the maps is based on GLODAPv2 gridded climatologies combined with the change from the NorESM1-ME. The dotted red line in panel **(a)** shows the location of the cross section presented in Fig. 10.

(2002), where they associated the fast increase in seawater $p$CO$_2$ with a large advective supply of anthropogenic carbon from the south and corresponding changes in the buffer capacity (see also Terhaar et al., 2020b).

The main driver of the present-day (1981–2019) pH decrease in the upper layer is increasing $C_T$, which is primarily caused by biogeochemical processes ($C_T$bg), including increasing anthropogenic carbon, along with a small freshwater contribution ($C_T$fw) caused by an increasing salinity (Fig. S2). The increasing salinity also results in an increasing $A_T$ (Fig. S4). As seen in Fig. 13, the freshwater components of $C_T$ and $A_T$ are of equal size but opposite sign, and there is, therefore, no net effect of freshwater fluxes on the pH change (see Sarmiento and Gruber, 2006, for a theoretical explanation). Also, the thermodynamic effect of increasing salinity on pH is negligible. This increasing salinity of the Nordic Seas is a result of changes in the inflowing Atlantic water related to subpolar gyre strength (Holliday et al., 2008; Lauvset et al., 2018). The contribution of the biogeochemical component of $A_T$ is generally negligible, except in the Barents Sea Opening, where it explains the lack of a significant pH decline (Fig. 7). In our dataset, the effect of changes in temperature on pH in the upper layer is relatively small. In contrast to several studies pointing towards a warming of the Nordic Seas (e.g., Holliday et al., 2008; Blindheim and Østerhus, 2013; Lauvset et al., 2018; Ruiz-Barradas et al., 2018), the Barents Sea Opening, the eastern Fram Strait, and the Iceland

Sea show no significant change in temperature. This might be an artifact of an unequal distribution of sampling over the seasons. When calculating trends with all available temperature data, and not only those accompanying the $C_T$ and $A_T$ data, we obtain a clear warming signal (not shown).

In deeper layers, there is an overall increase in $C_T$ and $A_T$ (except in the Iceland Sea), salinity, and temperature. Although the effect of increasing $C_T$bg is reduced away from the surface as a consequence of the gradual isolation of deeper waters from the atmosphere, it remains the main driver of pH change down to 2000 m. The significant trends of $C_T$bg at the 1000–2000 m depth level in the Greenland Sea could be a consequence of the deep winter mixing that has been shown to reach down to 1500 m in this region (Brakstad et al., 2019). In the other regions of the Nordic Seas, the winter mixed layers have not been documented to reach these depths (e.g., Ólafsson, 2003; Skjelvan et al., 2014; Våge et al., 2015). However, intermediate water masses from the Greenland Sea have been shown to spread horizontally in the Nordic Seas, which could also explain the significant trends in the Norwegian and Lofoten basins and in the Iceland Sea (Blindheim, 1990; Blindheim and Rey, 2004; Messias et al., 2008; Jeansson et al., 2017). The effect of the biogeochemical component of $A_T$ is negligible in deep waters, except for in the Barents Sea Opening, where the increase in $A_T$bg in the 200–500 m layer is as large as in the surface layer, and in the 1000–2000 m layer in the Norwegian Basin,

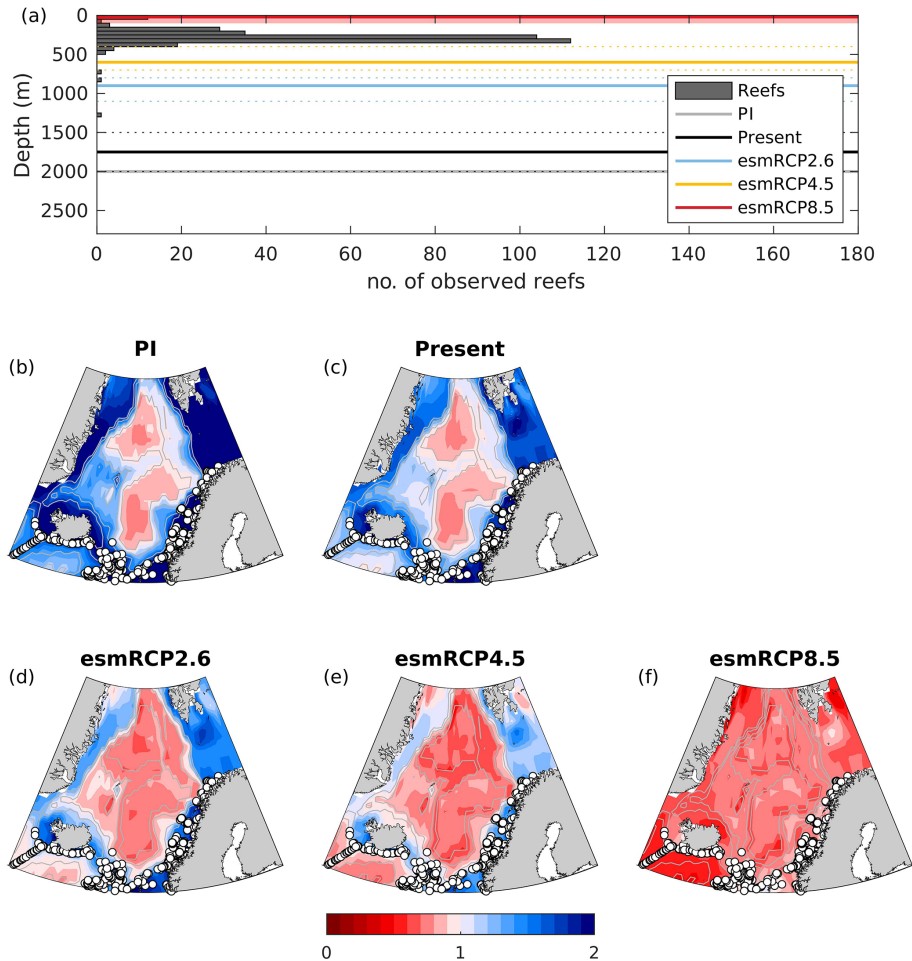

**Figure 12.** The number of observed reef sites per 50 m depth interval together with the aragonite saturation horizons (solid lines) in the Nordic Seas for the past (1850–1879), present day (1980–2005), and future (2070–2099) under the esmRCP2.6, esmRCP4.5, and esmRCP8.5 scenarios calculated from the GLODAPv2 climatology and NorESM1-ME simulations. The dashed lines show the uncertainty ($\sigma_{\text{field}}$). The red shading shows the projection uncertainty, as estimated from our ESM ensemble for esmRCP8.5 **(a)** and maps showing aragonite saturation state of bottom waters (calculated from the GLODAPv2 climatology and NorESM1-ME simulations), together with positions of observed reefs **(b–f)**.

where there is an increase in $A_{\text{T}}$bg that nearly cancels the effect of increasing $C_{\text{T}}$bg. The exceptionally strong trends in $A_{\text{T}}$bg in the upper and the 200–500 m layer in the Barents Sea Opening are intriguing. Considering that the strong $A_{\text{T}}$bg trend also exists in the 200–500 m layer, it is likely not a result of seasonal undersampling. One biogeochemical process that could have a potential impact on the Barents Sea $A_{\text{T}}$bg trend is the recurrent blooms of calcifying coccolithophorids (Giraudeau et al., 2016), which consume $A_{\text{T}}$ during growth and release $A_{\text{T}}$ when their shells are decomposed. There are indications of an increase in their presence in the Barents Sea (Giraudeau et al., 2016; Oziel et al., 2020). In which direction this would impact the $A_{\text{T}}$ depends on horizontal advection, remineralization, and burial and deserves separate dedicated process studies. The freshwater components of $C_{\text{T}}$ and $A_{\text{T}}$ are mainly detectable in the upper 500 m. As

for the surface, the thermodynamic effect of salinity changes on pH are negligible in the deep water. The warming seen in deep waters, which has a negative contribution on the pH trend, is an additional indication that the absence of a temperature trend in the upper layer is a result of seasonal undersampling. In deep waters, the warming signal does not only come from local vertical mixing. There is also an indication of decreased deep-water formation in the Greenland Sea, which has caused an increased exchange with warmer Arctic deep waters (e.g., Østerhus and Gammelsrød, 1999; Blindheim and Rey, 2004; Karstensen et al., 2005; Somavilla et al., 2013). Below 2000 m, there are barely any detectable changes in the various pH drivers. The water masses at these depths are increasingly dominated by old Arctic deep waters (e.g., Somavilla et al., 2013). With ages exceeding 200 years (Jutterström and Jeansson, 2008; Stöven et al., 2016), they

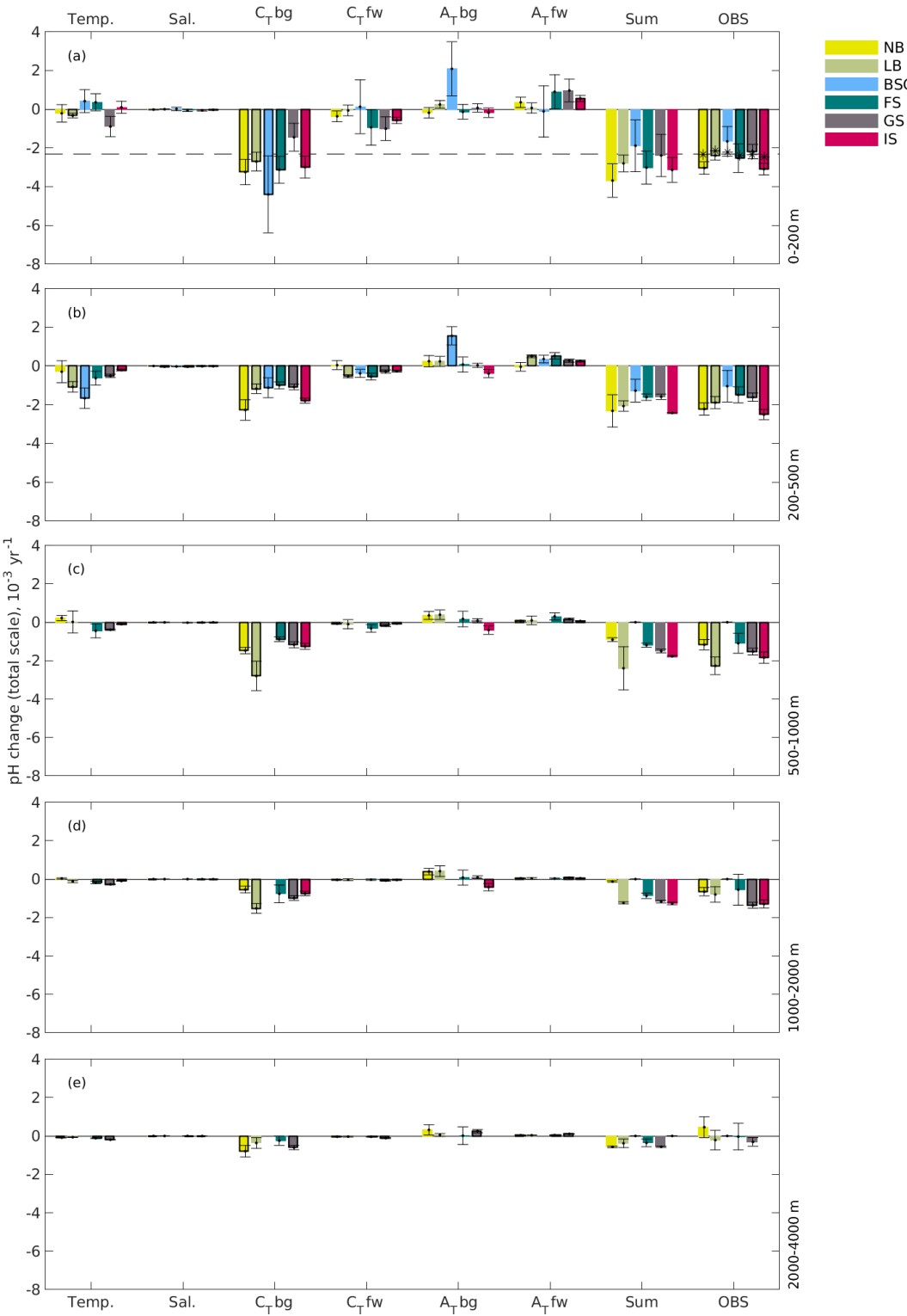

**Figure 13.** Contribution of observed changes in temperature, salinity, $C_T$, and $A_T$ to the observed trend in pH (OBS) over the 1981–2019 period in the Norwegian Basin (NB), Lofoten Basin (LB), Barents Sea Opening (BSO), Fram Strait (FS), Greenland Sea (GS), and Iceland Sea (IS; Fig. 1). The contribution of $C_T$, $A_T$ was divided into a freshwater (fw) component and a biogeochemical (bg) component. Bars showing trends that are significantly different from zero are outlined with a black line. The term "sum" indicates the total trend in pH calculated as the sum of the trends associated with these six driving factors. The dashed line and black asterisks indicate the pH trends expected from the change in atmospheric $CO_2$ during the same period for the whole area and for the separate basins, respectively.

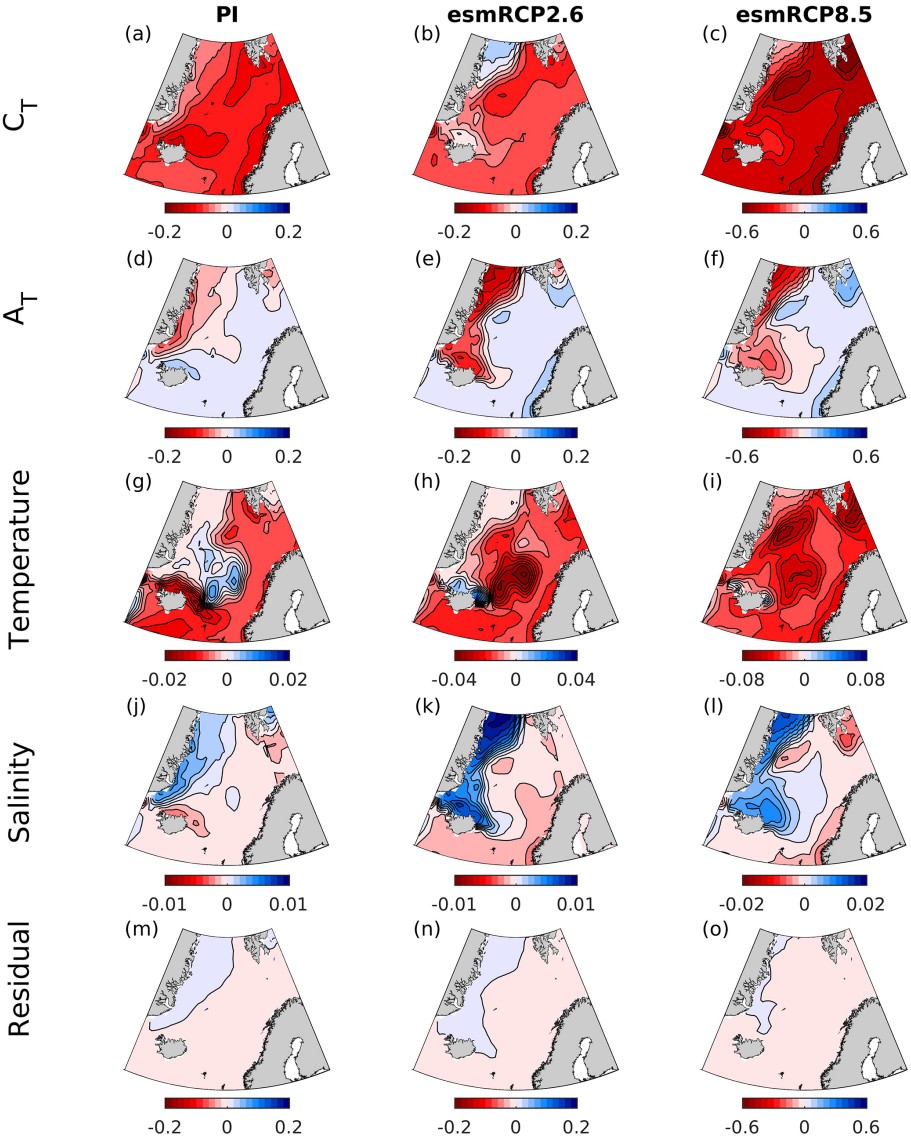

**Figure 14.** Contribution of modeled changes in surface $C_T$, $A_T$, temperature, and salinity to the change in pH between 1850–1859 and 1996–2005 (PI) and 1996–2005 and 2090–2099 (esmRCP2.6 and esmRCP8.5). The residual indicates the difference between the total change in pH, calculated as the sum of the trends associated with these four driving factors, and the actual change shown in Figs. 5, 9, and 11.

have been isolated from the increasing anthropogenic $CO_2$, which explains the weak trends at these depths.

### 5.7.2 Past and future drivers

For past and future changes, the drivers of surface pH change show similar spatial patterns over all time periods, except for temperature (Fig. 14). The main driver is an increase in $C_T$, which is larger in Atlantic water than in polar waters. This is explained by the dilution of $C_T$ in polar waters by the increased freshwater export from the Arctic Ocean (Fig. 15, Shu et al., 2018) that, to some degree, counteracts the effect of atmospheric $CO_2$ uptake. A similar freshwater effect has recently been observed also in the Arctic Ocean (Woosley

and Millero, 2020). The biogeochemical component of the $C_T$ driver (Fig. 15), which is primarily the effect of increasing anthropogenic carbon, is larger in polar waters for the changes from the present to future in both the esmRCP2.6 and esmRCP8.5 scenarios, which is what is expected from their lower buffer capacity (Sect. 2). The effect of $A_T$ is most prominent in polar waters, where a reduced $A_T$ concentration contributes to a pH decrease that is of the same order of magnitude as that driven by $C_T$ (Fig. 14). From the freshwater decomposition in Fig. 15, we see that the $A_T$ changes are mainly driven by freshwater fluxes, and that contributions from the biogeochemical component are negligible. $A_T$ dilution has also been shown to be important in the future in

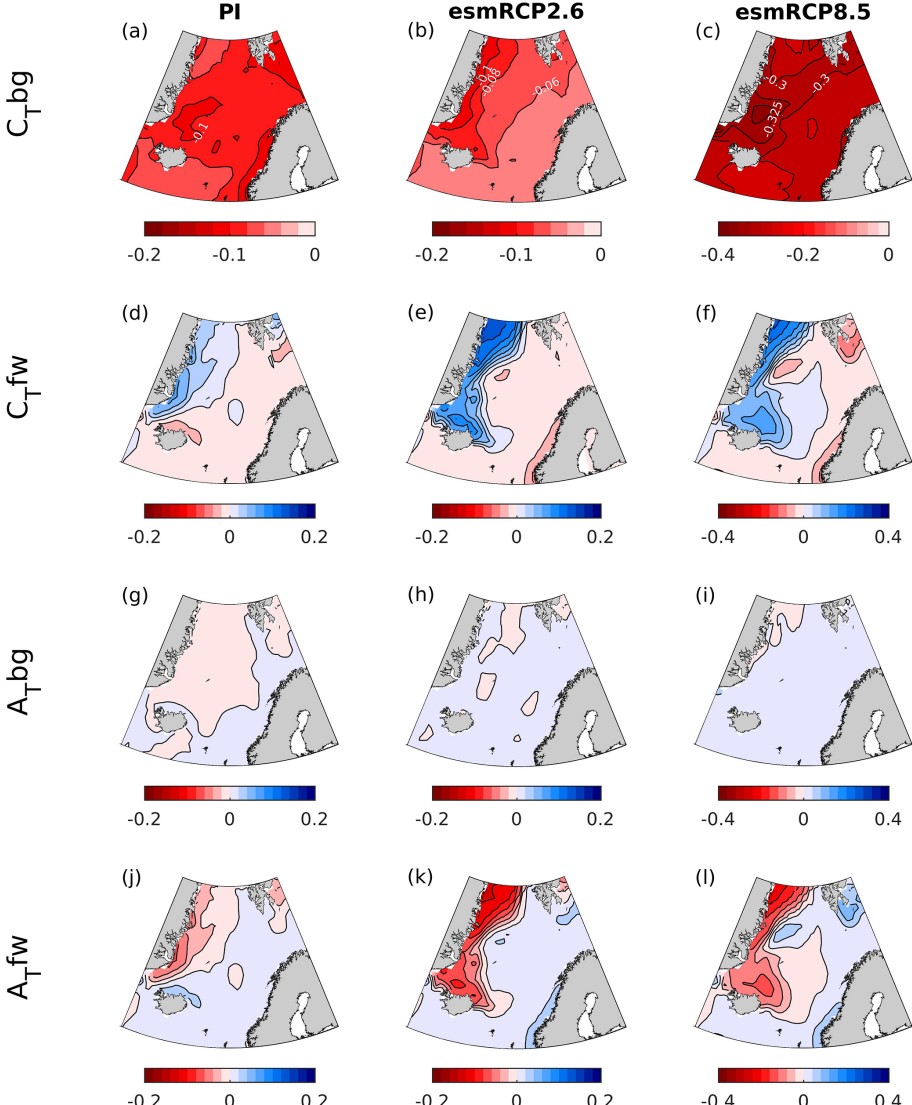

**Figure 15.** Contribution of the biogeochemical and freshwater components of $C_T$ and of $A_T$ ($A_T$bg TS8 and $A_T$fw) to the change in pH between 1850–1859 and 1996–2005 (PI) and 1996–2005 and 2090–2099 (esmRCP2.6 and esmRCP8.5).

the Arctic Ocean in several CMIP6 models (Terhaar et al., 2021). However, as discussed earlier, the net effect of these freshwater fluxes on pH are minor, as the dilution of $A_T$ and $C_T$ is similar, but have opposite effects on pH (compare Fig. 15d–f with Fig. 15j–l). The increasing freshwater export also results in a dilution of salinity in polar waters that has a positive contribution to the pH trend. The Atlantic waters show a tendency towards increasing salinity that partly amplifies the decrease in pH. Temperature has an overall negative effect on the pH trend as a result of an overall warming. From the preindustrial era to the present day and the present day to the future esmRCP2.6, the temperature increase is almost nonexistent in polar waters, indicating that it has been shielded from warming through the presence of sea ice. In some smaller regions, there is even a sign of a cooling, which

could be a result of an increased presence of polar waters due to the increasing freshwater export.

The combined effect of these drivers explain the zonal gradients in the pH decrease that are described in Sect. 5.3 and 5.5. From the past to the present day, the largest pH decrease takes place in the Atlantic water due to a stronger increase of anthropogenic carbon and a stronger warming in these waters. From the present day to the future, the acidification becomes larger in polar waters, compared to Atlantic water, due to the stronger increase in anthropogenic carbon in these waters. The increasing freshwater export from the Arctic that is seen in all time periods is of importance when regarding $C_T$ and $A_T$ concentrations separately, but their combined effect on pH is negligible. For the changes from the past to present-day and the present-day to future esmRCP2.6, the zonal gra-

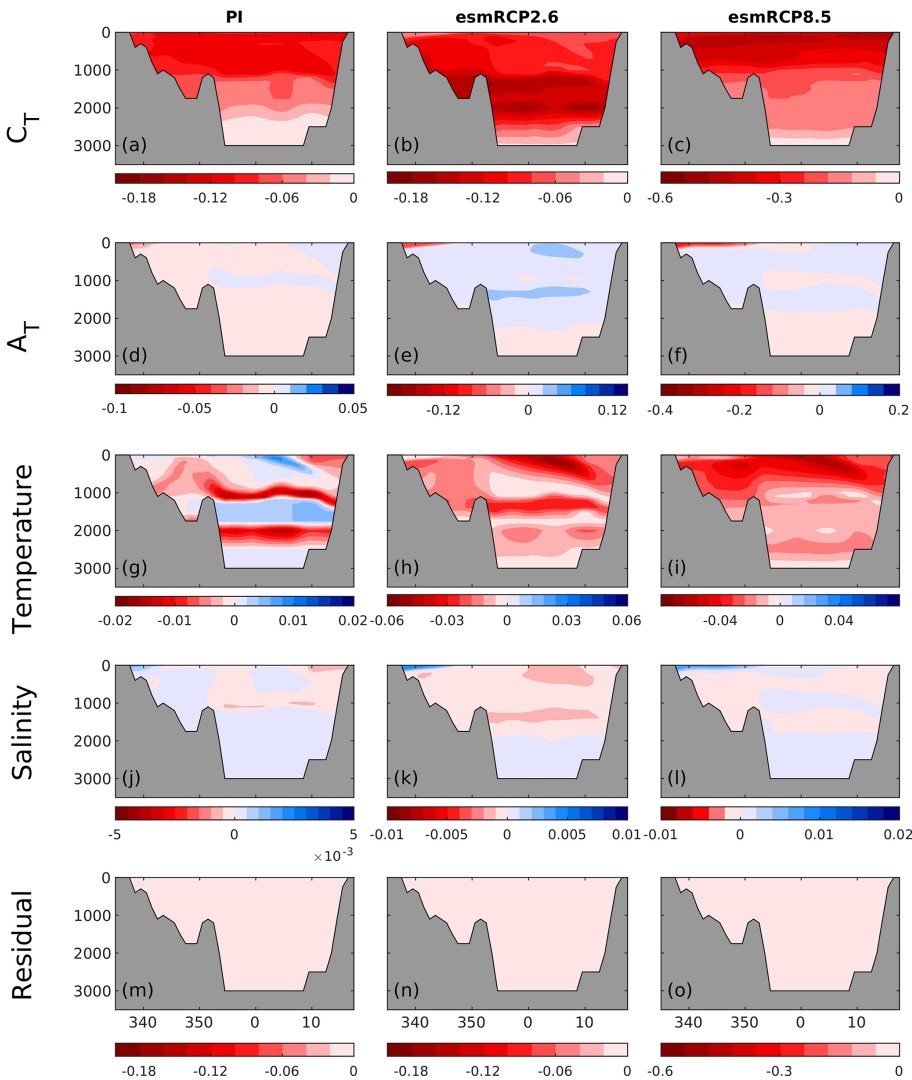

**Figure 16.** Contribution of modeled changes in surface temperature, salinity, $C_T$, and $A_T$ to the change in pH between 1850–1859 and 1996–2005 (PI) and 1996–2005 and 2090–2099 (esmRCP2.6 and esmRCP8.5) at the depth section at 70° N, as shown in Figs. 6 and 10. The residual indicates the difference between the total change in pH, calculated as the sum of the trends associated with these four driving factors, and the actual change shown in Figs. 6 and 10.

dient in the $\Omega_{Ar}$ trend follows that of pH, showing the importance of the $C_T$ driver. It is reinforced by the spatial variations in the warming, i.e., the stronger warming in the Atlantic water compared polar waters results in a relatively stronger drop in $\Omega_{Ar}$ in polar waters. In the esmRCP8.5 future, $\Omega_{Ar}$, in contrast to pH, exhibits a larger drop in the Atlantic water. This can be explained by the relatively small changes in temperature in this region compared to the rest of the Nordic Seas, which affects $\Omega_{Ar}$ in the opposite direction compared to pH.

Below the surface layer, $C_T$ is also the main driver of past and future pH changes (Fig. 16). The change from the preindustrial era to the present day indicates a gradually weaker impact of $C_T$ with depth, except for a tongue at about 1000 m depth that connects to the surface in the Iceland Sea. This is most likely related to the deep water for-

mation in this region that spreads at depth. The end-of-the-century $C_T$ increase for the esmRCP2.6 scenario is larger in the deep water than in the surface layer, resulting in the stronger pH reduction at mid-depths, as seen in Fig. 10. This mid-depth layer with a strong acidification is partly a result of the higher atmospheric $CO_2$ concentrations in the middle of the 21st century in combination with the rapid ventilation of the water column in this area, i.e., when these waters were at the surface, they were exposed to peak atmospheric $CO_2$. However, the large $C_T$ increase in deep waters is also partly explained by increased remineralization, as indicated by a $\sim 1\,\mathrm{mL\,O_2\,L^{-1}}$ TS9 increase in the apparent oxygen utilization (AOU) at depths of 1800–2100 m throughout the Nordic Seas in both esmRCP2.6 and esmRCP8.5 (not shown). Assuming a Redfield ratio of $O_2 : C = 132 : 106$, this

corresponds to a change in $C_T$ of $\sim 30\,\mu\mathrm{mol\,kg^{-1}}$, which results in a pH decrease of $\sim 0.1$ at the alkalinity in question. Impacts of changes in $A_T$, salinity, and temperature are relatively modest at depth.

The residual between the sum of the four drivers and the actual pH change is small (Figs. 14 and 16) and can be attributed to approximations involved in the decomposition, including the approximations of the partial derivatives, the assumption of a linear trend, and the use of temporal means (Takahashi et al., 1993; Lenton et al., 2012; Lauvset et al., 2015). Although the absolute numbers related to the drivers should be taken with care, this decomposition still gives a good estimate of the relative importance of the effects of temperature, salinity, $C_T$, and $A_T$ on pH changes.

In the historical run and all three future projections of NorESM1-ME, the change in surface ocean $p\mathrm{CO}_2$ differs from the change in the atmosphere (Fig. S16). From the preindustrial era to the present day, there is an increase in the undersaturation, i.e., the increase in the oceanic $p\mathrm{CO}_2$ lags behind the increase in the atmosphere. This means that the pH decrease is less than that expected from the increase in atmospheric $\mathrm{CO}_2$. The lag continues into all the future scenarios, but from around 2040 and onward, the oceanic $p\mathrm{CO}_2$ increases faster than that of the atmosphere, resulting in a decreasing undersaturation. In esmRCP2.6 and esmRCP4.5, this causes stronger decreases in pH (from 1996–2005 to 2090–2099) than expected from the rise in atmospheric $\mathrm{CO}_2$. In esmRCP8.5, however, the difference between the end-of-the-century ocean and atmospheric $p\mathrm{CO}_2$ is still larger than the present day, meaning that the decrease in pH is less than expected. As detailed above, there are several mechanisms underlying the undersaturation of surface ocean $p\mathrm{CO}_2$ in the Nordic Seas, but further analyses of these, including their potential future changes, is beyond the scope of this paper.

## 6   Summary and conclusions

We have provided a detailed analysis of the spatial and temporal variations in the past, present-day, and future acidification, and its drivers, in the Nordic Seas. We have further assessed the potential impacts of this acidification on aragonite saturation and cold-water coral reefs. This work builds on Skjelvan et al. (2014), who estimated pH trends, and their drivers, for various subregions of the Nordic Seas from observational data sampled between 1981 and 2013. Here, we have added data from the Iceland Sea and from later years to obtain the greatest possible temporal and spatial coverage. We have, additionally, made an analysis of past and future pH changes by the use of the gridded GLODAP climatology and ESM simulations to put the observed changes into the context of long-term climate change. In contrast to previous studies that have assessed the future pH changes in the Nordic Seas for single scenarios (Bellerby et al., 2005; Skogen et al., 2014, 2018), we here analyze the output from

one mitigation scenario, one stabilization scenario, and one high-emission scenario. To our knowledge, no previous studies have presented past pH changes in the Nordic Seas.

### 6.1   pH changes and its potential ecosystem impacts

From the preindustrial era (1850–1860) to the present day (1996–2005), a combination of NorESM1-ME with the GLODAPv2 preindustrial estimate suggests that the pH of Nordic Seas surface waters has dropped by 0.1. During this period, the aragonite saturation horizon has slightly shallowed but has remained well below the depths of known cold-water coral habitats. During 1981–2019, when regular sampling of carbon system variables were made in the region, the pH of the Nordic Seas upper layer has decreased at a rate of $-2.79 \pm 0.3 \times 10^{-3}\,\mathrm{yr^{-1}}$ on average, resulting in a pH decline of 0.11. The pH reductions are significant all over the Nordic Seas' upper layer (0–200 m), except in the Barents Sea Opening, where the lack of significant change is a result of a strong increase in $A_T$. In some regions, the acidification is detectable down to 2000 m, which we attribute to the deep water formation and spreading of these water masses at depth. The waters at 1000–2000 m throughout the Nordic Seas are now close to aragonite undersaturation. Our results are in overall agreement with Skjelvan et al. (2014), but the longer time series result in statistically significant ($p < 0.05$) trends in even more regions and depth layers. An additional pH drop of 0.1–0.4 in the surface waters is projected until the end of the 21st century, depending on the emission scenario. In the high-emission scenario, esmRCP8.5, all cold-water coral reefs will be exposed to corrosive waters by the end of the 21st century, threatening not only their existence but also that of their associated ecosystems. This is confirmed by an ensemble of six CMIP5 models that all agree on these consequences. The NorESM1-ME simulations suggest that some cold-water corals will be exposed to undersaturation also under the esmRCP4.5 scenario, and that this can only be avoided by keeping the emissions within the limits prescribed in the esmRCP2.6 scenario. Because NorESM1-ME tends to simulate a relatively strong decline in pH and shallow saturation horizons in comparison to our ESM ensemble for esmRCP8.5, our estimated aragonite saturation horizons for esmRCP2.6 and esmRCP4.5 should be considered as the shallow, lower bound of possible future states. Our estimates of the future pH and $\Omega_{\mathrm{Ar}}$ in the Nordic Seas add more possible future states to the ones presented for the A1B and RCP4.5 scenarios by Skogen et al. (2014, 2018).

### 6.2   pH drivers

The acidification during the last 39 years is, in all subregions, mainly driven by increasing $C_T$ in response to the rising anthropogenic carbon concentrations. This is in agreement with the results for the period of 1981–2013 from Skjelvan et al. (2014), who calculated the drivers of pH change for the Nor-

wegian Basin and the Greenland Sea. The effects of increasing $C_T$ are slightly opposed by increasing $A_T$. The increasing $A_T$ is partly a result of a salinification of the Nordic Seas. However, this salinification also results in a increase in $C_T$, which counteracts the effect of the freshwater-driven increase in $A_T$. The net effect of $C_T$ and $A_T$ on pH is, therefore, a result of biogeochemical processes. We find a clear warming signal in deep waters, which has contributed to the decreasing pH. In the upper 200 m, however, there is no clear temperature change. We find this to be a result of seasonal undersampling, which further complicates a comparison of the changes in sea surface $p\text{CO}_2$ to the atmospheric one. In the Barents Sea Opening, there is an exceptionally strong increase in $A_T$, which we cannot relate to increasing salinity. The reasons behind this strong increase are then either a result of biogeochemical processes or could also be a result of sampling issues. Unfortunately, we cannot pin this down with the dataset we have, and this remains as an open question for future investigations.

For past and future changes, we also find increasing $C_T$ to be the main driver of pH change in the Nordic Seas. This is in agreement with Skogen et al. (2014), but we distinguish some regional differences related to different water masses. Increasing temperatures, which amplify the effect of increasing $C_T$, are more prominent in Atlantic water in changes from the preindustrial era to the present-day, and CE3 the present-day to the future esmRCP2.6. The absence of a warming signal in polar waters is probably a result of the shielding effect of sea ice. In esmRCP8.5, however, the warming is more uniform over the Nordic Seas, which most likely is a result of the significantly reduced sea ice cover. In both past and future scenarios, there is a clear signal of an increasing freshwater export from the Arctic Ocean that dilutes $C_T$, $A_T$, and salinity in polar waters, and there is a tendency towards increasing salinity in the Atlantic water that also leads to increasing $C_T$ and $A_T$. The total effect of this change in the freshwater content on pH is negligible, as the effect of changing $C_T$ and $A_T$ oppose each other, and because the thermal effect of salinity is minor in comparison to the other drivers.

*Data availability.* The GLODAPv2.2019 data and GLODAPv2 mapped climatologies are available for download at https://www.glodap.info/index.php/merged-and-adjusted-data-product-v2-2019/ and https://www.glodap.info/index.php/mapped-data-product/ (Olsen et al., 2019; Lauvset et al., 2016), respectively.

The data from Ocean Weather Station M from 2001–2007 are available in GLODAPv2.2019 (Olsen et al., 2019). Data from the time period 2008–2019 are available at the Norwegian Marine Data Centre (NMDC) via https://doi.org/10.21335/NMDC-872095870 (Skjelvan, 2021).

The data from the time series station in the Iceland Sea can be obtained from the NCEI database (Ólafsson, 2012; Ólafsdóttir et al., 2020) at https://doi.org/10.25921/qhed-3h84 and https://doi.org/10.3334/cdiac.otg.carina_icelandsea.

The data from the Norwegian ocean acidification monitoring program (2011-2012 Tilførselsprogrammet and 2013–2019 Havforsuringsprogrammet) (Chierici et al., 2019a, b) and from the eastern Fram Strait (Chierici and Fransson, 2019) are available at the Norwegian Marine Data Centre (NMDC) via https://doi.org/10.21335/NMDC-1738969988, https://doi.org/10.21335/NMDC-1939716216, and https://doi.org/10.21335/NMDC-154415697.

The ESM simulations (Arora et al., 2011; Yukimoto et al., 2011; Dufresne et al., 2013; Dunne et al., 2013a, b; Giorgetta et al., 2013; Long et al., 2013; Tjiputra et al., 2013, 2016) can be downloaded at https://esgf-node.llnl.gov/search/cmip5/ (Lawrence Livermore National Laboratory, 2021) TS10.

The cold-water coral positions have been derived from data that are made available under the European Marine Observation and Data Network (EMODnet) Seabed Habitats initiative (https://www.emodnet-seabedhabitats.eu, EMODnet, 2020), financed by the European Union under Regulation (EU) no. 508/2014 of the European Parliament and of the Council of 15 May 2014 of the European Maritime and Fisheries Fund. The data owner and EMODnet Seabed Habitats consortium accept no liability for the use of these data or for any further analysis or interpretation of the data.

*Supplement.* The supplement related to this article is available online at: https://doi.org/10.5194/bg-19-1-2022-supplement.

*Author contributions.* AO, FiF, and FrF designed the research. FiF, FrF, and AO performed the data analysis, with input from NG, IS, MC, and EJ. FiF led the writing of the paper, with input from all co-authors. JT designed, tested, and performed the NorESM1-ME model simulations.

*Competing interests.* The contact author has declared that neither they nor their co-authors have any competing interests.

*Acknowledgements.* Filippa Fransner has been funded by the Bjerknes Centre for Climate Research, by the Research Council of Norway through The Nansen Legacy project (RCN; grant no. 276730) and by Det Kongelige Norske Videnskabers Selskap. Friederike Fröb has been funded by the Bjerknes Centre for Climate Research. Jerry Tjiputra acknowledges funding from the Research Council of Norway (Reef-Futures, grant no. 295340; Columbia, grant no. 275268; CE2COAST, grant no. 318477). Nadine Goris acknowledges funding from the Research Council of Norway (IM-POSE; grant no. 294930). Siv K. Lauvset acknowledges funding from the Research Council of Norway (NorArgo2; grant no. 269753). This work is a contribution to the project INTAROS. This project has received funding from the European Union's Horizon 2020 Research and Innovation programme (grant no. 727890). High-performance computing and storage resources were provided by the Norwegian infrastructure for computational science (project

nos. nn1002k and ns1002k). The Norwegian program "Monitoring ocean acidification in Norwegian waters" is supported by the Norwegian Environmental Agency (grant no. 17078007). Data from the Barents Sea are supported by Institute of Marine Research and the flagship program "Ocean acidification and ecosystem effects in northern waters" at FRAM – High North Research Centre for Climate and Environment. We acknowledge the World Climate Research Programme's Working Group on Coupled Modeling, which is responsible for CMIP, and we thank the climate modeling groups, for producing and making available their model output. For CMIP, the U.S. Department of Energy's Program for Climate Model Diagnosis and Intercomparison provided coordinating support and led the development of software infrastructure in partnership with the Global Organization for Earth System Science Portals. We thank the four reviewers, for their constructive feedback that greatly improved the paper.

*Financial support.* This research has been supported by the Bjerknes Centre for Climate Research, the Research Council of Norway (The Nansen Legacy, grant no. 276730; Reef-Futures, grant no. 295340; Columbia, grant no. 275268; CE2COAST, grant no. 318477; IMPOSE, grant no. 294930; NorArgo2; grant no. 269753), the European Union's Horizon 2020 Research and Innovation programme (grant no. 727890), and Det Kongelige Norske Videnskabers Selskap.

*Review statement.* This paper was edited by Jean-Pierre Gattuso and reviewed by James Orr and three anonymous referees.

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

**Remarks from the language copy-editor**

**Remarks from the typesetter**