# Peer review of "Acidification of the Nordic Seas"

_Biogeosciences, 2020_

## Referee Comment (RC1) · Anonymous Referee #1 · 15 Oct 2020

The authors have put together an overview of past, present, and future ocean acidification in the Nordic Seas using measurements, gridded assembled data, and model projections. The work thus presents an exhaustive overview of ocean acidification in the Nordic Seas that resembles most available information and will as such certainly prove to be very useful for other scientists and to people outside of Science, such as policy makers. Personally, I very much enjoyed reading the manuscript and want to congratulate the authors.

The manuscript would, however, significantly improve if underlying processes would be discussed further to explain the drivers between the different reaction of the regional seas to ocean acidification. What drives the changes in alkalinity, dissolved inorganic carbon, temperature and salinity? Is it changes in water masses, circulation, deep water formation, freshening from the Arctic Ocean or land ice melt from Greenland?  A better explanation would help to understand past and future changes and also help to understand how well the one model used in this study performs regarding the expected changes. Furthermore, I have a number of questions regarding the Methods that would need to be addressed before publication.

**Major comments:**

General: Please be quantitative as often as possible and avoid formulations like "it agrees well. This should be carefully addressed throughout the manuscript.

Introduction: I suggest merging sections 1,2, and 3. Some parts are repeated, and it would be helpful to have some information earlier. For example, I think a better explanation of how ocean acidification works chemically is needed and what the saturation state is and why it matters (or why not?). Why do the Nordic Seas have naturally low saturation states, etc. It is absolutely right, that the Nordic Seas are of high importance and very interesting, so strive on it. Why does it have short residence times and what is the link between the residence time and the $p$CO$_2$ undersaturation? Which water masses meet? It all makes more sense after reading the sections 2 and 3, but section 1 is not understandable for oceanographers who do not know the Nordic Seas. Overall, I think most of the information is in the manuscript, but restructuring could strongly improve the Introduction. And as you discuss future projections, you could also highlight the difficulties that models have in these regions.

Undersaturation of $p$CO2: Please introduce the concept of undersaturation carefully. Presented like this in the abstract without explanation leads to large confusion. Most of the time, I think I understand what you mean with the concept of undersaturation of $p$CO2. At other times, I am very confused when you compare the $p$CO$_2$ in deep waters (2000 m) to $p$CO$_2$ in the atmosphere at the same time. Given that the age of the water masses at 2000 m is likely much older, it has "seen" the atmosphere a long time ago. How does it make sense to speak of undersaturation or a perfectly tracked pH?

I think a lot of people are not familiar with this concept, including. How do you calculate/define undersaturation? Why does the undersaturation exists in pre-industrial conditions? I suppose that northward flowing waters cool down and their $p$CO$_2$ decreases

faster as their solubility increases than they can take up $CO_2$ from the atmosphere (Broecker et al., 1974). If that is the dominating process, the weakening of the undersaturation is somehow expected, as cold waters hold less $C_{ant}$. With increasing atmospheric $CO_2$, the differences in surface ocean $C_T$ should become smaller, which would also lead to a smaller difference in marine and atmospheric $pCO_2$ in the northward flowing waters in the Nordic Seas. It should be very similar to the processes in the Barents Sea, where Terhaar et al. (2020) showed that $C_{ant}$ is larger in the northward flowing waters than expected from atmospheric $CO_2$ concentrations. This larger $C_{ant}$ is equivalent to a weakening of the undersaturation of $C_T$. With a changing climate, other drivers could potentially also become important, such as a change in circulation, i.e. is the quantity of northward flowing waters larger or smaller, are they flowing faster or slower, do they extend further into the Nordic Seas? all these factors could come into play regarding the undersaturation and need to be understood to properly discuss this effect.

Overall, I think this concept needs to be introduced in more detailed with the respective literature if it is a core concept of the paper. In fact, much is explained in the Discussion (Lines 314-328). This should be moved to the Introduction. In the Discussion, it would be really great if you could pin down the most important mechanisms for each regional sea. As it is now, the manuscript is more an atlas to look up numbers, explaining the mechanisms would put it on a whole different level. I strongly encourage you to do this.

Line 9: You point out that the Barents Sea Opening is different to all other regions because of an increase in $A_T$. This sounds very interesting and it is thus even more disappointing that no explanation exists why the $A_T$ increases? Is it via biological processes such as an increased production of organic matter via primary production or is it a change in circulation, i.e. are Atlantic water masses with higher $A_T$ expanding further and displacing Arctic Ocean waters? I really would like to have an explanation here.

Line 16-17/ lines 119-122: No need to request further model studies. The output from CMIP5 and CMIP6 models are online available. While differences in pH and saturation among models are generally small between models, they can be large below the surface in the North Atlantic (Goris et al., 2018) and in the Arctic Ocean (Terhaar et al., 2020b). I suspect thus that the differences in the Nordic Seas would be large, too. Moreover, the NorESM-ME model seems to be at the high end of carbon uptake in both, the North Atlantic and the Arctic Ocean and thus likely at the high end in the Nordic Seas as well. This is likely caused by an overall strong AMOC in the model (Wang et al., 2014).

Given that the NorESM-ME model is at the high-end for the North Atlantic anthropogenic carbon sink, I doubt that it is representative for future ocean acidification. The cited papers only treat the surface (mainly driven by atmospheric $pCO_2$) and the bottom ocean where changes in all variables are small and models "automatically" agree. I am not aware that these papers analyze model differences in waters from 100 to 2000 m.

Given the detailed analysis of the historical data, I strongly believe that it is necessary to analyze a larger number of models with annual mean data. I feel that it is absolutely necessary in order to make strong statements about the future acidification in the Nordic Seas, such as that undersaturation can only be avoided for RCP2.6 and RCP4.5.

Lines 95-97: Annual means/seasonal bias: Although not enough data exist to de-seasonalize the data, the model output gives you several possibilities to assess the possible uncertainties.

You could subsample the model at the same time and place and compare these to the model annual mean values. Or even easier, you could compare the seasons within the model output to understand the seasonal variations. And are the under sampled months crucial in these regions, i.e. is winter under sampled and is deep-water formation occurring in winter? If that is the case, you might even expect a higher $pCO_2$ in winter. Given the strong seasonality, I feel that an analysis of the potential bias is necessary.

pH uncertainties: I have trouble in seeing the standard deviation over the entire Nordic Seas as an uncertainty. This is rather a measure of heterogeneity. The large gradients in T, S, $C_T$, and $A_T$ over the Nordic Seas thus increase artificially the uncertainties in pH when compared to observations. I have no good proposition but I would consider using uncertainties from the dissolution constants using mocsy2.0 (Orr et al., 2018) instead of the heterogeneity. Whatever you do in the end, it would be great if you could find a solution that permits to compare the model to the observations on the same basis.

Trends and significance: In many figures you show trends for each subsea and depth. I am still wondering how you calculate if a trend is significant (maybe I missed the explanation). Especially confusing are for example Figure 5f,g. Measurements are only available for the 2nd half of the period and uncertainties become enormous for 1981 as no measurement exists there. How is it possible to speak of a significant trend given the large gap in observations? I am not strong in statistics and would need a better explanation here.

**Minor notes:**

General: I would suggest using $C_T$ and $A_T$ instead of DIC and ALK as recommended by the guide for best practice (Dickson et al., 2007). Along the same lines, please write $p$CO$_2$ instead of pCO$_2$.

Line 2: Why do the high-latitude location lead to a higher sensitivity to acidification? People who are not ocean biogeochemists might not directly understand that high-latitude oceans have a higher solubility, hold more dissolved inorganic carbon ($C_T$) and have thus a naturally lower pH and lower saturation states. I feel that it would be helpful if you lead the reader a bit more here.
Moreover, aren't projected changes in the saturation states smaller in the high latitudes than in the tropics because the absolute numbers are already lower. It would probably be similar for pH? So strictly speaking, the Nordic Seas are more vulnerable towards ocean acidification because they start with more $C_T$ and lower saturation states and pH, but they are not more sensitive.

Line 4: Could you be more exact instead of writing pre-industrial? The definition of preindustrial is crucial for the historical anthropogenic carbon uptake (Bronselaer et al., 2017) and in consequence also for historical acidification.

Line 6: Please be quantitative. How much larger are they?

Line 10: What exactly is the "acidification signal"? Which regions are you speaking, please be more specific? And 1000-2000 m is not very precise for the saturation horizon at present-day conditions. It would also be helpful to know where it was in pre-industrial conditions? Did it increase by 50 m or by 1000 m?

Line 11: What do you mean by significant?

Line 12: The switch between two sentences from present day saturation states to future pH makes it hard to read and to compare.

Line 13: Again, a switch from surface pH to the saturation state over the entire water column. You are losing me here.

Line 13: I suggest to write "high-emission" instead of "worst case".

Line 15: At what depths exactly? Everywhere or online over a depth of 10 m?

Line 15: What is the majority? Literally all or 51%?

Line 20: Are the 25% exact? Or around 25%?

Line 22: "More serious downside" is not necessary. Just delete that part of the sentence to make it shorter.

Line 23-24: What threat are they imposing? Can you be more precise please?

Line 27-29: I think this sentence is wrong. I am not aware of well constrained ocean acidification at global scale. This might hold for the surface ocean but definitely not for subsurface to deep ocean basins. Differences in models lead to large differences (see bottom ocean pH in Kwiatkowski et al. (2020) in regions where waters reach the bottom. As soon as the waters leave the surface, I believe that historic acidification and its projections are rather ill constrained.

Line 34: Here, you could introduce your acidification signals.

Line 37-38: I like this last sentence, maybe you could make it a topic sentence of the paragraph. Please also write that and why the Nordic Seas have low saturation states.
Line 48: I am confused by the word "anthropogenic" here. Above, you used it in combination with anthropogenic carbon. I suppose now it is not to be read in combinations with carbon because changes in T, S, and $A_T$ are also anthropogenic, right?

Table 1: I am not sure if this table is needed. If you keep it, pleas define "minor".

Line 60: I think you can delete large parts of this sentence, such as "by the strongest base in seawater" and all after "which has been supplied …". It does not seem to be such an important information in this context.

Figure 1: Please add a colorbar. Could you change the color for the continents, it is hard to distinguish continents and shelf seas? It is also very difficult to see the dashed lines, could you somehow make them stand out better? A lot of points are outside your defined region, did you use these points? If not, I would suggest deleting them to make the figure less busy. Could you also highlight the two timer series stations that you mention in 4.1?

Section 4.1: The section is highly important, but it is difficult to read and to extract the information. For example, the topic sentence speaks about DIC, ALK, temperature and salinity. Later on, uncertainties are also reported for oxygen and nutrients. Did you also use oxygen and nutrients or only the 4 variables in the topic sentence? If you did not use nutrients and oxygen, no need to mention them.
Could you also please provide an assessment of how much volume of the Nordic Seas is covered by the GLODAPv2 data? Is it mainly surface data, how deep does it go, are some regions under sampled? Given, that you look at changes over the data collection time period, it would also be useful to see the measurement on a timeline to see if the measurements are biased towards the end of the period.
Please use paragraphs to divide between GLODAPv2, the two time series stations and the framework of the Norwegian ocean acidification monitoring program. This will likely help to read the section. By the way, what exactly it the Norwegian ocean acidification monitoring program? Could you introduce it with one sentence?

Line 86: Please be more precise than "approximately 4 times". Are they covering all seasons? Can be you be more precise than 10-20 depth levels? Are these depth levels concentrated at the surface or evenly spaced?

Line 109: Is there a reason why you did not include RCP6.0? Could you give the citations for the scenarios (Meinshausen et al., 2011; van Vuuren et al., 2011)?

Lines 110: What is similar? Please be quantitative. And please make the comparison for the Nordic Seas. If the NorESM2 model is doing a good job in the Pacific and Southern Ocean, what does it tell me with respect to the Nordic Seas?

Line 112: What is comparable? Please be quantitative here. The reader can then judge if that is good enough or not.

Line 113: This is definitely a red flag to me. What is broadly? How much? You do not even show me what happens under the surface. What simulated acidification? Over which period? The relatively close agreement at the surface is somehow expected as pH follows mainly the atmospheric $pCO_2$. Below the surface, I do not believe this statement without seeing it. Please revise the model evaluation and really try to convince me that the model works. As it is written now, I am absolutely not convinced.

Lines 113-115: Do you add relative changes in pH and saturation states to the absolute saturation states? I think you should not do this with non-linear variables (Fassbender et al., 2020). Please add simulated changes in $C_T$, $A_T$, T, and S to the respective variables in GLODAPv2 and use the projected $C_T$, $A_T$, T, and S to calculate projected pH and saturation states. And please precise how saturation states are calculated within the model world, using CO2SYS or mocsy2.0? Are you using simulated nutrients as you do for the observations?

Lines 124/125: Could you assess how good the pre-industrial $C_T$ values from GLODAPv2 are? If I understand right, a very basic version of the TTD method was used to calculate these numbers, although the TTD method is uncertain in the Nordic Seas and the Arctic Ocean? The time-dependece of saturation of CFCs and SF6 (Tanhua et al., 2008) as well as the change in saturation of surface ocean $C_T$ that you document here and that were documented in the Arctic Ocean (Terhaar et al., 2020). It would be good the let the reader know here that this pre- industrial $C_T$ has these uncertainties.

Line 130: I am a little bit confused here. Before you talked not really about nutrients when you discussed measurements and now, they are here. Did you have nutrient measurements everywhere? What did you do when nutrients measurmentes where not available?

Lines 134/135: Why do you need this sentence? It is confusing after the sentences above and it does not really add something. I suggest deleting it.

Lines 130-135: I suggest using the mocsy uncertainty propagation for the observations (Orr et al., 2018). That would help to know how reliable trends are.

Lines 139-142: This reads nicely but fits better into the Introduction. Here in the Methods, it would be helpful to motivate why you chose exactly these regions and not others. What makes them special? So far, I am still more or less in the dark about the special aspects of each regions, although this is a good start.

Line 144-145: If you reduce the Fram Strait, can you still call it the Fram Strait? To me, the Fram Strait is characterized by northward flowing Atlantic water and southward flowing Arctic water. If you chose the region in order to focus on the Atlantic waters and exclude almost 2 thirds of the width (looks like in Fig 1), you can hardly call it Fram Strait, can you? Maybe call it Eastern Fram Strait? Similar for the other regions, why do you choose such small subsamples. The argument of aliasing effects might be a good one, but at the moment it is not convincing to me. I think you have to elaborate a little bit on it.

Lines 146-147: Does 200 m correspond the maximum mixed layer? Or does a strong gradient exist between 0-100 m and 100-200 m?

Line 149: How did you define significant trends? Did you look at $r^2$ or p values? An explanation is necessary.

Line 150: For pH trends and uncertainties, wouldn't it be better to look at [H+]? Otherwise it is not a linear trend and the uncertainties are not linear either? (Fassbender et al., 2020).

Line 151: Didn't you introduce T and S already earlier?

Line 171: To be clear, you used only GLODAPv2 pre-industrial for past changes (section 4.3), but you used model output for the driver of these past changes? I think that I am not understanding something here.

Line 172: You do not consider changes in nutrients as potential driver, so I take it that they are negligible? Or are they somehow included in the alkalinity?

Line 173-175: As explained above, this needs much more introduction. At this point of the manuscript, I still have no idea why this should matter.

Line 176/177: If possible, use a topic sentence here to tell the reader what this first paragraph is about. And please be quantitative here ("Agrees well" does not tell me anything). Do you mean that it agrees within the uncertainties?

Line 179: What does 'it' refer to at the beginning of the sentence?

Line 179: Can you compare the two uncertainties? I suppose that you used the entire region simulated in the mode whereas the GLODAPv2 does not cover the entire region (or did you use the gridded version here from Lauvset et al.? In any way, please be more precise. In case that the model and the data product do not cover the same regions, I think you cannot compare the uncertainties as they are a measure of the homogeneity of the sample and not the real uncertainty. As the observations are likely concentrated in some regions, this might reduce the homogeneity and thus the uncertainty. Moreover, how confident are you with respect to the pre-industrial uncertainties given that you did not account for changes in T, S, and $A_T$?

Line 180: I do not buy this sentence. The different pH largely stems from the temperature, doesn't it? Globally, waters are warmer than in the Nordic Seas and thus the global surface average pH is smaller, right?

Line 181: Do you show this somewhere? You could simply illustrate that you are right, if you calculate the average of surface pH values in the observations. Otherwise this is just a speculative guess.

Line 182: Can you be more precise than "about 0.05"? Uncertainties would be good and regional differences as well. I also would make a single paragraph about the comparison to the global ocean to make it easier to read. As it is presented now, the comparison to the global ocean makes it hard to grab the information about the Nordic Seas.

Line 183: See comment above. As observations and model are not sampled at the same places, I think you compare measurement uncertainties with the spatial heterogeneity (which is large in the Nordic Seas). I do not think that this valid here. Moreover, the trend in the observations seems to be much larger than the simulated trend. It would be helpful to sub-sample the model at the same places as the observations to make a proper comparison.

Line 184-187: The trends are significantly different. Slightly stronger is misleading when the difference is 29% and more than two times the standard deviation. Could you discuss this please and try to explain why this is the case? You mention the variability at the beginning of the time period, but I only see one measurement at the beginning of the time period (first 5 years). And if I only look at the years from 1994-2020 (excluding the beginning, the trend seems to be even stronger). I think you cannot wave this away by mentioning variability.

Line 188/189: Please be quantitative: How much more than 0.4? How much below 7.7? What are the uncertainties? You have the numbers, so just give them here.
Please be consistent with the significant digits in earlier reported values, either always give two numbers behind the comma or one.

Line 189/190: As written above, the shrinking difference between global pH and high-latitude ocean pH is expected as cold waters can hold less $C_{ant}$. Please tell the reader here how much closer they get. It would be better to make a paragraph for the global-Nordic Sea comparison.

Lines 190/191: This hypothesis is very speculative and comes out of nowhere. This is your results section. If you mention it here, you have to show results that support your claim. Can you somehow use the model output to show what drives the shrinking difference between global and Nordic Sea surface ocean pH?

Lines 193-195: Again, please be exact and give uncertainties. Please also restructure the paragraph. First you mention RCP8.5, then you switch to the concept of reduced undersaturation with possible explanations and then you go back to the projections. It would be more readable if you first present the results for the scenarios and then present other mechanisms. Furthermore, the information that the pH decrease under RCP4.5 is 0.2 and half of RCP8.5 is redundant. Everyone is cabable of comparing 0.2 and 0.4.

Line 195: Can you give a reference that supports that the RCP2.6 is necessary to limit warming to 2°C. I think, depended on the model other scenarios might also be good enough?

Line 196: What is "slightly" above 8? As mentioned above, please use the same number of significant digits throughout the manuscript. Please also give uncertainties for the end-of century pH under RCP2.6

Figure 2: I do not know if the global line is needed, it makes it more complicated. Please make the shading transparent so that the other scenarios can also be seen in the first half of the 21$^{st}$ century. In the label you can delete the phrase "including those outside our regional boxes", it is redundant as you say before that you use all observations in the Nordic Seas.
I suppose your observations for every year come from a limited number of cruises that are in a well-defined part of the Nordic Seas and does not represent the entire Nordic Seas. Does it this make sense to make a global 'Nordic Sea' trend? If a cruise takes place north of Iceland or in the Fram Strait in that year, the pH should be relatively large, but if it takes place close to Norway, it would probably be small as suggested by Figure 3b. The fact that the mapped product (Lauvset et al., 2016) is in good agreement with the model suggest that the stronger trend from 1981-2019 in the observations might just be due to the sampling locations.
Could you also try to explain why the model uncertainty in 1850 is almost twice as large as the one from the gridded GLODAPv2 product? Is the model heterogeneity too large, or does the gridded product overlook the real heterogeneity because of the extrapolation of a limited number of measurements?
Overall, I would propose a very different model-data comparison as mentioned above. I suggest sampling the model at the same time and place as the obs. I would then make a scatter plot with the observations of the x axis and the model data on the y axis. You could then plot one point per year and the points should be on one line. If the points are constantly above or below, a constant bias could be seen. Maybe color the dots depending on the year to get the time dimension in the plot.

Section 5.1: I like that you discuss this with the water masses. I would make this even more prominent. Could you for example show the different water masses approximately with lines in Figure 3? How are these water masses defined? This would certainly help to understand changes in each water mass or changes of the extend of water masses. Instead of writing warm waters from the South, I suggest for example writing warm Atlantic waters.
Moreover, this section has a lot of qualitative assessments that are not supported by evidence (see comments below). Please go carefully through it and see how you can support all the claims with evidence.

Lines 205/206: You write that $C_T$ relates to salinity. I think you want to speak here about freshening from Greenland ice melt. If that is what you are writing about, please make it clear and do not leave the author left in the unknown about the relationship between $C_T$ and salinity.

Line 206/207: I would present it here as southward flowing Arctic waters along the Greenland coast with high pH and northward flowing Atlantic waters with high pH or something similar. And how do you define quantitively that temperature is the main driver? The alkalinity

distribution looks also similar to the pH distribution. If you make a statement about the main driver, can you give evidence?

Lines 206/207: I do not think that $A_T$ and $C_T$ effects dominate along the Greenland coast. It seems to be rather salinity/freshening. I also suggest deleting that a southeast northwest gradient exist. As you mention just afterwards, that gradient is not stable over the entire Nordic Seas.

Line 210: This information is needed in the Methods (see comments above). How well do the observations cover the Nordic Seas in space and time? Without this information, it is almost impossible to put the results in context.

Line 211: Here you speak about contrast of polar waters and Atlantic waters. That is great! I would present the results all around this idea. Polar waters, Atlantic waters, Greenland coastal waters. I think that would make it much easier to understand all these mechanisms and great findings that you present. However, at the moment I do not really now what I would expect as the paragraph above does not really use the words polar waters and Atlantic waters.

Line 214: Are the two citations the best ones here? Furthermore, the relationship between carbonate ion and temperature holds globally, but in regions with strong salinity and $A_T$ gradients, I would not think that it holds well enough. Do you have evidence that the relationship holds in these conditions? I really miss the effect of freshening, which can be large in high-latitude oceans.

Lines 216-218: This is speculative and no result. I would suggest deleting it.

Line 222: Please be quantitative, what do you mean by "rather uniform"? Isn't there a gradient from west to east? Especially along the Norwegian coast, the changes seem to be larger. Can you explain this?

Line 223-224/227-2289: Isn't the change in the saturation states just larger in the Atlantic waters because the saturation states are larger and therefore the same changes in the drivers ($C_T$, $A_T$, T, and S) lead to different changes in the saturation state due to the non-linear scale of the saturation state? Wouldn't it be better to look at [H+]?

Line 225: The $p$CO$_2$ undersaturation comes out of the blue here. Do you have a figure to show this? Do you show somewhere the larger CO$_2$ uptake in Atlantic waters? In any way, wouldn't it be expected that Atlantic waters take up more $C_T$ given the higher Revelle factor in Arctic waters compared to Atlantic waters? And what is the role of sea ice changes over the historical period? I find this a little bit fast given the complexity of the system.

Line 229: I do not agree that the impact of acidification changes with depth. It is the acidification that changes with depth or the impact of changes in $C_T$ on acidification. This is a difference. Furthermore, I would not say "limited connection" but rather refer to the time it takes to ventilate the deeper ocean and the age of the deep waters.

Line 232: Is this change significant when accounting for uncertainties? See Terhaar et al. (2020) as an example of uncertainties related to the ASH.

Figure 3: Please adjust the colorbars. For your work, the Baltic Sea is not important, so you might not need to go down to pH values below 8.05. Along the same lines, it would be helpful to adjust the range of the colorbars of the saturation states. For calcite the shown range seems to be 2-5 while the colorbar goes from 0 to 5. Similarily, you might be able to better show changes (c,f,i) if you chose an adjusted range. Consider also to mask the Baltic Sea as ESMs are usually not made for these small basins.

Figure 4: Can you show the section on a map, maybe in figure 3? As mentioned for figure 3, could you try to adjust the range of the colorbar. The shown colors do not seem to go far below the 8, but the range of the colorbar goes down to 7.8. Moreover, I find it confusing to see pH and the saturation horizon on the same plot. Wouldn't it be better to show the filled contours of the saturation state?

Section 5.3: In line with the comment above, I suggest changing the term "Present trends". I would furthermore suggest to restructure the section and to explain regions that are similar together. Going from region type to region type might be easier to follow than going from the surface ocean to the deep ocean. I struggled to read this.

Line 237: What do you mean by decreasing order?

Line 237-238: It is normal that largest decreases are seen at the surface, right? As the increase in atmospheric $CO_2$ is exponential, the decrease in pH should be larger in years that were recently in contact with the surface.

Line 238: Can you explain what this uncertainty refers to, how it is calculated?

Line 239: If you only have observations for a short time, does it make sense to calculate a trend over the entire period? You make an extrapolation and assume that the trend holds. I really do not think that this is a good idea.

Line 243: Can you provide uncertainties for the estimates by Skjelvan et al. (2014)?

Line 244: If Skjelvan et al. (2014) used different observations (sampling period, different region, and different seasons), can you compare these two estimates? If you provide this as an explanation for the difference, I would like a prove. Could you use the same region as Skelvan et al. (2014) and the same time period (1981-2013) to demonstrate that your hypothesis is right? If not, I do not see how these two estimates are comparable. And if they are comparable, I would like to know why you find a significantly reduced trend in pH. I would be really interested in knowing if a shift exists in the reaction of pH in the Norwegian basin after 2013. Alternatively, the change in the trend might also be due to the non-linear scale of pH referring to the question if a linear trend in pH (a non-linear unit) makes sense.

Line 247: I would not write relatively strong and just give the units.

Lines 253-259: Most of this is material for the Discussion and I suggest moving it there.

Lines 269-270: What exactly is "close"? Can you be quantitative?

Figure 5/6: Please zoom into each subplot. The way it is represented at the moment, it is hard or impossible to read the data and the uncertainties (see 6t as an example). I see that you want to have the same limits for all depth levels for better comparison but that makes most subplots unreadable.
Could you precise how you calculated the uncertainties in the aragonite saturation state and pH? Are they calculated in GLODAPv2? Did you calculate them using uncertainty propagation? Did you account for uncertainties in the equilibrium constants?

Line 278/279: Why do you present these numbers twice in the Results? I think I have already seen them earlier. Please add the uncertainties.

Line 283: What exactly is a "Small region"

Line 284: Please show the changes in $pCO_2$. You prominently highlight $pCO_2$ in the abstract, so I would like to see changes.

Line 295: Wasn't the seafloor already undersaturated? And isnt't it the change in the saturation state that leads to undersaturation?

Line 295/296: This is speculative. Can you add evidence for this hypothesis?

Figures 7/9: Please adapt the colorbar range according to the colors shown in the plots. I strongly suggest changing the colors around saturation states around 1. Having the same color for saturation states from 0.5 to 1.5 does not clearly indicate where waters change from undersaturated to oversaturated, although this is the most interesting region.

Figures 8/10: I suggest merging both figures. I would also show the aragonite saturation state instead of pH and I would add uncertainties for the saturation horizon (See comment above). More importantly, how representative is this section for the Nordic Seas. As you show, the acidification trends in deeper waters strongly depend on the regional sea. Thus, the rise in the saturation horizon should also depend on the regional sea. Is this section at the higher end or lower end of Nordic Sea acidification?

Lines 299-300: See comments above about the idea of changes in ocean pH expected from atmospheric $CO_2$ increase. It would be quite naïve to think that it the ocean pH depends only on the atmospheric $CO_2$ increase. Although it might be a useful comparison, I would not state it this prominently at the beginning of the Discussion.

Line 301: When did you show something about climate variability? Do you mean variability in the atmospheric climate? I really do not know what you are referring to here.

Line 307: Can you extend the buffer capacity explanation a little bit more?

Line 312: In the Fram Strait and the Greenland Sea, the slope of atmospheric $CO_2$ and marine $pCO_2$ is almost the same. The difference between atmosphere and ocean is only valid in a subset of the Nordic Seas, right? And if that is true, can you explain these differences?

Lines 314-328: You mention a lot of possible mechanisms and explain them well. This is what I would like to see earlier (in the Introduction). Here, it becomes all more understandable why you insist on the change in undersaturation. You say that any further exploration is beyond the scope of this study. I find this very disappointing. Your manuscript concentrates on this, leads the reader to this point. You show all changes and then stop at explaining them. It would really make the manuscript much stronger if you could somehow at least conceptually find the most important mechanisms in each region.

Line 330: Could you add some Discussion about the alkalinity behavior in the Barents Sea Opening. In light of the recent study by Asbjørnsen et al. (2020), it seems that Atlantic water extends far more northward and would thus increase alkalinity. It is more a displacement of Arctic Water in the BSO than a change of the existing water mass. Could you somehow elaborate on this and discuss possible impacts? You somehow waver around it, without really explaining the driving mechanisms.

Lines 337-341: It looks surprising that all the cited studies find a warming of the Nordic Seas while you find a cooling at the surface. You only give a possible explanation without evidence. This is disappointing as differences to other studies are most of the time what helps understanding the underlying mechanisms. Could you somehow test, if the sampling is the reason for the opposite trends? If it is, how robust are your estimates overall? If it is not the sampling, why are your results differen?

Lines 343-345: Is the increase in temperature in these depths caused by increased temperatures in Atlantic waters that flow northward at these depths towards the Arctic Ocean? Please use the Discussion to explain the mechanisms. What did Osterhus and Gammelsrod (1999) say about it.

Line 346: I think you already mentioned the compensation of $C_T$ and $A_T$? Do you need to repeat it here?

Lines 346-349: You point out regions where changes of $C_T$ and $A_T$ cancel each other without mentioning the drivers. I find that unsatisfactory. Once, you show me something interesting, I would like to know why this happens. Can I expect that to continue in the future, or is $A_T$ only temporarily slowing down acidification and once the $A_T$ stops, acidification will be even stronger than before? A lot of open questions that I would like to see discussed.

Lines 349-352: If the $C_T$ signal goes deeper in the Lofoten basin, this looks like stronger deep-water formation in this region. Is this the case? Could you test it in the model or with observations of transient tracers?

Lines 353-360: Can you explain why alkalinity changes are very important in the present day trends, but in the past and the future $C_T$ is by far the most important driver in all regions? I would suspect that it might be due to the model. If I understand everything right, Figure 11 is

based on observations, but Figures S9-S11 are based on the model. Could you make Figure 11 only with model results. If the alkalinity contribution is not significant in the model but in the observations, I think that the model misses important processes. In this case, many of the conclusions in this paragraph do not hold ($C_T$ being by far the main driver for the past and the future). If the model shows similar results as the observations, however, than it would be really interesting to know what is different in the present-day situation, maybe it is the beginning shift from constant to declining land and sea ice? And if the NorESM-ME model does not show the changes in alkalinity, it would be interesting to look at other models (something I think would be necessary anyway).

Line 360: This is speculative: If it is related to freshwater export from the Arctic Ocean, why would it only be seen in the Iceland Sea.

Lines 363/364: What do you mean by "We relate this … excess $CO_2$"? Please extend your explanation here. I think the undersaturation is just a "normal" part of the Nordic Seas and only in the RCP2.6 it is gone, because the atmospheric $CO_2$ trend reverses.

Line 374: This is speculative. You need to show the AOU and try to quantify the effect on $C_T$ if you want to make this point.

Line 376: You continue to speculate, including changes in deep-water formation. This is really interesting and makes it even more frustrating not to know why this happens. Can you try to elaborate on this? A reader does not want to be kept in suspense.

Line 389: Could you quantify this instead of saying most of? It is 321 out of 324, right? So you could just say: Out of 324 reefs in the Nordic Seas, 321 are at depths of 0-500 m for example.

Lines 393/395: I would not really spend so much time on one out of 324 reefs. It is not that important and somehow takes the wind out of the sails of your really strong message here.

Line 396: If I understand it right, under RCP2.6 and 4.5 only 3 out of 324 reefs will be exposed to undersaturated waters. That is good news. Can you try to make this clearer as it might be interesting to policy makers. At the moment you say the deepest reefs without saying how many reefs.

Figure 11: I really like the decomposition. Nice plot. However, as mentioned above, I do not think that the stars (expected trend) make sense for waters below the surface. As the deeper waters are not in contact with the atmosphere, they cannot see the accelerated trend in pH (see figure 2).

Conclusion: I find the Conclusion relatively weak in comparison to the findings of your paper. At the beginning it reads more like a summary of your results and it ends with a relatively complicated paragraph about the difference in partial pressure that is very technical. I would like to encourage you to really highlight your main findings and why they are important. You have many interesting messages in this manuscript!

Supplementary Figures: Please correct them in accordance with my comments on the figures in the main manuscripts.

**Technical notes:**

Line 4: I think it should be "from…to" instead of "since…to"

Lines 5-6: It is difficult to compare "in the last 40 years" to "between 1850-1980". Could you use the same format, for example "between 1980 and 2020" and "between 1850 and 1980"?

Line 13: I suggest to write "is projected to" instead of "will be"

Lines 20 and 24/25: "Since 1750" is repeated again, maybe you can look for another formulation?

Line 33: return**s**

Line 39: "Projected" instead of "expected"

Line 51: "are" instead of "is"

Line 51: Maybe write "qualitative effects" instead of "(direction only)"

Line 86: Is "visited" the right word?

Line 90: Are all these citations necessary? Can they be somehow grouped?

Line 136: I find it hard to understand the word "present trends". I suggest writing the trends over the last 40 years or something similar.

Line 184: An uncertainty of 0.00 looks weird.

Lines 185/186: Please use the same number of significant digits. And is mpH a common unit? I suggest writing $1e^{-3}$ pH, but it is your call.

Line 187: I would start a new paragraph here.

Line 212: To be consistent, I would not use the word saturation state but only $\Omega$.

Lines 247-250: Sometimes you give positive and negative trends. I see how that fits into your writing, but it is confusing. Could you stick to write about decreases and only give positive numbers or speak about trends and then give the negative numbers?

Line 275: Close to undersaturation or close to being undersaturated.

Line 273-277: I am not sure if this summary of the previous sections is needed.

Line 284: Delete "interestingly".

General: I think it is undersaturated with respect to. Could you change this throughout the manuscript please?

**References**

Asbjørnsen, H., Årthun, M., Skagseth, Ø., & Eldevik, T. (2020). Mechanisms underlying recent Arctic Atlantification. Geophysical Research Letters, 47, e2020GL088036. https://doi.org/10.1029/2020GL088036

Broecker, Wallace S., and T-H. Peng. "Gas exchange rates between air and sea." Tellus 26.1-2 (1974): 21-35.

Bronselaer, B., Winton, M., Russell, J., Sabine, C. L., & Khatiwala, S. (2017). Agreement of CMIP5 simulated and observed ocean anthropogenic CO2 uptake. Geophysical Research Letters, 44, 12,298– 12,305. https://doi.org/10.1002/2017GL074435

Dickson, Andrew Gilmore, Christopher L. Sabine, and James Robert Christian. Guide to best practices for ocean CO2 measurements. North Pacific Marine Science Organization, 2007.

Fassbender, A. J., Orr, J. C., and Dickson, A. G.: Technical note: Interpreting pH changes, Biogeosciences Discuss., https://doi.org/10.5194/bg-2020-348, in review, 2020.

Goris, Nadine, et al. "Constraining projection-based estimates of the future North Atlantic carbon uptake." Journal of Climate 31.10 (2018): 3959-3978.

Kwiatkowski, L. et al. Twenty-first century ocean warming, acidification, deoxygenation, and upper-ocean nutrient and primary production decline from CMIP6 model projections, Biogeosciences, 17, 3439–3470, https://doi.org/10.5194/bg-17-3439-2020, 2020.

Meinshausen, M., Smith, S.J., Calvin, K. et al. The RCP greenhouse gas concentrations and their extensions from 1765 to 2300. Climatic Change 109, 213 (2011). https://doi.org/10.1007/s10584-011-0156-z

Orr, J. C., et al. "Routine uncertainty propagation for the marine carbon dioxide system." Marine Chemistry 207 (2018): 84-107.

Tanhua, T., D. W. Waugh, and D. W. R. Wallace (2008), Use of SF6 to estimate anthropogenic carbon in the upper ocean, J. Geophys. Res., 113, C04037, doi:10.1029/2007JC004416.

Terhaar, J., Tanhua, T., Stöven, T., Orr, J. C., & Bopp, L. (2020). Evaluation of data-based estimates of anthropogenic carbon in the Arctic Ocean. Journal of Geophysical Research: Oceans, 125, e2020JC016124. https://doi.org/10.1029/2020JC016124

Terhaar, Jens, Lester Kwiatkowski, and Laurent Bopp. "Emergent constraint on Arctic Ocean acidification in the twenty-first century." Nature 582.7812 (2020b): 379-383.

van Vuuren, D.P., Edmonds, J., Kainuma, M. et al. The representative concentration pathways: an overview. Climatic Change 109, 5 (2011). https://doi.org/10.1007/s10584-011-0148-z

Wang, Chunzai, et al. "A global perspective on CMIP5 climate model biases." Nature Climate Change 4.3 (2014): 201-205.

---

## Referee Comment (RC2) · James Orr (Referee) · 28 Oct 2020

This manuscript uses observations and a model to assess the regional details of acidification of the Nordic Seas during the industrial era through to the end of this century. The authors find that during 1981-2019, the change in surface ocean pH is larger than would be expected from the corresponding change in atmospheric $CO_2$. They ascribe the cause to an evolution of surface ocean $pCO_2$, which while remaining undersaturated with respect to the atmosphere, increases at a rate faster than atmospheric $pCO_2$. They suggest that the main driver of the change in pH is the DIC increase associated with ocean uptake of anthropogenic $CO_2$. They also find that observed pH changes may be detected down to 2000 m in some parts of the Norwegian seas. The authors further focus on corresponding changes in the saturation state of waters with respect to aragonite and corresponding changes in the aragonite saturation horizon and what those changes may mean for cold water corals. In their model, most

cold water corals would not be exposed to waters that are undersaturated with respect to aragonite under the low-end RCP2.6 and mid-range RCP4.5 emissions scenarios. But under the high-end RCP8.5 emissions scenario, most of those corals would be exposed to such conditions, which are are unfavorable for their long-term survival.

Overall the authors have addressed an important topic, the details of acidification of the Norwegian Seas, a regional focus that has not been addressed previously. They appear to have used all the best data available for this assessment, thanks to the many coauthors with observational expertise in the Norwegian Seas. Also included are coauthors who are experts in using the chosen model routinely to assess ocean acidification and related aspects of ocean biogeochemistry. The Abstract and Introduction (sections 1-3) generally establish the need for this study, the Methods section appears to provide sufficient detail except for the final subsection, and the Results and Discussion sections reveal much effort being devoted to the analysis.

Yet despite these positive aspects, there is also much room for improvement.

CONCERNS in order of importance

(1) Unfortunately, there seems to be a complete lack of understanding of what a pH change actually means. Although pH offers a convenient way to represent the hydrogen ion concentration, its log scale means a pH change actually represents a relative change in $[H^+]$, not an absolute change (Kwiatkowski and Orr, 2018). That relative change is unlike the change in any other $CO_2$ system variable, all being absolute. Focusing only on pH and not $[H^+]$ can give a completely wrong impression, e.g., as in this manuscript when it is used to compare changes at different depths and at different locations (Fassbender et al., 2020). Looking only at pH change, as in the manuscript, we cannot know what part of the change is due to a change in $[H^+]$ and what part is actually due to differences in the reference $[H^+]$, the starting point. The manuscript neglects this key point entirely, not even mentioning hydrogen ion concentration. Remedying this problem will require major revisions.
(2) Projections with only one model are unreliable. Model projections are hard to publish nowadays without using multiple models and for good reason. One model can give very different results from others. A range of models provides an estimate of model uncertainty, and the model mean typically performs better than any given model. Because the ocean component of the NorESM1-ME model relies on a dynamic isopycnic vertical coordinate, we might expect it to have very different results in simulated deep-ocean anthropogenic carbon concentrations relative to most other CMIP models. Modeling centers such as the one where some of the authors of this manuscript are associated seem to now have access to and experience working with the CMIP5 or CMIP6 models. All analyses in the current manuscript need NOT be repeated with all models. But it will be needed to show at least where the NorESM1-ME model is situated relative to other Earth system models, in terms of the depth distribution of anthropogenic carbon concentrations (and perhaps also [H$^+$] and $\Omega_{Ar}$) in the different regions of the Norwegian Seas.

(3) The description of the decomposition of the drivers (namely the equations in section 4.4) is weak and incomplete.

a) Eq. (1) comes from Takahashi et al. (1993) and is fine except that the authors will need to replace the Greek delta $\delta$ with the correct partial sign $\partial$ in all the partial derivatives. This is not a major problem, just the convention of multivariate calculus. The $\delta$ is used for something else (inexact differential). Please don't confuse them.

b) Eq. (2) is added by the authors but is unnecessary. That equation comes from Metzl et al. (2010), who expanded each partial derivative in Eq. (1) to get at so-called "known quantities". Such complexity is no longer necessary because all of the partial derivatives in Eq. (1) are now easy available as precise quantities in "derivnum", an add-on package to CO2SYS-MATLAB (Orr et al., 2018). See

https://github.com/jamesorr/CO2SYS-MATLAB

The simpler choice, just deleting Eq. (2), is preferred and avoids unnecessary complexity that can lead to mistakes in implementation. For instance, the authors four definitions that immediately follow Eq. (2) are ambiguous because they are missing key parentheses. Hopefully their actual code is less ambiguous. There is no longer any need to introduce all these extra terms.

c) Eq. (3) should be recast in the same pattern as Eq. (1), i.e., replacing $fCO_2$ with $[H^+]$. It should not be cast in terms of pH (as in the current manuscript) for reasons mentioned in (1) above. The partial derivatives of $[H^+]$ are also available in derivnum. That routine is called with the same arguments as CO2SYS, with one argument added in the beginning to specify what the user wants to take partial derivatives with respect to. This further move towards simplicity will avoid the old-fashioned complexity that is now in the manuscript. Furthermore, this change will help avoid misinterpretation of what changes in pH mean.

d) An equation is missing in Section 4.4 concerning the freshwater Taylor-series decomposition, results of which are presented in Fig. S8. With that equation, the appropriate citations need to be given, starting with Lovenduski et al. (2007). For the associated salinity normalization, the authors must also specify their choices of the regional salinity references and if those remain constant or change with time. Furthermore, the authors will need to mention why they generally seem to prefer to use the older, less complicated decomposition from Takahashi et al. (1993).

e) Another equation is missing in Section 4.4 concerning what the authors call "pHperf". Currently that term is mentioned in the short final paragraph of section 4.4, where the authors attempt to describe how they compute "the pH change in seawater that perfectly tracks atmospheric $CO_2$". Unfortunately, the current description does not tell us exactly what the authors have done. For instance, in the calculation of pHperf, do the authors use *i)* the actual atmospheric $pCO_2$ as the reference value along with the atmospheric $pCO_2$ change or *ii)* the oceanic $pCO_2$ as the reference value, to which they add the change in atmospheric $pCO_2$?

The importance of this question is illustrated with a simple example. Suppose atmospheric $pCO_2$ is at 400 $\mu$atm and oceanic $pCO_2$ is at 300 $\mu$atm. Although a 1 $\mu$atm change in $pCO_2$ starting at 300 $\mu$atm produces only a 0.7% greater change in $[H^+]$ when compared to starting at 400 $\mu$atm, the corresponding change in pH is 30% greater in the former relative to the latter. The reason is that a change in pH represents a relative change in $[H^+]$, i.e., relative to the $[H^+]$ of the starting point. These numbers slightly depend on the other reference conditions, which I have arbitrarily set to T=2°C, S=35, ALK=2300 $\mu$mol/kg, nutrients=0. If the authors have used approach *(i)*, the results will be wrong. The authors should be able to resolve this issue by using approach *(ii)* and by adding an equation and improving the text to avoid ambiguities.

A related minor question: Do the authors actually use atmospheric $xCO_2$ (ppm) or do they first convert that to atmospheric $pCO_2$ ($\mu$atm), making corrections for water vapor pressure and atmospheric pressure?

(4) The section on cold-water corals is too cursory. The authors' analysis of the change in the aragonite saturation state to which cold-water corals are exposed has potential, but the authors devote only one rather short paragraph to describing the results, which are presented in one figure. They authors also neglect to clearly attribute previous studies that have attempted the same type of exercise using model projections and cold-water coral positions. Additionally, the data set used in the manuscript for coral positions is not cited adequately, and the authors do not give enough information about their procedure for extracting the saturation state from the model. For instance, is the model sampled at the depth of the coral (as provided in the data base) or is the depth taken to be that of the model's bottom depth at a coral's latitude and longitude? More discussion of results and the addition of uncertainties from a multi-model analysis would seem critical.

(5) The writing needs improvement. Getting through this manuscript was not easy. Although there are few if any errors in English, and individual sentences generally work well, the manuscript would benefit if the authors could redouble their efforts to improve

flow between sentences. That is, connections between sentences are often rough, causing the reader to slow down and sometimes stop. Also lacking is coherence in many individual paragraphs. My recommendation would be for the authors to consult the book by J. M. Williams (Style The Basics of Clarity and Grace), and in particular the short chapter on Cohesion and Coherence. Then they could go through the manuscript trying to improve both aspects. If one cannot borrow this book from a library or colleague, older editions only cost about 10 euros. It offers the potential to dramatically improve one's writing by applyng a few basic principles.

(6) The figures need improvements. Some figures appear to have too many panels, some figures should be combined, and some figures should be deleted. There are also other issues.

a) In Fig. S6, it seems that only 3 out of the 6 regions seem to show a trend in surface ocean $pCO_2$ that is significantly greater (statistically speaking) than that of atmospheric $pCO_2$. Thus I am unconvinced by the statement that it is only the Barents Sea Opening does not follow this pattern. More care is needed when handling this subject in the revised manuscript.

b) In Figs. 3, 7, and 9, the third row of maps for $\Omega_{Ca}$ should be deleted because it exhibits the same patterns as for $\Omega_{Ar}$ in the second row, only differing by a constant. Their constant relationship could be briefly mentioned once in the text rather than wasting space in each of those three figures. Likewise, Fig. S5 for $\Omega_{Ca}$ should be deleted because it shows exactly the same patterns as $\Omega_{Ar}$ in Fig. 6.

c) In Figs. 5 and 6, the numbers given in each panel for the slope and uncertainty should be moved to a table, where it will be easier to compare numbers between regions and depth layers. The same goes for the corresponding supplementary figures (Figs S1-S4). In addition, there are often too many significant figures in the slope and uncertainty, and the number of digits is not always consistent. Furthermore, in those same supplementary figures, the slopes have the wrong units. In regards to these

and other figures, when statistical significance is mentioned in the text, that should be backed up with a statement of how it was determined. Such is not the currently the case in the manuscript, but it is critical, e.g., when discussing if oceanic $pCO_2$ is increasing more rapidly than atmospheric $pCO_2$ (Fig. S6).

d) Figs. 8 and 10 should be combined.

e) Fig. 11 includes some details that might need to be deleted, and corresponding supplementary figures should also be refined. What is the rationale for including the dashed line and black stars in subsurface layers? Those layers have been isolated from the atmosphere for some time and we would not expect them to track atmospheric $CO_2$. Showing these details in subsurface layers will confuse the reader. Moreover, would it not be better to devote a separate figure just to the subject of ocean $pCO_2$ tracking atmospheric $CO_2$ rather than trying to squeeze that information here into a very small space? Fig. S6 fills this need well. That could be brought up into the main paper. Only the top level (0-200 m) of Fig. S6 would need to be shown as we do not expect subsurface levels to track current levels of atmospheric $CO_2$. I also worry about how representative the 0-200 m layer is of surface ocean $pCO_2$. Some discussion on that and perhaps a modified figure seems necessary.

In corresponding supplementary figures for the model (Figs. S9-S11), the authors miss the opportunity to compare the model results over the same 1981-2019 period as used for the model. By the way, why is this time span often referred to in the text as lasting 40 years; actually, it lasts only 39 years. My impression is that relative to the observations, the model is dominated even more by the change in DIC, based on the analogous plots for the previous and subsequent time periods. These supplementary figures concern the model, but readers will be confused because 'OBS' is used to designate the model result, both in the caption and in the figure itself. Please change 'OBS' to 'MOD'.

f) Fig. 12 has too much white space.

g) The supplementary figures are mentioned out of order.

These are issues that I hope the authors will be able to address with major revisions to the current manuscript. I look forward to the opportunity of seeing a revised manuscript.

REFERENCES

Fassbender, A. J., Orr, J. C., and Dickson, A. G. (2020). Technical note: Interpreting pH changes, Biogeosciences Discuss., https://doi.org/10.5194/bg-2020-348, in review.

Kwiatkowski, L. and Orr, J. C. (2018) Diverging seasonal extremes for ocean acidification during the twenty-first century. Nature Climate Change 8, 141–145, https://doi.org/10.1038/s41558-017-0054-0

Lovenduski, N. S., Gruber, N., Doney, S. C. and Lima, I. D. (2007) Enhanced $CO_2$ outgassing in the Southern Ocean from a positive phase of the Southern Annular Mode. Global Biogeochem. Cycles, 21, https://doi.org/10.1029/2006gb002900

Orr, J. C., Epitalon, J.-M., Dickson, A. G. and Gattuso, J.-P. (2018) Routine uncertainty propagation for the marine carbon dioxide system. Marine Chemistry 207, 84–107, https://doi.org/10.1016/j.marchem.2018.10.006

---

## Referee Comment (RC3) · Anonymous Referee #3 · 3 Nov 2020

This is an interesting and ambitious manuscript with important goals. I commend the authors on their substantial efforts in synthesising all the varied data streams (real and modelled) to put together the trend analyses, projections, and regional maps. While these synthesis figures don't deliver revolutionary new insight in a purely academic sense, they are extremely important and highly sought after in more policy-oriented applications. These results are certainly worthy of eventual publication.

I write this having also read the two existing peer reviews of this manuscript. I agree with the concerns of the other reviewers that many results are presented with either no or insufficient quantification, and/or too vague or incomplete conceptual explanation. This is my main concern about the manuscript as it is.

This review is so brief because there are not many points left to make without simply repeating the thorough work of the other reviewers. Other than the main concern noted

above, I have only a few minor additions:

Abstract: is "window to the deep ocean" the right metaphor here? A place through which the deep ocean can be observed – is that the intended meaning?

Abstract: sensitivity to OA in the Nordic Seas is not directly due to high latitude, but rather due to low water temperature?

Sections 1 and 2, and probably also 3, are very much introductory material and I would also suggest to consider combining them, as mentioned by another reviewer.

In Section 2 and Table 1, an important aspect of discussion is absent, that is about the timescale of the effects shown in Table 1. Are you showing instantaneous effects of T/S/DIC/TA increases, or effects after $CO_2$ re-equilibration with a constant atmosphere? Looks like it's the former – is that really appropriate, given the context?

Methods: given the relatively low temperature of your observations, why not use the Sulpis et al. (2020) carbonic acid constant parameterisation for your $CO_2$ system calculations?

Throughout: pH is dimensionless; pH values do not need the word "units" after them, and $\times 10{-3}$ can be used in place of "mpH".

Line 205: "DIC also relates to salinity" is very vague, please explain the mechanism– including timescale considerations noted above for Section 2 / Table 1. See e.g. Wu et al. (2019).

Line 424 "both" implies two options when there are three (past, present and future). I am not sure that the chain of causality is properly represented in this and the subsequent sentences (i.e. which are drivers and which are responses in terms of air-sea $CO_2$ disequilibrium, pH change and hydrographic conditions), please be careful with the exact phrasing here.

The request for more research at the very end of the manuscript is very unspecific

and is unexpected given that the rest of the paragraph implies that all the observed phenomena have indeed been explained here.

**1  References**

Sulpis, O., Lauvset, S. K., and Hagens, M.: Current estimates of $K_1^*$ and $K_2^*$ appear inconsistent with measured $CO_2$ system parameters in cold oceanic regions, Ocean Sci., 16, 847–862, https://doi.org/10.5194/os-16-847-2020, 2020.

Wu, Y., Hain, M. P., Humphreys, M. P., Hartman, S., and Tyrrell, T.: What drives the latitudinal gradient in open-ocean surface dissolved inorganic carbon concentration?, Biogeosciences, 16, 2661–2681, https://doi.org/10.5194/bg-16-2661-2019, 2019.

---

## Author Comment (AC1) · 23 Jun 2021

**Responses to Reviewer 1:**

*We want to thank reviewer 1 for the extensive work and very detailed review. We enjoyed working through the comments, and we think that they have led to a major improvement of the manuscript. Please find below our responses (in blue italics) to all the comments (in black.)*

The authors have put together an overview of past, present, and future ocean acidification in the Nordic Seas using measurements, gridded assembled data, and model projections. The work thus presents an exhaustive overview of ocean acidification in the Nordic Seas that resembles most available information and will as such certainly prove to be very useful for other scientists and to people outside of Science, such as policy makers. Personally, I very much enjoyed reading the manuscript and want to congratulate the authors. The manuscript would, however, significantly improve if underlying processes would be discussed further to explain the drivers between the different reaction of the regional seas to ocean acidification. What drives the changes in alkalinity, dissolved inorganic carbon, temperature and salinity? Is it changes in water masses, circulation, deep water formation, freshening from the Arctic Ocean or land ice melt from Greenland? A better explanation would help to understand past and future changes and also help to understand how well the one model used in this study performs regarding the expected changes. Furthermore, I have a number of questions regarding the Methods that would need to be addressed before publication.

*We agree that the discussion on underlying processes of observed changes can be improved, and we therefore address this more fully in the revised version.*
*We have found that the most important driver in all regions and over all time scales is the increase in DIC from anthropogenic $CO_2$ uptake. The second most important driver is a general increase in alkalinity. Because changes in alkalinity related to salinity changes are cancelled out by the respective changes in DIC, the processes of alkalinity-driven pH changes are due to biogeochemical processes. The third most important driver of pH change is increases in temperature. We have added references that discuss the reasons behind these changes, and the relative role of different water masses in Section 4.1.*

**Major comments:**
General: Please be quantitative as often as possible and avoid formulations like "it agrees well. This should be carefully addressed throughout the manuscript.

*We have carefully gone through the manuscript to make sure that we provide quantitative assessments. In particular, we have made an additional analysis of the present distribution of pH and saturation states to quantify the underlying drivers of spatial variations (sections 2.2.1 and 3.1).*
*However, we prefer not being too quantitative/detailed in the abstract; it would make it too exhaustive. This is why we have not followed the suggestions of the reviewer (that are found further down) in this part of the manuscript.*

Introduction: I suggest merging sections 1,2, and 3. Some parts are repeated, and it would be helpful to have some information earlier. For example, I think a better explanation of how ocean acidification works chemically is needed and what the saturation state is and why it matters (or why not?). Why do the Nordic Seas have naturally low saturation states, etc. It is

absolutely right, that the Nordic Seas are of high importance and very interesting, so strive on it. Why does it have short residence times and what is the link between the residence time and the pCO 2 undersaturation? Which water masses meet? It all makes more sense after reading the sections 2 and 3, but section 1 is not understandable for oceanographers who do not know the Nordic Seas. Overall, I think most of the information is in the manuscript, but restructuring could strongly improve the Introduction. And as you discuss future projections, you could also highlight the difficulties that models have in these regions.

*It is a good idea to do some merging of these sections. After some thinking, we ended up dividing the introduction into two subsections; one general introduction on ocean acidification and the Nordic Seas, and one on "theoretical background", where we give an introduction to how ocean acidification works with some chemcical background. We also added some text on current modelling of the region.*

Undersaturation of p$CO_2$ : Please introduce the concept of undersaturation carefully. Presented like this in the abstract without explanation leads to large confusion. Most of the time, I think I understand what you mean with the concept of undersaturation of p$CO_2$ . At other times, I am very confused when you compare the pCO 2 in deep waters (2000 m) to pCO 2 in the atmosphere at the same time. Given that the age of the water masses at 2000 m is likely much older, it has "seen" the atmosphere a long time ago. How does it make sense to speak of undersaturation or a perfectly tracked pH? I think a lot of people are not familiar with this concept, including. How do you calculate/define undersaturation? Why does the undersaturation exists in pre-industrial conditions? I suppose that northward flowing waters cool down and their pCO 2 decreases faster as their solubility increases than they can take up CO 2 from the atmosphere (Broecker et al., 1974). If that is the dominating process, the weakening of the undersaturation is somehow expected, as cold waters hold less C ant . With increasing atmospheric CO 2 , the differences in surface ocean C T should become smaller, which would also lead to a smaller difference in marine and atmospheric pCO 2 in the northward flowing waters in the Nordic Seas. It should be very similar to the processes in the Barents Sea, where Terhaar et al. (2020) showed that C ant is larger in the northward flowing waters than expected from atmospheric CO 2 concentrations. This larger C ant is equivalent to a weakening of the undersaturation of C T . With a changing climate, other drivers could potentially also become important, such as a change in circulation, i.e. is the quantity of northward flowing waters larger or smaller, are they flowing faster or slower, do they extend further into the Nordic Seas? all these factors could come into play regarding the undersaturation and need to be understood to properly discuss this effect.
Overall, I think this concept needs to be introduced in more detailed with the respective literature if it is a core concept of the paper. In fact, much is explained in the Discussion (Lines 314-328). This should be moved to the Introduction. In the Discussion, it would be really great if you could pin down the most important mechanisms for each regional sea. As it is now, the manuscript is more an atlas to look up numbers, explaining the mechanisms would put it on a whole different level. I strongly encourage you to do this.

*We agree that the concept of undersaturation of p$CO_2$ needs a better introduction and discussion, and be used more carefully, such as avoiding this term when discussing deep waters. We have therefore added some text about the air-sea p$CO_2$ difference in the introduction, both in the paragraph about the Nordic Seas, and also in the subsection theoretical background. Here, we now explain what we mean by p$CO_2$ undersaturation, and*

*the processes underlying the pCO$_2$ undersaturation in the Nordic Seas (lines 38-41). We have further removed the comparison with atmospheric CO$_2$ change in the subplots of current trends below surface waters. However, because changes in the degree of undersaturation is not the main topic of the paper, we don't want to go too much into details in the introduction. To this end, to actually pin down the underlying processes we would need an extensive analysis far beyond the scope of this paper, for example one would need to take into account upstream changes in the seawater chemistry, which would require a larger set of observational data.  This is why we have left the discussion on the possible mechanisms behind a change in the pCO$_2$ undersaturation as it is. Here, we are also discuss the potential role of an increasing import of anthropogenic carbon as suggested by Olsen et al. (2006) and Anderson and Olsen (2002), which is similar to the process that you mention was described in Terhaar et al., 2020.*
*Also, please note that after the analysis of  the impact of seasonal undersampling, we found that the uncertainty is too large for us to state that the ocean pCO$_2$ is increasing faster than the atmospheric one. We therefore removed the statement on pCO$_2$ from the abstract. We added a discussion on this in Section 4.1.*

*Anderson, L. G. and Olsen, A.: Air–sea flux of anthropogenic carbon dioxide in the North Atlantic, Geophysical Research Letters, 29, 16–1–16–4, https://doi.org/10.1029/2002GL014820, https://agupubs.onlinelibrary.wiley.com/doi/abs/10.1029/2002GL014820, 2002.*

*Olsen, A., Omar, A. M., Bellerby, R. G. J., Johannessen, T., Ninnemann, U., Brown, K. R., Olsson, K. A., Olafsson, J., Nondal, G., Kivimäe,C., Kringstad, S., Neill, C., and Olafsdottir, S.: Magnitude and origin of the anthropogenic CO$_2$ increase and 13C Suess effect in the Nordic seas since 1981, Global Biogeochemical Cycles, 20, https://doi.org/10.1029/2005GB002669, https://agupubs.onlinelibrary.wiley. com/doi/abs/10.1029/2005GB002669, 2006.*

Line 9: You point out that the Barents Sea Opening is different to all other regions because of an increase in AT . This sounds very interesting and it is thus even more disappointing that no explanation exists why the A T increases? Is it via biological processes such as an increased production of organic matter via primary production or is it a change in circulation, i.e. are Atlantic water masses with higher AT expanding further and displacing Arctic Ocean waters? I really would like to have an explanation here.

*We agree that the strong increase in AT is very interesting. Unfortunately, it is not possible to make any conclusions on the underlying mechanisms from the little data we have, and we can only discuss the possible reasons behind. The decomposition into a freshwater-driven component and a biogeochemical - driven component (Fig. S19) suggest that it is not a result of changing salinity, but of biogeochemical processes, or possibly sampling issues. One biogeochemical process that could impact AT is the recurrent blooms of coccolithophores in the Barents Sea, which have shown a tendency to increase.  We are now discussing this in section 4.1 at lines 490-502.*
*Please note that, after investigating the effect of seasonal undersampling, we found that it is the apparent cooling that lies behind the weak trend in pH, and not the strong increase in alkalinity (there is also an exceptionally strong increase in CT, counteracting the effect of alkalinity).*

Line 16-17/ lines 119-122: No need to request further model studies. The output from CMIP5 and CMIP6 models are online available. While differences in pH and saturation among models are generally small between models, they can be large below the surface in the North Atlantic (Goris et al., 2018) and in the Arctic Ocean (Terhaar et al., 2020b). I suspect thus that the differences in the Nordic Seas would be large, too. Moreover, the NorESM-ME

model seems to be at the high end of carbon uptake in both, the North Atlantic and the Arctic Ocean and thus likely at the high end in the Nordic Seas as well. This is likely caused by an overall strong AMOC in the model (Wang et al., 2014).

Given that the NorESM-ME model is at the high-end for the North Atlantic anthropogenic carbon sink, I doubt that it is representative for future ocean acidification. The cited papers only treat the surface (mainly driven by atmospheric pCO 2 ) and the bottom ocean where changes in all variables are small and models "automatically" agree. I am not aware that these papers analyze model differences in waters from 100 to 2000 m. Given the detailed analysis of the historical data, I strongly believe that it is necessary to analyze a larger number of models with annual mean data. I feel that it is absolutely necessary in order to make strong statements about the future acidification in the Nordic Seas, such as that undersaturation can only be avoided for RCP2.6 and RCP4.5.

*Thank you for this comment. We decided to analyze the output of several CMIP5 models for the section on cold-water corals (section 4.2), where the results (i.e. the future location of the saturation horizon), is sensitive to model uncertainties. For the other sections, which mostly focus on drivers and thus process understanding, we decided to stick with NorESM1-ME only. We additionally made a time-series plot showing the simulated pH at three different depths of the Nordic Seas from NorESM1 and several CMIP5 models, that we put in the supplementary material (Fig. S5) and that we mention in Sections 3 and 4.2. Unfortunately, for emission driven runs, only output for the high emission scenario (which we used for the manuscript) are available on the CMIP5 data portals, which is why we could not make a sensitivity analysis for the other scenarios.*

Lines 95-97: Annual means/seasonal bias: Although not enough data exist to de-seasonalize the data, the model output gives you several possibilities to assess the possible uncertainties.You could subsample the model at the same time and place and compare these to the model annual mean values. Or even easier, you could compare the seasons within the model output to understand the seasonal variations. And are the under sampled months crucial in these regions, i.e. is winter under sampled and is deep-water formation occurring in winter? If that is the case, you might even expect a higher pCO 2 in winter. Given the strong seasonality, I feel that an analysis of the potential bias is necessary.

*Thank you very much for pointing this out!*
*The idea of subsample the model to get an estimate of the uncertainty related to the seasonal undersampling is a good thought. However, there are also uncertainties related to such a model-observation comparison, which would make it questionable. These uncertainties are related to the model's capability to simulate the seasonal cycle (including the magnitude of the seasonal drawdown of DIC, and the timing of blooms), which have not yet been fully evaluated for the Nordic Seas. We believe that an evaluation of the seasonal cycle of NorESM1-ME lies beyond the scope of this paper, as we focus on longer time scales.*

*To address the issue of trends being biased due to an uneven sampling between winter and summer months, we instead calculated the trends for the productive season only (months March -September).*

*It did not result in any significant differences in the pH trends in the 0-200m layer, but it made us realise that the seasonal subsampling has quite an effect in the regions that we thought were sticking out. This includes the weak negative trend of pH in the Barents Sea Opening and the apparent cooling we see in the surface. Further, we found that our results with the sea-surface pCO₂ increasing at a faster rate than the atmospheric one were very sensitive to the choice of months, which makes these results very uncertain. This is discussed in section 4.1.*

pH uncertainties: I have trouble in seeing the standard deviation over the entire Nordic Seas as an uncertainty. This is rather a measure of heterogeneity. The large gradients in T, S, C T , and A T over the Nordic Seas thus increase artificially the uncertainties in pH when compared to observations. I have no good proposition but I would consider using uncertainties from the dissolution constants using mocsy2.0 (Orr et al., 2018) instead of the heterogeneity. Whatever you do in the end, it would be great if you could find a solution that permits to compare the model to the observations on the same basis.

*The purpose of figure 2 is to give an overview of the different datasets and the acidification in the Nordic Seas. The idea behind the use of spatial STD for the model output is to show that the model and observations compare reasonably well. Because we use a spatial mean for the whole region from the model, which is not exactly comparable to a mean of the pointwise measurements from observations, we also want to put in the modelled spatial heterogeneity.*
*Considering that the objective of this paper is to analyse the observation and model data separately (present, past and future climate), we do not think that we need to make a more advanced comparison between the two datasets.*

*You are, however, right that it is not correct to call it an uncertainty, and that it should be heterogeneity. We have corrected this.*

Trends and significance: In many figures you show trends for each subsea and depth. I am still wondering how you calculate if a trend is significant (maybe I missed the explanation). Especially confusing are for example Figure 5f,g. Measurements are only available for the 2 nd half of the period and uncertainties become enormous for 1981 as no measurement exists there. How is it possible to speak of a significant trend given the large gap in observations? I am not strong in statistics and would need a better explanation here.

*We now explain how the significance is calculated (lines 225-227 in the revised manuscript). We have additionally removed the 95% confidence intervals (the parabolic lines) from all figures. These lines showed the confidence intervals of y-values when using the equation obtained from the regression. We realise that this does not add any value to the paper, and that the significance of the slope, indicated in Table 5 and 6, is enough.*

**Minor notes:**
General: I would suggest using C T and A T instead of DIC and ALK as recommended by the guide for best practice (Dickson et al., 2007). Along the same lines, please write *pCO₂* instead of p$CO_2$ .

*We have changed this according to your suggestions.*

Line 2: Why do the high-latitude location lead to a higher sensitivity to acidification? People who are not ocean biogeochemists might not directly understand that high-latitude oceans have a higher solubility, hold more dissolved inorganic carbon (C T ) and have thus a naturally lower pH and lower saturation states. I feel that it would be helpful if you lead the reader a bit more here. Moreover, aren't projected changes in the saturation states smaller in the high latitudes than in the tropics because the absolute numbers are already lower. It would probably be similar for pH? So strictly speaking, the Nordic Seas are more vulnerable towards ocean acidification because they start with more C T and lower saturation states and pH, but they are not more sensitive.

*The reviewer is correct that we want to refer to the lower saturation states induced by the low temperatures. It is therefore more appropriate to use the word vulnerable than the word sensitive. We have corrected this. However, projected changes in the high latitudes are smaller because of low initial carbonate ion concentration. It is the opposite for pH, we expect larger changes at high latitudes because lower carbonate concentration equates with lower buffer capacity. We now also give a more thorough introduction to this in the subsection Theoretical Background.*

Line 4: Could you be more exact instead of writing pre-industrial? The definition of preindustrial is crucial for the historical anthropogenic carbon uptake (Bronselaer et al., 2017) and in consequence also for historical acidification.

*In general we want to avoid too much detail in the abstract, and will therefore not clarify it here. However, we are now clarifying pre-industrial in the main manuscript in relation to each data-source.*

Line 6: Please be quantitative. How much larger are they?

*We have removed the lineas about the change in $pCO_2$ undersaturation from the abstract, as we found these results to be very sensitive to the choice of season (see explanation above).*

Line 10: What exactly is the "acidification signal"? Which regions are you speaking, please be more specific? And 1000-2000 m is not very precise for the saturation horizon at present-day conditions. It would also be helpful to know where it was in pre-industrial conditions? Did it increase by 50 m or by 1000 m?

*We have removed the use of "acidification signal" from the manuscript.*
*Regarding the saturation horizon: with the current analysis that we do in the manuscript we cannot be more precise than this. We are now mentioning the depth of the pre-industrial saturation horizon in the abstract.*

Line 11: What do you mean by significant?

*We have removed the word significant from the abstract. We now explain how we calculate significance in section 2.2.2 in the main manuscript.*

Line 12: The switch between two sentences from present day saturation states to future pH makes it hard to read and to compare.

*We have made an attempt to make the abstract clearer by some rephrasing and restructuring.*

Line 13: Again, a switch from surface pH to the saturation state over the entire water column. You are losing me here.

*We have made an attempt to make the abstract clearer by some rephrasing and restructuring.*

Line 13: I suggest to write "high-emission" instead of "worst case".

*We have changed this according to your suggestion.*

Line 15: At what depths exactly? Everywhere or online over a depth of 10 m?

*After revision of the abstract this sentence has been removed.*

Line 15: What is the majority? Literally all or 51%?

*After revision of the abstract this sentence has been removed.*

Line 20: Are the 25% exact? Or around 25%?

*You are right that it is not exactly 25%. We will change this to "about 25%".*

Line 22: "More serious downside" is not necessary. Just delete that part of the sentence to make it shorter.

*We have revised the sentence taking into account the reviewer's comment.*

Line 23-24: What threat are they imposing? Can you be more precise please?

*We have added some text explaining this in the first paragraph of the introduction.*

Line 27-29: I think this sentence is wrong. I am not aware of well constrained ocean acidification at global scale. This might hold for the surface ocean but definitely not for subsurface to deep ocean basins. Differences in models lead to large differences (see bottom ocean pH in Kwiatkowski et al. (2020) in regions where waters reach the bottom. As soon as the waters leave the surface, I believe that historic acidification and its projections are rather ill constrained.

*The reviewer is correct that also the subsurface is ill-constrained, which is one of the reasons that we put some focus on that in this manuscript. We have reformulated the sentence.*

Line 34: Here, you could introduce your acidification signals.

*We now avoid the term " acidification signal" in the manuscript.*

Line 37-38: I like this last sentence, maybe you could make it a topic sentence of the paragraph. Please also write that and why the Nordic Seas have low saturation states.

*We now give a deeper explanation of the connection between cold waters and low saturation states in the subsection Theoretical background. It is a good thought to make this sentence a topic sentence of the last paragraph. However, after all the revisions of the introduction, we think that it would not fit in anymore.*
*We also explain the low saturation states in the paragraph on the Nordic Seas.*

Line 48: I am confused by the word "anthropogenic" here. Above, you used it in combination with anthropogenic carbon. I suppose now it is not to be read in combinations with carbon because changes in T, S, and A T are also anthropogenic, right?

*It is correct that it should have been anthropogenic carbon. After the re-organisation of the introduction, this sentence has been removed.*

Table 1: I am not sure if this table is needed. If you keep it, pleas define "minor".

*We think that the table gives a visual aid for the reader to follow the rest of the manuscript, and would therefore like to keep it. We now write in what direction salinity affects pH.*

Line 60: I think you can delete large parts of this sentence, such as "by the strongest base in seawater" and all after "which has been supplied ...". It does not seem to be such an important information in this context.

*It has been deleted.*

Figure 1: Please add a colorbar. Could you change the color for the continents, it is hard to distinguish continents and shelf seas? It is also very difficult to see the dashed lines, could you somehow make them stand out better? A lot of points are outside your defined region, did you use these points? If not, I would suggest deleting them to make the figure less busy. Could you also highlight the two timer series stations that you mention in 4.1?

*We have edited the figure after the suggestions of the reviewer. The only thing that we did not do was to remove the data-points outside of the Nordic Seas. We think that it is important to show how the choice of the region-boundaries impacts the amount of data.*

Section 4.1: The section is highly important, but it is difficult to read and to extract the information. For example, the topic sentence speaks about DIC, ALK, temperature and

salinity. Later on, uncertainties are also reported for oxygen and nutrients. Did you also use oxygen and nutrients or only the 4 variables in the topic sentence? If you did not use nutrients and oxygen, no need to mention them.

*Nutrients were used for the CO2sys calculations, so we will keep them. We also added silicate and phosphate to the topic sentence. However, we did not use oxygen, so it has been removed.*

Could you also please provide an assessment of how much volume of the Nordic Seas is covered by the GLODAPv2 data? Is it mainly surface data, how deep does it go, are some regions under sampled? Given, that you look at changes over the data collection time period, it would also be useful to see the measurement on a timeline to see if the measurements are biased towards the end of the period.

*We think that our time series figures ( Fig. 6 & 7), together with the map showing the regional distribution of samples (Fig. 1) well illustrate the time/space coverage of the data we use and provide enough information for the content of the paper. We have, however, added some figures showing the sampled seasons per year in the supplementary material (S8-S13), that will support our discussion on seasonal undersampling.*

Please use paragraphs to divide between GLODAPv2, the two time series stations and the framework of the Norwegian ocean acidification monitoring program. This will likely help to read the section. By the way, what exactly it the Norwegian ocean acidification monitoring program? Could you introduce it with one sentence?

*We have divided the text into paragraphs and added some information on the Norwegian ocean acidification monitoring program.*

Line 86: Please be more precise than "approximately 4 times". Are they covering all seasons? Can be you be more precise than 10-20 depth levels? Are these depth levels concentrated at the surface or evenly spaced?

*We are afraid that we cannot be more precise than this because it has been varying in time. We think that our time series figures, together with the new supplementary figures S8-S13, provide the information on the time/depth coverage that is necessary for the manuscript.*

Line 109: Is there a reason why you did not include RCP6.0? Could you give the citations for the scenarios (Meinshausen et al., 2011; van Vuuren et al., 2011)?.

*We have added the citations. With the choice of RCP 2.6, 4.5 and 8.5 we wanted to include one mitigation scenario, one stabilization scenario and one high-emission scenario (van Vuuren et al., 2011), which we now explain in the manuscript. RCP6.0 is another stabilization scenario, and is not included because it would make the paper too long.*

Lines 110: What is similar? Please be quantitative. And please make the comparison for the Nordic Seas. If the NorESM2 model is doing a good job in the Pacific and Southern Ocean, what does it tell me with respect to the Nordic Seas?

*We removed this sentence because this paper does not discuss the performance of NorESM1-ME in the Nordic Seas.*

Line 112: What is comparable? Please be quantitative here. The reader can then judge if that is good enough or not.

*We have added some text to provide a more quantitative assessment.*

Line 113: This is definitely a red flag to me. What is broadly? How much? You do not even show me what happens under the surface. What simulated acidification? Over which period? The relatively close agreement at the surface is somehow expected as pH follows mainly the atmospheric $pCO_2$. Below the surface, I do not believe this statement without seeing it. Please revise the model evaluation and really try to convince me that the model works. As it is written now, I am absolutely not convinced.

*We have added a timeseries figure with modelled pH from several CMIP5 models at three depth ranges in the supplementary material (fig. S11).*

Lines 113-115: Do you add relative changes in pH and saturation states to the absolute saturation states? I think you should not do this with non-linear variables (Fassbender et al., 2020). Please add simulated changes in $C_T$, $A_T$, T, and S to the respective variables in GLODAPv2 and use the projected $C_T$, $A_T$, T, and S to calculate projected pH and saturation states. And please precise how saturation states are calculated within the model world, using CO2SYS or mocsy2.0? Are you using simulated nutrients as you do for the observations?

*We are grateful that the reviewer (and also reviewer 2) identified this problem. In the first version of the manuscript we added the modelled changes of pH and saturation states. In the revised version we have calculated future and past pH from 3D fields containing the GLODAP climatology plus the modelled changes of temperature, salinity, alkalinity, DIC and nutrients. This is described in Section 2.2.2.*

Lines 124/125: Could you assess how good the pre-industrial $C_T$ values from GLODAPv2 are? If I understand right, a very basic version of the TTD method was used to calculate these numbers, although the TTD method is uncertain in the Nordic Seas and the Arctic Ocean? The time-dependece of saturation of CFCs and SF6 (Tanhua et al., 2008) as well as the change in saturation of surface ocean $C_T$ that you document here and that were documented in the Arctic Ocean (Terhaar et al., 2020). It would be good the let the reader know here that this pre- industrial $C_T$ has these uncertainties.

*We now provide an estimate of the uncertainty of the GLODAPv2 anthropogenic carbon.*

Line 130: I am a little bit confused here. Before you talked not really about nutrients when you discussed measurements and now, they are here. Did you have nutrient measurements everywhere? What did you do when nutrients measurmentes where not available?

*We used nutrients for the CO2sys calculations. When measurements of phosphate and silicate were not available, their concentrations were set to 1 and 5 umol/kg, respectively, the errors incurred by this approximation are negligible. This is now clarified.*

Lines 134/135: Why do you need this sentence? It is confusing after the sentences above and it does not really add something. I suggest deleting it.

*We think it is important to refer to this work as it has shown that the ratio works well in the Nordic Seas, which validates our choice. We have revised the text to make it less confusing.*

Lines 130-135: I suggest using the mocsy uncertainty propagation for the observations (Orr et al., 2018). That would help to know how reliable trends are.

*We thank the reviewer for this suggestion, we think that this is a good idea, and are now estimating uncertainties with the CO2SYS uncertainty propagation (section 2.2.3).*

Lines 139-142: This reads nicely but fits better into the Introduction. Here in the Methods, it would be helpful to motivate why you chose exactly these regions and not others. What makes them special? So far, I am still more or less in the dark about the special aspects of each regions, although this is a good start.

*A description of the watermasses and the deep water convection is now in the introduction. However, one of the reasons behind the choice of these regions was to get a representation of the different water-masses. We will therefore keep this argument here. Another reason behind the choice of these regions was the data availability, which we now also clarify.*

Line 144-145: If you reduce the Fram Strait, can you still call it the Fram Strait? To me, the Fram Strait is characterized by northward flowing Atlantic water and southward flowing Arctic water. If you chose the region in order to focus on the Atlantic waters and exclude almost 2 thirds of the width (looks like in Fig 1), you can hardly call it Fram Strait, can you? Maybe call it Eastern Fram Strait? Similar for the other regions, why do you choose such small subsamples. The argument of aliasing effects might be a good one, but at the moment it is not convincing to me. I think you have to elaborate a little bit on it.

*The aliasing effect is one argument. As we wrote in the manuscript, we want to prevent including different water masses in the same region which could have large effects on the trends. Further, we wanted to constrain the regions to the areas with regular sampling, and avoid regions that are less sampled. We have elaborated the text on this.*

*We now clarify that it is the Eastern Fram Strait at lines. We have changed Fram Strait to Eastern Fram Strait throughout the manuscript.*

Lines 146-147: Does 200 m correspond the maximum mixed layer? Or does a strong gradient exist between 0-100 m and 100-200 m?

*Summertime MLDs are normally less than 200m, but 200m were chosen anyhow as this is the absolutely lower limit for seasonal changes. We revised the text to make it clear.*

Line 149: How did you define significant trends? Did you look at r 2 or p values? An explanation is necessary.

*We now explain how the significance was calculated in section 2.2.2.*

Line 150: For pH trends and uncertainties, wouldn't it be better to look at [H+]? Otherwise it is not a linear trend and the uncertainties are not linear either? (Fassbender et al., 2020).

*Also reviewer 2 commented on this, and that the results can be different if looking at [H+] instead of pH, because pH shows a relative change. However, in this study the pH-variations are relatively small, meaning that the results won't change much if looking at H+ instead. To verify this we made all plots showing pH change also for H+ (we included one representative plot in the Supplementary, Fig. S17). Also, we do want to stick to pH because we aim to make the paper accessible also for outside of science, such as people dealing with environmental policies, and pH is a more well known quantity.*

Line 151: Didn't you introduce T and S already earlier?

*We only use the abbreviations T and S in the equations, and therefore think that it is better to introduce them just before these.*

Line 171: To be clear, you used only GLODAPv2 pre-industrial for past changes (section 4.3), but you used model output for the driver of these past changes? I think that I am not understanding something here.

*The figures of past changes show model output. We decided to stick to the model data to be consistent throughout the manuscript. The GLODAP preindustrial data is only shown in figure 2. We have now clarified this in the figure captions and in sections 2.1.3 and 2.2.2.*

Line 172: You do not consider changes in nutrients as potential driver, so I take it that they are negligible? Or are they somehow included in the alkalinity?

*No, we do not consider changes in nutrients as a direct driver of pH, due to their relatively low concentrations their impact is small. Nutrients are included in the alkalinity.*

Line 173-175: As explained above, this needs much more introduction. At this point of the manuscript, I still have no idea why this should matter.

*We provide a small introduction to $pCO_2$ in the introduction of the revised manuscript.*

Line 176/177: If possible, use a topic sentence here to tell the reader what this first paragraph is about. And please be quantitative here ("Agrees well" does not tell me anything). Do you mean that it agrees within the uncertainties?

*We have revised the text as suggested by the reviewer. With good agreement, we mean that the mean pH lies within the spatial range (std) of pH of the two different sources.*

Line 179: What does 'it' refer to at the beginning of the sentence?

*It refers to the Nordic Seas surface pH. The sentence has been revised.*

Line 179: Can you compare the two uncertainties? I suppose that you used the entire region simulated in the mode whereas the GLODAPv2 does not cover the entire region (or did you use the gridded version here from Lauvset et al.? In any way, please be more precise. In case that the model and the data product do not cover the same regions, I think you cannot compare the uncertainties as they are a measure of the homogeneity of the sample and not the real uncertainty. As the observations are likely concentrated in some regions, this might reduce the homogeneity and thus the uncertainty. Moreover, how confident are you with respect to the pre-industrial uncertainties given that you did not account for changes in T, S, and A T ?

*The standard deviations are a measure of pH heterogeneity in the region, not uncertainty, we have revised the text to make this clear.. We do not compare the standard deviations, we are just showing that the mean of the two different data sources are within the standard deviations of each other, showing that they are overlapping.*
*The GLODAPv2 pre-industrial estimate is gridded.*

Line 180: I do not buy this sentence. The different pH largely stems from the temperature, doesn't it? Globally, waters are warmer than in the Nordic Seas and thus the global surface average pH is smaller, right?

*Jiang et al., 2019 showed a strong correlation between the sea surface pH and the $pCO_2$ , and suggested the low $pCO_2$ in the high latitude seas to be a result of the $pCO_2$ undersaturation. The $pCO_2$ variations can in turn be an effect of temperature. This is something we discuss in section 3.1*

Line 181: Do you show this somewhere? You could simply illustrate that you are right, if you calculate the average of surface pH values in the observations. Otherwise this is just a speculative guess.

*We did check this. We think that it is enough to write it in text ,and that a figure illustrating this would be excessive.*

Line 182: Can you be more precise than "about 0.05"? Uncertainties would be good and regional differences as well. I also would make a single paragraph about the comparison to the global ocean to make it easier to read. As it is presented now, the comparison to the global ocean makes it hard to grab the information about the Nordic Seas.

*We have corrected this to 0.06. However, unlike the decreases in pH estimated from the observations, the modelled decrease does not have an uncertainty (unless we estimate a trend). This is because we have a full time-and space coverage of the model data, and we know the exact decrease. We go into regional differences later in the manuscript, and will not present it in this section. We have moved the comparison with the global ocean to its own paragraph.*

Line 183: See comment above. As observations and model are not sampled at the same places, I think you compare measurement uncertainties with the spatial heterogeneity (which is large in the Nordic Seas). I do not think that this valid here. Moreover, the trend in the observations seems to be much larger than the simulated trend. It would be helpful to sub-sample the model at the same places as the observations to make a proper comparison.

*The standard deviations of the measurements show an estimate of the spread of the pH over the year and within the Nordic Seas, it is not an uncertainty. We have clarified this in the figure caption. As described before, we do not want to make an exact comparison, we just want to show that the two data-sources are overlapping. We want to include the whole Nordic Seas in the model data in this figure to give a global picture of the acidification of the Nordic Seas, which cannot be obtained from observations.*

Line 184-187: The trends are significantly different. Slightly stronger is misleading when the difference is 29% and more than two times the standard deviation. Could you discuss this please and try to explain why this is the case? You mention the variability at the beginning of the time period, but I only see one measurement at the beginning of the time period (first 5 years). And if I only look at the years from 1994-2020 (excluding the beginning, the trend seems to be even stronger). I think you cannot wave this away by mentioning variability.

*Please note that after the revision, the trend estimates have changed a little as we 1) masked out the Baltic Sea and 2) found that the area went to far south in the script making Fig. 2, compared to the one defined in Fig. 1. The modelled trend and the trend form the observations are not significantly different anymore. Still, we do mention what can give rise to differences between the model and the observational data in the revised manuscript (lines 305-307).*

Line 188/189: Please be quantitative: How much more than 0.4? How much below 7.7? What are the uncertainties? You have the numbers, so just give them here. Please be consistent with the significant digits in earlier reported values, either always give two numbers behind the comma or one.

*We have revised this part according to the suggestions of the reviewer, and now provide quantitative estimates.*

Line 189/190: As written above, the shrinking difference between global pH and high-latitude ocean pH is expected as cold waters can hold less C ant . Please tell the reader here how much closer they get. It would be better to make a paragraph for the global-Nordic Sea comparison.

*As suggested by the reviewer, we have moved the global-Nordic Sea comparison to its own paragraph (lines 318-326),  and now provide quantitative assessment of the changes. It is correct that also the lower temperatures of the Nordic Seas play a role (giving them a lower buffer capacity). We have added this to the discussion.*

Lines 190/191: This hypothesis is very speculative and comes out of nowhere. This is your

results section. If you mention it here, you have to show results that support your claim. Can you somehow use the model output to show what drives the shrinking difference between global and Nordic Sea surface ocean pH?

*We agree that this is a hypothesis and that it is more of a discussion. To make this clear we are now writing " This is most likely...". We think that an analysis of the drivers of the pH change of the global ocean is out of the scope of this paper.*

Lines 193-195: Again, please be exact and give uncertainties. Please also restructure the paragraph. First you mention RCP8.5, then you switch to the concept of reduced undersaturation with possible explanations and then you go back to the projections. It would be more readable if you first present the results for the scenarios and then present other mechanisms. Furthermore, the information that the pH decrease under RCP4.5 is 0.2 and half of RCP8.5 is redundant. Everyone is cabable of comparing 0.2 and 0.4.

*We have restructured the text and provided standard deviations (spatial) next to the numbers.*

Line 195: Can you give a reference that supports that the RCP2.6 is necessary to limit warming to 2°C. I think, depended on the model other scenarios might also be good enough?

*The reference is already in the manuscript, but it was misplaced. We have moved it so that it is after this statement.*

Line 196: What is "slightly" above 8? As mentioned above, please use the same number of significant digits throughout the manuscript. Please also give uncertainties for the end-of century pH under RCP2.6

*We have put spatial standard deviation next to the numbers, and made sure that we use the same amount of significant digits throughout the manuscript.*

Figure 2: I do not know if the global line is needed, it makes it more complicated.

*We decided to keep the global line, we think it is important to show that the pH -dynamics of the Nordic Seas are different.*

Please make the shading transparent so that the other scenarios can also be seen in the first half of the 21 st century. In the label you can delete the phrase "including those outside our regional boxes", it is redundant as you say before that you use all observations in the Nordic Seas.

*We have followed all the suggestions of the reviewer.*

I suppose your observations for every year come from a limited number of cruises that are in a well-defined part of the Nordic Seas and does not represent the entire Nordic Seas. Does it this make sense to make a global 'Nordic Sea' trend? If a cruise takes place north of

Iceland or in the Fram Strait in that year, the pH should be relatively large, but if it takes place close to Norway, it would probably be small as suggested by Figure 3b. The fact that the mapped product (Lauvset et al., 2016) is in good agreement with the model suggest that the stronger trend from 1981-2019 in the observations might just be due to the sampling locations.

*We were also hesitating about making a Nordic Seas trend from the observations due to the reasons presented by the reviewer. However, we think it still can be there, if we mention the caveats of doing so, which we now do on lines 305-307.*

Could you also try to explain why the model uncertainty in 1850 is almost twice as large as the one from the gridded GLODAPv2 product? Is the model heterogeneity too large, or does the gridded product overlook the real heterogeneity because of the extrapolation of a limited number of measurements?

*We found that a few grid-points of the Baltic Sea were included in the area (Fig. 1) when extracting the model data, which caused the larger standard deviation of the model data. After masking the Baltic Sea, the standard deviation of the model approaches that of the GLODAPv2 climatology.*

Overall, I would propose a very different model-data comparison as mentioned above. I suggest sampling the model at the same time and place as the obs. I would then make a scatter plot with the observations of the x axis and the model data on the y axis. You could then plot one point per year and the points should be on one line. If the points are constantly above or below, a constant bias could be seen. Maybe color the dots depending on the year to get the time dimension in the plot.

*This would have been a good idea if the focus of the paper would have been different, i.e. if we would have used model and observational data together to understand a certain process in present climate. But for this, we would also have needed another model. ESMs are adapted for longer simulations spanning over several decades, and not for going into details on shorter time and space scales. The main reason being the relatively coarse spatial resolution, and that the climatic modes of the ESMs are out of phase with the observed ones (unless using data-assimilation).*
*The aim of our paper is to get an understanding of past, present, and future acidification using the best available information for the various time periods. We now clarify this in the introduction. The purpose of figure 2 is mainly to give an overview of the different time periods and data sources. For the application of our manuscript, we think that it is enough to show that the two data-sources are overlapping.*

Section 5.1: I like that you discuss this with the water masses. I would make this even more prominent. Could you for example show the different water masses approximately with lines in Figure 3?

*We have now put a line in figure 3 showing the approximate boundaries between the main water masses.*

How are these water masses defined? This would certainly help to understand changes in each water mass or changes of the extend of water masses. Instead of writing warm waters from the South, I suggest for example writing warm Atlantic waters. Moreover, this section has a lot of qualitative assessments that are not supported by evidence (see comments below). Please go carefully through it and see how you can support all the claims with evidence.

*A definition of the water masses is provided (Atlantic Water has a salinity >34.5, polar waters have a salinity < 34.5). We have also made a thorough revision of this section and are now able to quantify the impact of the various drivers.*

Lines 205/206: You write that C T relates to salinity. I think you want to speak here about freshening from Greenland ice melt. If that is what you are writing about, please make it clear and do not leave the author left in the unknown about the relationship between C T and salinity.

*This section has been almost completely revised, and we do no longer mention the relation between CT and salinity. Instead, we have written that the chemical properties co-vary as a result of the contrasting properties of the water masses in the Nordic Seas.*

Line 206/207: I would present it here as southward flowing Arctic waters along the Greenland coast with high pH and northward flowing Atlantic waters with high pH or something similar. And how do you define quantitively that temperature is the main driver? The alkalinity distribution looks also similar to the pH distribution. If you make a statement about the main driver, can you give evidence?

*We have made a thorough revision of this section and are now able to quantify the impact of the various drivers.*

Lines 206/207: I do not think that A T and C T effects dominate along the Greenland coast. It seems to be rather salinity/freshening. I also suggest deleting that a southeast northwest gradient exist. As you mention just afterwards, that gradient is not stable over the entire Nordic Seas.

*With the new analysis provided in the revised manuscript, these lines have been removed.*

Line 210: This information is needed in the Methods (see comments above). How well do the observations cover the Nordic Seas in space and time? Without this information, it is almost impossible to put the results in context.

*We have moved it to Methods.*

Line 211: Here you speak about contrast of polar waters and Atlantic waters. That is great! I would present the results all around this idea. Polar waters, Atlantic waters, Greenland coastal waters. I think that would make it much easier to understand all these mechanisms and great findings that you present. However, at the moment I do not really now what I would expect as the paragraph above does not really use the words polar waters and Atlantic waters.

*We are now introducing polar and Atlantic waters earlier.*

*Your idea of presenting the results focusing on different watermasses is interesting. As we see it, there are two ways of structuring the paper:*
1. *focusing at specific regions, as we do now*
2. *focusing on different water masses, as suggested by the reviewer. This could be done by constraining the analysis to measurements with specific temperature/salinity.*

*However, if putting the focus on polar, Atlantic and Greenland water masses, we would mainly constrain ourselves to the surface. The depth analysis that we have done would not fit in anymore, unless we look into more water masses. The structure of the paper would change drastically.*

Line 214: Are the two citations the best ones here? Furthermore, the relationship between carbonate ion and temperature holds globally, but in regions with strong salinity and A T gradients, I would not think that it holds well enough. Do you have evidence that the relationship holds in these conditions? I really miss the effect of freshening, which can be large in high-latitude oceans.

*We think that these citations are appropriate. With the new analysis that we present in the revised manuscript, we are able to separate the effect of temperature on the solubility and show that this relationship holds.*

Lines 216-218: This is speculative and no result. I would suggest deleting it.

*The sentence has been deleted.*

Line 222: Please be quantitative, what do you mean by "rather uniform"? Isn't there a gradient from west to east? Especially along the Norwegian coast, the changes seem to be larger. Can you explain this?

*We have changed this to be more quantitative.*

Line 223-224/227-2289: Isn't the change in the saturation states just larger in the Atlantic waters because the saturation states are larger and therefore the same changes in the drivers (C T , A T , T, and S) lead to different changes in the saturation state due to the non-linear scale of the saturation state? Wouldn't it be better to look at [H+]?

*With the new figures in section 4.1 (Figs. 12 and 13), we now can better quantify the reasons behind the changes in pH.*

Line 225: The pCO 2 undersaturation comes out of the blue here. Do you have a figure to show this? Do you show somewhere the larger CO 2 uptake in Atlantic waters? In any way, wouldn't it be expected that Atlantic waters take up more C T given the higher Revelle factor in Arctic waters compared to Atlantic waters? And what is the role of sea ice changes over the historical period? I find this a little bit fast given the complexity of the system.

*We are now discussing the patterns of change in section 4.1. With the new figures, we put less focus on changes in pCO$_2$ undersaturation.*

Line 229: I do not agree that the impact of acidification changes with depth. It is the acidification that changes with depth or the impact of changes in C T on acidification. This is a difference. Furthermore, I would not say "limited connection" but rather refer to the time it takes to ventilate the deeper ocean and the age of the deep waters.

*We realize that the use of the pharising "impact of acidification" is not appropriate here, as it can be understood as ecosystem impacts of OA. What is investigated here is only OA and not the impacts. We have reformulated the text.*

Line 232: Is this change significant when accounting for uncertainties? See Terhaar et al. (2020) as an example of uncertainties related to the ASH.

*We now present uncertainties related to this.*

Figure 3: Please adjust the colorbars. For your work, the Baltic Sea is not important, so you might not need to go down to pH values below 8.05.

*We have gone through all the plots and adjusted the colorbars. For this specific plot, the choice was not related to the Baltic Sea, but it was to include the waters close to the Greenland coast. We have therefore kept the colorbar as it is.*

Along the same lines, it would be helpful to adjust the range of the colorbars of the saturation states. For calcite the shown range seems to be 2-5 while the colorbar goes from 0 to 5.

*After the suggestion of reviewer 2 we have removed the maps of calcite.*

Similarly, you might be able to better show changes (c,f,i) if you chose an adjusted range.
*We have gone through all the plots and optimized the colorbars. The patterns of change look a bit different after redoing the calculations with the modelled changes of temp, sal, alk, DIC, and nutrients to the GLODAPv2 climatology.*
Consider also to mask the Baltic Sea as ESMs are usually not made for these small basins.
*We have masked the Baltic Sea.*

Figure 4: Can you show the section on a map, maybe in figure 3?
*We have put a line representing the section in figures 4,8 and 10..*
As mentioned for figure 3, could you try to adjust the range of the colorbar. The shown colors do not seem to go far
below the 8, but the range of the colorbar goes down to 7.8.
*We have revised the colorbar ranges.*
 Moreover, I find it confusing to
see pH and the saturation horizon on the same plot. Wouldn't it be better to show the filled contours of the saturation state?

*We tried this out, but then we decided to put in lines showing the saturation horizons to reduce the number of plots.*

Section 5.3: In line with the comment above, I suggest changing the term "Present trends". I would furthermore suggest to restructure the section and to explain regions that are similar together. Going from region type to region type might be easier to follow than going from the surface ocean to the deep ocean. I struggled to read this.

*We have changed "Present trends" to "Present day changes (1981-2019)"*
*We have revised/restructured the section, and we think that it reads more easily.*

Line 237: What do you mean by decreasing order?
*We have reformulated the sentence.*

Line 237-238: It is normal that largest decreases are seen at the surface, right? As the increase in atmospheric CO 2 is exponential, the decrease in pH should be larger in years that were recently in contact with the surface.

*Yes it is normal, we have added a sentence on this.*

Line 238: Can you explain what this uncertainty refers to, how it is calculated?

*The uncertainties are standard errors, we explain this in the revised manuscript.*

Line 239: If you only have observations for a short time, does it make sense to calculate a trend over the entire period? You make an extrapolation and assume that the trend holds. I really do not think that this is a good idea.

*The trend is calculated over the period of data. We do not extrapolate it. We want to keep the trend lines over the whole time period to make them more visible.*
*However, we have now removed the 95% prediction intervals showing the possible y-values when using the regression equation obtained in the regression analysis. We realized that it does not provide any useful information for this manuscript.*

Line 243: Can you provide uncertainties for the estimates by Skjelvan et al. (2014)?

*They do not provide any uncertainties of their trends. We realise that we cannot tell anything about statistical difference without this. We have therefore revised the sentence.*

Line 244: If Skjelvan et al. (2014) used different observations (sampling period, different region, and different seasons), can you compare these two estimates? If you provide this as an explanation for the difference, I would like a prove. Could you use the same region as Skelvan et al. (2014) and the same time period (1981-2013) to demonstrate that your hypothesis is right? If not, I do not see how these two estimates are comparable. And if they are comparable, I would like to know why you find a significantly reduced trend in pH. I would be really interested in knowing if a shift exists in the reaction of pH in the Norwegian

basin after 2013. Alternatively, the change in the trend might also be due to the non-linear scale of pH referring to the question if a linear trend in pH (a non-linear unit) makes sense.

*We calculated the trend for the period 1981-2013, which ended up as -3.3+-0.48\*+-10^(-3), which overlaps with their estimate. We will not attempt to use exactly the same region as them as they were using circular regions.*
*Although we cannot make an exact comparison, we still think that we should mention the difference, and give potential reasons to why they differ.*

Line 247: I would not write relatively strong and just give the units.

*We have removed the "relatively strong".*

Lines 253-259: Most of this is material for the Discussion and I suggest moving it there.

*It has been moved to the discussion.*

Lines 269-270: What exactly is "close"? Can you be quantitative?

*We are now providing more quantitative estimates.*

Figure 5/6: Please zoom into each subplot. The way it is represented at the moment, it is hard or impossible to read the data and the uncertainties (see 6t as an example). I see that you want to have the same limits for all depth levels for better comparison but that makes most subplots unreadable. Could you precise how you calculated the uncertainties in the aragonite saturation state and pH? Are they calculated in GLODAPv2? Did you calculate them using uncertainty propagation? Did you account for uncertainties in the equilibrium constants?

*We want to keep the same limits for all subplots to make them comparable. However, after the suggestion of reviewer 2, we have moved the trend estimates to tables.*
*The errorbars show standard deviations of the measurements in each year, which we explain in the figure caption. As explained earlier in this document, the 95% confidence intervals, showing the possible y-values that can be obtained with the equation obtained with the linear regression, have been removed.*
*The uncertainties in pH and saturation state obtained from the error propagation are discussed/showed separately in Table 2 and Figures S6-S7.*

Line 278/279: Why do you present these numbers twice in the Results? I think I have already seen them earlier. Please add the uncertainties.

*The reviewer is correct that this has been presented before, we have therefore removed this sentence. The spatial standard deviations have been added.*

Line 283: What exactly is a "Small region"

*After the revision of the calculations of future pH and Omega, the future projections do not suggest that any region will be undersaturated in OmegaAr. This sentence has therefore been removed.*

Line 284: Please show the changes in pCO $_2$ . You prominently highlight pCO $_2$ in the abstract, so I would like to see changes.

*Acknowledging that changes in $pCO_2$ is not a driver of pH change, but only another indicator, we decided to put less focus on it in the discussion, and in the manuscript in general. To better understand the regional differences in projected pH changes, we instead made maps and cross sections of the different drivers. These are now shown in Section 4.1 (Figs. 12 and 13). Still, we think that it is still interesting to mention that the seasurface $pCO_2$ of the Nordic Seas does not perfectly track the atmospheric one. We therefore put a figure showing this in the supplementary material (Fig. S14).*
*Because of the large uncertainty in the $pCO_2$ trends for the present climate (1981-2019), we decided to remove the sentences on $pCO_2$ from the abstract.*

Line 295: Wasn't the seafloor already undersaturated? And isnt't it the change in the saturation state that leads to undersaturation?

*Only the seafloor below 2000 was undersaturated.. we clarified this by writing "the entire seafloor ". It is correct that we should write the change in saturation state, and not the change in pH. This has been changed.*

Line 295/296: This is speculative. Can you add evidence for this hypothesis?

*We have removed this sentence.*

Figures 7/9: Please adapt the colorbar range according to the colors shown in the plots. I strongly suggest changing the colors around saturation states around 1. Having the same color for saturation states from 0.5 to 1.5 does not clearly indicate where waters change from undersaturated to oversaturated, although this is the most interesting region.

*We have put red-blue colorbars instead of a red-white-blue. We have revised the range of the colorbars to better comply with the range in the plots.*

Figures 8/10: I suggest merging both figures. I would also show the aragonite saturation state instead of pH and I would add uncertainties for the saturation horizon (See comment above). More importantly, how representative is this section for the Nordic Seas. As you show, the acidification trends in deeper waters strongly depend on the regional sea. Thus, the rise in the saturation horizon should also depend on the regional sea. Is this section at the higher end or lower end of Nordic Sea acidification?

*We have merged the figures. We chose this section because it is in the middle of the domain and it nicely shows the impact of deepwater formation in the West, and the Atlantic surface layer in the East. Although the acidification rates may vary slightly, it captures the main physical features that are important for the Nordic Seas.*

Lines 299-300: See comments above about the idea of changes in ocean pH expected from atmospheric CO 2 increase. It would be quite naïve to think that it the ocean pH depends only on the atmospheric CO 2 increase. Although it might be a useful comparison, I would not state it this prominently at the beginning of the Discussion.

*We have removed the first lines of the Discussion.*

Line 301: When did you show something about climate variability? Do you mean variability in the atmospheric climate? I really do not know what you are referring to here.

*We have also removed this part to avoid confusion. It was not necessary for the manuscript.*

Line 307: Can you extend the buffer capacity explanation a little bit more?
*We decided to remove this sentence. We realised that it can confuse the reader because it speaks about inter-basin variations, which we cannot assess with the observational data due to the temporal difference in data-coverage.*

Line 312: In the Fram Strait and the Greenland Sea, the slope of atmospheric CO 2 and marine pCO 2 is almost the same. The difference between atmosphere and ocean is only valid in a subset of the Nordic Seas, right? And if that is true, can you explain these differences?
*After recalculating the expected change in pH after the suggestions of reviewer 2, and also checking the significance, the difference between the change in oceanic and atmospheric $pCO_2$ is only significant in the Norwegian Basin and the Iceland Sea.*
*As described in the manuscript, we will not attempt to compare the different regions, due to the difference in data coverage.*

Lines 314-328: You mention a lot of possible mechanisms and explain them well. This is what I would like to see earlier (in the Introduction). Here, it becomes all more understandable why you insist on the change in undersaturation. You say that any further exploration is beyond the scope of this study. I find this very disappointing. Your manuscript concentrates on this, leads the reader to this point. You show all changes and then stop at explaining them. It would really make the manuscript much stronger if you could somehow at least conceptually find the most important mechanisms in each region.

*We are now briefly mentioning the $pCO_2$ undersaturation in the introduction. As explained earlier, we do not want to go into the same detail in the introduction as in the discussion because that would move the focus away from pH.*
*We think that finding out the reasons behind the change in $pCO_2$ undersaturation goes way beyond the scope of this paper. The reasons can be any, or a combination of, of the mechanisms discussed. To actually pin that down we would need extensive additional analyses and as such this requires a separate manuscript*

Line 330: Could you add some Discussion about the alkalinity behavior in the Barents Sea Opening. In light of the recent study by Asbjørnsen et al. (2020), it seems that Atlantic water extends far more northward and would thus increase alkalinity. It is more a displacement of Arctic Water in the BSO than a change of the existing water mass. Could you somehow

elaborate on this and discuss possible impacts? You somehow waver around it, without really explaining the driving mechanisms.

*We have evolved the discussion on this in the manuscript. The decomposition into a freshwater-driven component and a biogeochemical - driven component (Fig. S19) suggest that this strong increase in alkalinity is not a result of changing salinity (and thus a shift in water masses), but rather of biogeochemical processes or possibly uncertainties related to sampling. We discuss this in Section 4.1 at lines 490-502.*

Lines 337-341: It looks surprising that all the cited studies find a warming of the Nordic Seas while you find a cooling at the surface. You only give a possible explanation without evidence. This is disappointing as differences to other studies are most of the time what helps understanding the underlying mechanisms. Could you somehow test, if the sampling is the reason for the opposite trends? If it is, how robust are your estimates overall? If it is not the sampling, why are your results differen?

*We tested this by plotting all available temperature data, and not only the ones taken at the same time as the DIC and alkalinity samples. When doing so we get a warming trend, and it is therefore clearly an issue of undersampling. We are now discussing this in section 4.1.*

Lines 343-345: Is the increase in temperature in these depths caused by increased temperatures in Atlantic waters that flow northward at these depths towards the Arctic Ocean? Please use the Discussion to explain the mechanisms. What did Osterhus and Gammelsrod (1999) say about it.

*The temperature increase is likely a result of a reduction in the deep-water formation, which has led to an increased presence of Arctic Ocean Deep water . We have added a number of references on this, including the one of Osterhus and Gammelsrod, in section 4.1 .*

Line 346: I think you already mentioned the compensation of $C_T$ and $A_T$ ? Do you need to repeat it here?

*We mentioned it for the surface waters. Here we are discussing the layer below. We have revised the sentence to make this clear.*

Lines 346-349: You point out regions where changes of $C_T$ and $A_T$ cancel each other without mentioning the drivers. I find that unsatisfactory. Once, you show me something interesting, I would like to know why this happens. Can I expect that to continue in the future, or is $A_T$ only temporarily slowing down acidification and once the $A_T$ stops, acidification will be even stronger than before? A lot of open questions that I would like to see discussed.

*We have evolved the discussion around this, in particular for the Barents Sea Opening where the AT increase is strong and significant.*

Lines 349-352: If the $C_T$ signal goes deeper in the Lofoten basin, this looks like stronger deep- water formation in this region. Is this the case? Could you test it in the model or with observations of transient tracers?

*It would indeed be interesting to get a deeper understanding about the role of deep winter mixing in the transfer of ocean acidification to the deep Nordic Seas, and how this contributes to eventual regional differences. However, for this we would rather need a regional model (ESM's rarely reproduce the exact locations of deep water formation). Further, eventual observations of transient tracers would need to be combined with information about horizontal advection in order to understand how the deep water masses formed during convection spread at depth. We believe that such an extensive data analysis is out of the scope of this paper. Instead, we refer to studies that have already been done on this subject.*

*The literature on the Nordic Seas deep water formation and water masses do not indicate that the Lofoten Basin is a region of deep winter mixing. This takes place in the Greenland and Iceland Seas, and these water masses have then been shown to spread in the deeper layers of the Nordic Seas. The discussion about this was already there in the first manuscript, but we have revised the section in an attempt to make it clearer.*

Lines 353-360: Can you explain why alkalinity changes are very important in the present day trends, but in the past and the future C T is by far the most important driver in all regions? I would suspect that it might be due to the model. If I understand everything right, Figure 11 is based on observations, but Figures S9-S11 are based on the model. Could you make Figure 11 only with model results. If the alkalinity contribution is not significant in the model but in the observations, I think that the model misses important processes. In this case, many of the conclusions in this paragraph do not hold (C T being by far the main driver for the past and the future). If the model shows similar results as the observations, however, than it would be really interesting to know what is different in the present-day situation, maybe it is the beginning shift from constant to declining land and sea ice? And if the NorESM-ME model does not show the changes in alkalinity, it would be interesting to look at other models (something I think would be necessary anyway).

*Please note that also for present, CT is the main driver.*

*Indeed, figure 11 is based on observations, while S9-S11 are based on the model (Note that these figures have been replaced by figures 12 and 13 in the revised manuscript).*

*Because of the different time scales considered for the changes between past-present (145 years), 1981-2019 (39 years), and present-future (95 years), a comparison between the different time-periods should not be made.*
*At the timescales considered for past and future changes, the external forcing, in this case climate change, becomes an important driver. When it comes to CT being the most important driver, this is a result of the strong increase in atmospheric $CO_2$, and thus the emission scenario. Here, eventual model deficiencies play a minor role. On the other hand, at the time-scale of our period of observations, climate variability can still play an important role. However, we do still see that CT is the main driver, and that the changes in CT are in good agreement with the change in atmospheric $CO_2$.*

*As explained before, the aim of this manuscript is to assess the acidification in the different time periods, using the best available data, and we do not attempt to compare the different periods per se. An ESM is more adapted for long term studies of climate change stretching over a century, and not short-term studies. On the time scale of 40 years, climate variability can have an important impact on the observed changes, and the climate modes of an ESM are not in phase with the observed ones, which would complicate a comparison between observed and modelled changes.*

Line 360: This is speculative: If it is related to freshwater export from the Arctic Ocean, why would it only be seen in the Iceland Sea.

*There is a recirculation pattern of polar waters in the Iceland Sea. With the new figure 12 in section 4.1 this becomes very clear.*

Lines 363/364: What do you mean by "We relate this ... excess CO 2 "? Please extend your explanation here. I think the undersaturation is just a "normal" part of the Nordic Seas and only in the RCP2.6 it is gone, because the atmospheric CO 2 trend reverses..

*We have reformulated the paragraph and now also provide a figure in the supplementary showing the evolution of the pco2 difference between the ocean and the atmosphere (Figure S14).*

Line 374: This is speculative. You need to show the AOU and try to quantify the effect on C T if you want to make this point.

*We are now providing a short calculation on this on lines 541-546.*

Line 376: You continue to speculate, including changes in deep-water formation. This is really interesting and makes it even more frustrating not to know why this happens. Can you try to elaborate on this? A reader does not want to be kept in suspense.

*With the revision of this section, this phrase has been removed.*

Line 389: Could you quantify this instead of saying most of? It is 321 out of 324, right? So you could just say: Out of 324 reefs in the Nordic Seas, 321 are at depths of 0-500 m for example.

*We have quantified this.*

Lines 393/395: I would not really spend so much time on one out of 324 reefs. It is not that important and somehow takes the wind out of the sails of your really strong message here.

*We have removed the text about the three deepest reefs.*

Line 396: If I understand it right, under RCP2.6 and 4.5 only 3 out of 324 reefs will be exposed to undersaturated waters. That is good news. Can you try to make this clearer as it might be interesting to policy makers. At the moment you say the deepest reefs without saying how many reefs.

*With the new uncertainty analysis the results for the RCP4.5 are not that clear anymore. We have revised the text to make the results more clear.*

Figure 11: I really like the decomposition. Nice plot. However, as mentioned above, I do not think that the stars (expected trend) make sense for waters below the surface. As the deeper waters are not in contact with the atmosphere, they cannot see the accelerated trend in pH (see figure 2).

*We have removed the stars for waters below the surface.*

Conclusion: I find the Conclusion relatively weak in comparison to the findings of your paper. At the beginning it reads more like a summary of your results and it ends with a relatively complicated paragraph about the difference in partial pressure that is very technical. I would like to encourage you to really highlight your main findings and why they are important. You have many interesting messages in this manuscript!

*We now call this section Summary and Conclusions, and we have separated into two subsections. We think that it now reads better, and hope that the reviewer agree.*

Supplementary Figures: Please correct them in accordance with my comments on the figures in the main manuscripts.Technical notes:

*We have corrected them.*

Line 4: I think it should be "from...to" instead of "since...to"

*ok*

Lines 5-6: It is difficult to compare "in the last 40 years" to "between 1850-1980". Could you use the same format, for example "between 1980 and 2020" and "between 1850 and 1980"?

*ok*

Line 13: I suggest to write "is projected to" instead of "will be"

*ok*

Lines 20 and 24/25: "Since 1750" is repeated again, maybe you can look for another formulation?

*we will revise this*

Line 33: returns

*after the revision of the introduction, this is not used anymore*

Line 39: "Projected" instead of "expected"

*ok*

Line 51: "are" instead of "is"

*ok*

Line 51: Maybe write "qualitative effects" instead of "(direction only)"

*ok*

Line 86: Is "visited" the right word?

*We tried to think of another word but could not find a good substitute.*

Line 90: Are all these citations necessary? Can they be somehow grouped?

*They should be there.*

Line 136: I find it hard to understand the word "present trends". I suggest writing the trends over the last 40 years or something similar.

*We have changed this to present day trends (1981-2019).*

Line 184: An uncertainty of 0.00 looks weird.

*This is what it is rounded to two digits.*

Lines 185/186: Please use the same number of significant digits. And is mpH a common unit? I suggest writing 1e -3 pH, but it is your call.

*We have changed mpH to 1e -3 . We have also gone through the manuscript to make sure that we use the same amount of significant digits.*

Line 187: I would start a new paragraph here.

*ok*

Line 212: To be consistent, I would not use the word saturation state but only W.

*We have changed this.*

Lines 247-250: Sometimes you give positive and negative trends. I see how that fits into your writing, but it is confusing. Could you stick to write about decreases and only give positive numbers or speak about trends and then give the negative numbers?

*ok*

Line 275: Close to undersaturation or close to being undersaturated.

*ok*

Line 273-277: I am not sure if this summary of the previous sections is needed.

*We have removed it.*

Line 284: Delete "interestingly".

*ok*

General: I think it is undersaturated with respect to. Could you change this throughout the manuscript please?

*ok*

---

## Author Comment (AC2) · 23 Jun 2021

**Responses to James Orr (Reviewer 2 )**

*We thank James Orr for his suggestions and all the work he has done. The manuscript will greatly improve from this, both scientifically and structurally. Please find below our responses (in blue italics) to the comments (in black). References to line numbers, figures and tables refer to the revised manuscript, unless stated otherwise.*

This manuscript uses observations and a model to assess the regional details of acidification of the Nordic Seas during the industrial era through to the end of this century. The authors find that during 1981-2019, the change in surface ocean pH is larger than would be expected from the corresponding change in atmospheric $CO_2$ . They ascribe the cause to an evolution of surface ocean $pCO_2$ , which while remaining undersaturated with respect to the atmosphere, increases at a rate faster than atmospheric $pCO_2$ . They suggest that the main driver of the change in pH is the DIC increase associated with ocean uptake of anthropogenic $CO_2$ . They also find that observed pH changes may be detected down to 2000 m in some parts of the Norwegian seas. The authors further focus on corresponding changes in the saturation state of waters with respect to aragonite and corresponding changes in the aragonite saturation horizon and what those changes may mean for cold water corals. In their model, most cold water corals would not be exposed to waters that are undersaturated with respect to aragonite under the low-end RCP2.6 and mid-range RCP4.5 emissions scenarios. But under the high-end RCP8.5 emissions scenario, most of those corals would be exposed to such conditions, which are are unfavorable for their long-term survival.

Overall the authors have addressed an important topic, the details of acidification of the Norwegian Seas, a regional focus that has not been addressed previously. They appear to have used all the best data available for this assessment, thanks to the many coauthors with observational expertise in the Norwegian Seas. Also included are coauthors who are experts in using the chosen model routinely to assess ocean acidification and related aspects of ocean biogeochemistry. The Abstract and Introduction (sections 1-3) generally establish the need for this study, the Methods section appears to provide sufficient detail except for the final subsection, and the Results and Discussion sections reveal much effort being devoted to the analysis.
Yet despite these positive aspects, there is also much room for improvement.

CONCERNS in order of importance:
(1) Unfortunately, there seems to be a complete lack of understanding of what a pH change actually means. Although pH offers a convenient way to represent the hydrogen ion concentration, its log scale means a pH change actually represents a relative change in [$H^+$ ], not an absolute change (Kwiatkowski and Orr, 2018). That relative change is unlike the change in any other $CO_2$ system variable, all being absolute. Focusing only on pH and not [$H^+$ ] can give a completely wrong impression, e.g., as in this manuscript when it is used to compare changes at different depths and at different locations (Fassbender et al., 2020). Looking only at pH change, as in the manuscript, we cannot know what part of the change is due to a change in [$H^+$ ] and what part is actually due to differences in the reference [$H^+$ ], the starting point. The manuscript neglects this key point entirely, not even mentioning hydrogen ion concentration. Remedying this problem will require major revisions.

*Thank you for pointing this out, it is a very important remark. Please note that the pH variations in this study are relatively small, and our results do therefore not look significantly different if analyzing $H^+$ instead (we remade all our plots showing pH change to verify this). Because the manuscript is intended to address people outside of science, we prefer staying with pH in the plots in the main manuscript. We believe that this quantity is more well known than $H^+$ concentration. However, we have carefully revised the text in the manuscript to avoid any misinterpretations related to this, and to discuss this issue. The revisions include:*

*i) link between pH and $H^+$ in the introduction (lines 23-24, and section 1.1)*
*ii) A short paragraph on the potential caveats, with the references, that you mentioned (lines 200-204)*
*ii) one representative figure in supplementary material, with accompanying table, (Fig. S17 and Table 7), showing the trends in $H^+$ calculated from the observations.*

(2) Projections with only one model are unreliable. Model projections are hard to pub-lish nowadays without using multiple models and for good reason. One model can give very different results from others. A range of models provides an estimate of model un-certainty, and the model mean typically performs better than any given model. Because the ocean component of the NorESM1-ME model relies on a dynamic isopycnic vertical coordinate, we might expect it to have very different results in simulated deep-ocean anthropogenic carbon concentrations relative to most other CMIP models. Modeling centers such as the one where some of the authors of this manuscript are associated seem to now have access to and experience working with the CMIP5 or CMIP6 models. All analyses in the current manuscript need NOT be repeated with all models. But it will be needed to show at least where the NorESM1-ME model is situated relative to other Earth system models, in terms of the depth distribution of anthropogenic carbon con-centrations (and perhaps also [$H^+$] and $\Omega_{Ar}$) in the different regions of the Norwegian Seas.

*Thank you for this comment, we agree with this. In order to show where the NorESM1-ME model is situated, we additionally made a timeseries plot showing the simulated pH at three different depths of the Nordic Seas for an ensemble of CMIP5 models that we put in the supplementary material and that we mention in the discussion (Fig. S5). Furthermore, we decided to analyze the output of these CMIP5 models for the section on future saturation horizon and cold-water corals (Section 4.2).*
*Unfortunately, the CMIP5 database only contains output for the RCP8.5 when considering emission driven runs (which we used for the manuscript), which is why we could not make a sensitivity analysis for the other scenarios. Considering that NorESM1-ME1 simulates among the higher acidification rates in deep waters in our model-ensemble, it is likely that the estimates of the future saturation horizon from NorESM is in the upper bound.*

(3) The description of the decomposition of the drivers (namely the equations in section 4.4) is weak and incomplete.

a) Eq. (1) comes from Takahashi et al. (1993) and is fine except that the authors

will need to replace the Greek delta δ with the correct partial sign ∂ in all the partial derivatives. This is not a major problem, just the convention of multivariate calculus. The δ is used for something else (inexact differential). Please don't confuse them.'

*Thank you for noticing this, we have changed the delta to the correct partial sign.*

b) Eq. (2) is added by the authors but is unnecessary. That equation comes from Metzl et al. (2010), who expanded each partial derivative in Eq. (1) to get at so-called "known quantities". Such complexity is no longer necessary because all of the partial derivatives in Eq. (1) are now easy available as precise quantities in "derivnum", an add-on package to CO2SYS-MATLAB (Orr et al., 2018). See
https://github.com/jamesorr/CO2SYS-MATLAB
The simpler choice, just deleting Eq. (2), is preferred and avoids unnecessary complexity that can lead to mistakes in implementation. For instance, the authors four definitions that immediately follow Eq. (2) are ambiguous because they are missing key parentheses. Hopefully their actual code is less ambiguous. There is no longer any need to introduce all these extra terms.

*We have followed the recommendations of the reviewer and removed equation 2, we agree that this is not needed, and we instead added some short text on how the derivatives were calculated. The derivnum-package would have been very useful for us, and is something we will implement in future studies. However, for this study we did not use it as the scripts and analyses were written/starting before its release.*

c) Eq. (3) should be recast in the same pattern as Eq. (1), i.e., replacing $fCO_2$ with $[H^+]$. It should not be cast in terms of pH (as in the current manuscript) for reasons mentioned in (1) above. The partial derivatives of $[H^+]$ are also available in derivnum. That routine is called with the same arguments as CO2SYS, with one argument added in the beginning to specify what the user wants to take partial derivatives with respect to. This further move towards simplicity will avoid the old-fashioned complexity that is now in the manuscript. Furthermore, this change will help avoid misinterpretation of what changes in pH mean.

*We have rewritten equation 3 in terms of $d[H^+]/dt$ to prevent any confusions of what pH changes mean. Because we did not use the derivnum as explained above, we will keep the same pattern as it currently has.*

d) An equation is missing in Section 4.4 concerning the freshwater Taylor-series decomposition, results of which are presented in Fig. S8. With that equation, the appropriate citations need to be given, starting with Lovenduski et al. (2007). For the associated salinity normalization, the authors must also specify their choices of the regional salinity references and if those remain constant or change with time. Furthermore, the authors will need to mention why they generally seem to prefer to use the older, less complicated decomposition from Takahashi et al. (1993).

*You are correct that we forgot to add this, we have added a description with the necessary equation and references in section 1 in the Supplementary material. We prefer to stay with*

*the older version because the net impact of freshwater is generally negligible due to the opposing effects on DIC and alkalinity (we explain this at lines 236-239).*

e) Another equation is missing in Section 4.4 concerning what the authors call "pHperf". Currently that term is mentioned in the short final paragraph of section 4.4, where the authors attempt to describe how they compute "the pH change in seawater that perfectly tracks atmospheric $CO_2$ ". Unfortunately, the current description does not tell us exactly what the authors have done. For instance, in the calculation of pHperf, do the authors use i) the actual atmospheric $pCO_2$ as the reference value along with the atmospheric $pCO_2$ change or ii) the oceanic $pCO_2$ as the reference value, to which they add the change in atmospheric $pCO_2$ ? The importance of this question is illustrated with a simple example. Suppose atmospheric $pCO_2$ is at 400 µatm and oceanic $pCO_2$ is at 300 µatm. Although a 1
µatm change in $pCO_2$ starting at 300 µatm produces only a 0.7% greater change in $[H^+]$ when compared to starting at 400 µatm, the corresponding change in pH is 30% greater in the former relative to the latter. The reason is that a change in pH represents a relative change in $[H^+]$, i.e., relative to the $[H^+]$ of the starting point. These numbers slightly depend on the other reference conditions, which I have arbitrarily set to T=2 ◦ C, S=35, ALK=2300 µmol/kg, nutrients=0. If the authors have used approach (i), the re-sults will be wrong. The authors should be able to resolve this issue by using approach (ii) and by adding an equation and improving the text to avoid ambiguities.

*Thank you for the illustrative example showing the importance of this. We used approach (i). We fully agree with the reviewer that approach ii) should be used. We therefore remade the calculations and the plots and we have added some text at lines 240-245 to make the description clearer. Because the calculation is relatively simple, we think that an equation is not necessary.*

A related minor question: Do the authors actually use atmospheric $xCO_2$ (ppm) or do they first convert that to atmospheric pCO2 (µatm), making corrections for water vapor pressure and atmospheric pressure?

*We did not make any corrections for water vapour and atmospheric pressure because the rates of change for $xCO_2$ and $pCO_2$ are the same, this is now clarified.*

(4) The section on cold-water corals is too cursory. The authors' analysis of the change in the aragonite saturation state to which cold-water corals are exposed has potential, but the authors devote only one rather short paragraph to describing the results, which are presented in one figure. They authors also neglect to clearly attribute previous studies that have attempted the same type of exercise using model projections and cold-water coral positions. Additionally, the data set used in the manuscript for coral positions is not cited adequately, and the authors do not give enough information about their procedure for extracting the saturation state from the model. For instance, is the model sampled at the depth of the coral (as provided in the data base) or is the depth taken to be that of the model's bottom depth at a coral's latitude and longitude? More discussion of results and the addition of uncertainties from a multi-model analysis would seem critical.

*We agree that this section was very short. We have revised it by:*

*i) working on the text and including more details about our results*
*ii) adding results from several ESMs*
*iii) adding a few references using a similar approach*

*We additionally provide more details about the dataset and calculations in Section 2.1.4.*

(5) The writing needs improvement. Getting through this manuscript was not easy. Although there are few if any errors in English, and individual sentences generally work well, the manuscript would benefit if the authors could redouble their efforts to improve flow between sentences. That is, connections between sentences are often rough, causing the reader to slow down and sometimes stop. Also lacking is coherence in many individual paragraphs. My recommendation would be for the authors to consult the book by J. M. Williams (Style The Basics of Clarity and Grace), and in particular the short chapter on Cohesion and Coherence. Then they could go through the manuscript trying to improve both aspects. If one cannot borrow this book from a library or colleague, older editions only cost about 10 euros. It offers the potential to dramatically improve one's writing by applying a few basic principles.

*We are grateful for the book suggestion. We have worked on the flow of the manuscript , and think that it now reads better and we hope that you agree.*

(6) The figures need improvements. Some figures appear to have too many panels, some figures should be combined, and some figures should be deleted. There are also other issues.

*The figures have been revised:*

- *Fig. 1: Addition of colorbar and position for time -series stations*
- *Fig. 2: The spatial standard deviation of future scenarios are in transparent*
- *Maps and sections: color ranges have been revised, plots for calcite have been removed*
- *Figs: 6 & 7: the parabolic lines showing the uncertainty range of the regression model has been removed, and the trend-numbers have been moved to tables*
- *The sections showing future changes have been combined to one figure (Fig. 9 in the revised manuscript)*
- *figure 11: the pH expected from the atmospheric change in pCO2 has been removed from deeper layers*
- *figure 14: white space has been removed, and we have added results from several CMIP5 models*

a) In Fig. S6, it seems that only 3 out of the 6 regions seem to show a trend in surface ocean $pCO_2$ that is significantly greater (statistically speaking) than that of atmospheric $pCO_2$ . Thus I am unconvinced by the statement that it is only the Barents Sea Opening does not follow this pattern. More care is needed when handling this subject in the revised manuscript.

*The reviewer is correct, we did not pay enough attention to the significance of the trends and their differences when we were discussing this. After checking the significance, we found that the trend is greater than the atmospheric one in the Norwegian Basin and in the Iceland basin. Based on the comments from reviewer 1 we also made an attempt to estimate the impact of seasonal undersampling on the trends by calculating the trends for the productive season only (April-September, and June-August, respectively). We found these results to be sensitive to which months that are used for the calculation, showing that there are large uncertainties related to the seasonal distribution of the sampling. We therefore decided to put less focus on the $pCO_2$ change (although we still discuss it), and we also removed this from the abstract.*

b) In Figs. 3, 7, and 9, the third row of maps for $\Omega_{Ca}$ should be deleted because it exhibits the same patterns as for $\Omega_{Ar}$ in the second row, only differing by a constant. Their constant relationship could be briefly mentioned once in the text rather than wasting space in each of those three figures. Likewise, Fig. S5 for $\Omega_{Ca}$ should be deleted because it shows exactly the same patterns as $\Omega_{Ar}$ in Fig. 6.

*Thank you for this remark, we have removed the figures showing $\Omega_{Ca}$.*

c) In Figs. 5 and 6, the numbers given in each panel for the slope and uncertainty should be moved to a table, where it will be easier to compare numbers between regions and depth layers. The same goes for the corresponding supplementary figures (Figs S1-S4). In addition, there are often too many significant figures in the slope and uncertainty, and the number of digits is not always consistent. Furthermore, in those same supplementary figures, the slopes have the wrong units. In regards to these C6and other figures, when statistical significance is mentioned in the text, that should be backed up with a statement of how it was determined. Such is not the currently the case in the manuscript, but it is critical, e.g., when discussing if oceanic $pCO_2$ is increasing more rapidly than atmospheric $pCO_2$ (Fig. S6).

*We have adopted the suggestions of the reviewer:*
*i) the numbers for the slope and uncertainty have been moved to tables, both in the manuscript and in the supplementary material*
*ii) we have made sure that the number of significant digits is consistent*
*iii) we have put in the right units*

*A description of how the statistical significance is calculated is now found at lines 225-227.*

d) Figs. 8 and 10 should be combined.

*Done*

e) Fig. 11 includes some details that might need to be deleted, and corresponding supplementary figures should also be refined. What is the rationale for including the dashed line and black stars in subsurface layers? Those layers have been isolated from the atmosphere for some time and we would not expect them to track atmospheric $CO_2$. Showing these details in subsurface layers will confuse the reader. Moreover, would

it not be better to devote a separate figure just to the subject of ocean $pCO_2$ tracking atmospheric $CO_2$ rather than trying to squeeze that information here into a very small space? Fig. S6 fills this need well. That could be brought up into the main paper. Only the top level (0-200 m) of Fig. S6 would need to be shown as we do not expect subsurface levels to track current levels of atmospheric $CO_2$ . I also worry about how representative the 0-200 m layer is of surface ocean $pCO_2$ . Some discussion on that and perhaps a modified figure seems necessary.

*We agree that the dashed line and stars can lead to confusion, and we therefore removed them from the deep layers. After redoing the calculations and calculating the significance, we found that the trend is only significantly greater than the atmospheric one in two of the basins (NB and IS). We additionally tried to do the same for a thinner surface layer (0-50m), then the trend is only significantly greater in the NB.  Initially, we thought that it was a good idea to bring the $pCO_2$ figure into the manuscript, but since we cannot state with any certainty that the trend is larger than that of the atmosphere, we do not think that it would fit in anymore, and would bring the reader away from the focus on pH.*

In corresponding supplementary figures for the model (Figs. S9-S11), the authors miss the opportunity to compare the model results over the same 1981-2019 period as used for the model.
*These figures have been removed in the revised version of the manuscript. Instead we made contour plots (maps and sections) showing the drivers associated with the modelled change, which we put in the main manuscript (Figs. 12 & 13). We think that these are more illustrative.*

*Regarding the suggestion about a comparison between the drivers in the model and observations, this was also suggested by reviewer 1:*

*The aim of this manuscript is to assess the acidification in the different time periods, using the best available data, and we do not attempt to compare the different periods per se. Such a comparison would thus not fill any function in the manuscript, with its current storyline. Further, an ESM is more adapted for long term studies of climate change stretching over a century, and not short-term studies. On the time scale of 40 years, climate variability can have an important impact on the observed changes, and the climate modes of an ESM are not in phase with the observed ones, which would complicate a comparison between observed and modelled changes.*

By the way, why is this time span often referred to in the text as lasting 40 years; actually, it lasts only 39 years.
*correct, we have corrected it in the manuscript*
My impression is that relative to the observations, the model is dominated even more by the change in DIC, based on the analogous plots for the previous and subsequent time periods. These supplementary figures concern the model, but readers will be confused because 'OBS' is used to designate the model result, both in the caption and in the figure itself. Please change 'OBS' to 'MOD'.
*These plots have been replaced by Figures 12 and 13 in the new manuscript.*

f) Fig. 12 has too much white space.

*We edited the figure to adjust for this.*

g) The supplementary figures are mentioned out of order.
*This has been corrected for.*

---

## Author Comment (AC3) · 23 Jun 2021

**Responses to reviewer 3**

*We are grateful to reviewer 3 for all the work done. The comments are identifying weak points of the manuscript that will help us to improve it. Please find below our responses (in blue italics) to the comments (in black). References to line numbers, figures and tables refer to the revised manuscript, unless stated otherwise.*

This is an interesting and ambitious manuscript with important goals. I commend the authors on their substantial efforts in synthesising all the varied data streams (real and modelled) to put together the trend analyses, projections, and regional maps. While these synthesis figures don't deliver revolutionary new insight in a purely academic sense, they are extremely important and highly sought after in more policy-oriented applications. These results are certainly worthy of eventual publication.

I write this having also read the two existing peer reviews of this manuscript. I agree with the concerns of the other reviewers that many results are presented with either no or insufficient quantification, and/or too vague or incomplete conceptual explanation. This is my main concern about the manuscript as it is.

This review is so brief because there are not many points left to make without simply repeating the thorough work of the other reviewers. Other than the main concern noted above, I have only a few minor additions:

Abstract: is "window to the deep ocean" the right metaphor here? A place through which the deep ocean can be observed – is that the intended meaning?

*With this phrase we wanted to refer to the strong connection between surface and deep waters through deep water formation. The introductory sentence of the abstract has been revised.*

Abstract: sensitivity to OA in the Nordic Seas is not directly due to high latitude, but rather due to low water temperature?

*You are correct, the two introductory sentences of the abstract have been replaced with the following:"With prevailing low temperatures, deep winter mixing, and cold-water coral reefs, the Nordic Seas is vulnerable to ocean acidification." An explanation on the impact of low temperatures is given later in section 1.1.*

Sections 1 and 2, and probably also 3, are very much introductory material and I would also suggest to consider combining them, as mentioned by another reviewer.

*It is a good idea to do some merging of these sections. After some thinking, we ended up dividing the introduction into two sections; one general introduction on ocean acidification and the Nordic Seas, and one on "theoretical background".*

In Section 2 and Table 1, an important aspect of discussion is absent, that is about

the timescale of the effects shown in Table 1. Are you showing instantaneous effects of T/S/DIC/TA increases, or effects after $CO_2$ re-equilibration with a constant atmosphere? Looks like it's the former – is that really appropriate, given the context?

*We agree that this is an important aspect that should be discussed. Table 1 shows the instantaneous effects without any gas exchange, we now make this clear in the label and when we refer to it. We think that it is important to show this rather than the effect after $CO_2$-re-equilibrium because the Nordic Seas surface waters are not equilibrated, and therefore somewhere in between the instantaneous and secondary effect. We now discuss this also in Section 3.1.*

Methods: given the relatively low temperature of your observations, why not use the Sulpis et al. (2020) carbonic acid constant parameterisation for your CO 2 system calculations?

*The Sulpis et al (2020) paper was published in late July 2020, when all our analyses had been done, which explains why those parameterizations were not used. This could indeed be important for our estimates of the aragonite saturation horizon. But, looking at their figure 8, we see that the differences in the aragonite saturation state they get with the Sulphis parameterization and the Lueker parameterization, is about the same size as our uncertainty estimates (Table 2). These uncertainty estimates become even larger for past and future when taking into account the mapping error. This indicates that the results will not change substantially if switching between the two parameterizations. The Sulphis parameterization is however something that we will consider in our future work.*

Throughout: pH is dimensionless; pH values do not need the word "units" after them, and $\times 10^{-3}$ can be used in place of "mpH".

*We have removed the "units" and replaced mpH with $\times 10^{-3}$ .*

Line 205: "DIC also relates to salinity" is very vague, please explain the mechanism–including timescale considerations noted above for Section 2 / Table 1. See e.g. Wu et al. (2019).

*This section has been strongly revised in the new version of the manuscript, and we now also discuss the importance of timescales. The phrase "DIC also relates to salinity" is no longer in use.*

Line 424 "both" implies two options when there are three (past, present and future). I am not sure that the chain of causality is properly represented in this and the subsequent sentences (i.e. which are drivers and which are responses in terms of air-sea CO 2 disequilibrium, pH change and hydrographic conditions), please be careful with the exact phrasing here.

*In the revised manuscript we have put less focus on changes in $pCO_2$, and we have removed this paragraph. It is, indeed, not a driver of pH change, but just another indicator of a changing carbonate-system.*

The request for more research at the very end of the manuscript is very unspecific and is unexpected given that the rest of the paragraph implies that all the observed phenomena have indeed been explained here.

*We have removed this sentence.*

---

## Author Response (AR2)

Dear Dr. Gattuso,

We are happy to submit a revised version of our manuscript "Acidification of the Nordic Seas". The manuscript has undergone major revisions, and it has improved much from the last version.

Major changes include:
- We have included plots of H+ concentration changes in the supplementary material.
- We have assessed the sensitivity of our results to the choice between pH and H+ as the main variable in subsection 4.3.
- We now discuss the implications of using emission driven runs.
- We have clarified that, and why, we work with a 200 m upper layer when we analyze the observational data.
- We have worked on the introduction and the conclusions to enlighten what new our study brings to the scientific community, and how it relates to other studies.
-  Please note that we have slightly changed the title after the suggestion of reviewer 3.
- We have revised the language/structure of the text to make it flow better.
- Please note that we found a minor bug in the code for creating Figure 14. When correcting this the residual became smaller compared to the last version of the manuscript.
- We slightly changed the region for which we calculate the aragonite saturation horizon depths in figure 12. In the last version it was for the entire Nordic Seas. Because cold-water corals are constrained to Atlantic Water, we excluded polar waters in the updated version of the manuscript by excluding areas west of 0 °E and 64 °N. The effect on our results are minor.
- We have included the freshwater decomposition of pH drivers in the main-manuscript.

Please find our responses (in blue) to all the reviewers comments (in black) at the following pages.

We want to thank you and the three reviewers for all the time you put into this manuscript.

Best regards,
Filippa Fransner and co-authors

REFEREE #1:

First, a huge 'Thank you' to the authors for this review. The invested work is immense and must have taken a lot of time. The improvement is clearly visible. Great job. It was a pleasure to re-read it. I have a few important general comments and then just a list of minor comment, which are rather suggestions. If you do not agree with the minor suggestions, you can keep it the way you wrote it. My only 'major' point is the choice of the surface layer (first 200 m). If that point is addressed, I am looking forward to seeing this great work published.

We want to thank referee 1 for going into such depth and detail also for this second revision, and putting so much time into reviewing the manuscript. The comments have been very helpful and they lead to a great improvement of the manuscript.

We have addressed the comments as described below.

General comments

1) Using the word "present": The use of the word present for the period from 1981 to 2019 is still confusing to me. For example, the abstract states that pH has dropped by 0.06 from pre-industrial to present. It sounds as if present is now (2021), In the next sentence you use the period between 1981 to 2019 and say that pH has decreased by 0.10. How can the decrease be larger in only a small part of the period? Logically this means that pH has increased by 0.04 between 1850 and 1981. When you refer to until "present" you refer to the period form 1850-1981. This is highly misleading in the abstract and has to be explained here. You wrote this very clear in the Summary, maybe make it similar in the abstract. However, I would suggest not using the word present for the time from 1981 to 2019 but rather call it the period of regular ocean observations. Something like: "From pre-industrial to the beginning of regular ocean observations (1850-1981), pH is estimated to have dropped by 0.06 on average… From 1981 to 2019, when observations of the ocean are available, pH has dropped by another 0.10…". In Fig. 4 present times refers to the period from 1996-2005, again different.

We agree with the reviewer that the use of the word "present" was unclear in the abstract, which has been thoroughly revised in the latest version of the manuscript. We have also gone through the manuscript to clarify this where needed. Additionally, In many places we have replaced present with present-day, which by definition refers to a longer period of time. This is to avoid that present being interpreted as "today" or "this year".

2) It is often written "uptake of anthropogenic CO2". However, Anderson and Olsen (2002) suggest that the anthropogenic CO2 is taken up by the ocean South of Nordic Seas and mostly advected laterally into the Nordic Seas, where some of it is even lost to the atmosphere, similar to the Arctic Ocean (Terhaar et al., 2019) and the Barents Sea in particular (Terhaar et al. 2020b). Maybe just rephrase it carefully to "mainly driven by an increase in anthropogenic carbon".

Thank you for this remark. We agree that it is better to write "increase in anthropogenic carbon", instead of "uptake". We have corrected this.

3) The difference between total, natural, and anthropogenic CO2 is not always clear to me. It becomes particularly confusing when opposing the abstract and the second paragraph of the Introduction, where you write that the Nordic Seas are a sink of CO2. However, this uptake does not enhance acidification if it is part of the natural equilibrium, because the Nordic Seas have always taken up CO2 because they are colder and transported that water away. It hence does not affect acidification (which by definition is a change of the natural state). Could you try to explain this precisely?

We have revised this part of the introduction (lines 45-51), and also added some text about the advective supply of anthropogenic CO2. We prefer not to introduce "natural" and "total" CO2, as we do not use these later on in the manuscript. We think that could introduce some confusion.

4) Why do you use the Mauna Loa data and not the global data (tab next to Mauna Loa)? Mauna Loa is around 1 ppm higher than global numbers and thus increases the difference between the ocean and atmosphere of total CO2 in the Nordic Seas.

In our manuscript we are using the Mauna Loa record to determine the annual growth rate in atmospheric CO2, This is not different across the global set of stations (https://gml.noaa.gov/ccgg/about/global_means.html). The absolute value, which indeed is different, does not play an important role in our calculations. We clarified this on lines 148-150.

Further, while the Mauna Loa record is based on direct measurements, the global estimate is a result of several steps of data processing. In addition, new sites have been added to the global record over the years. Using the Manual Loa record is therefore more straightforward.

5) I would still suggest writing the Methods mostly in passive tense as the book mentioned by James Orr suggests.

We have rewritten it to mostly passive tense. In some cases, especially where we explain certain choices/assumptions that we have made, we keep an active tense, as appropriate.

6) The surface layer of 200m seems to be far too deep to me, especially in fresh Arctic waters. The argument that primary production influences the first 200m does hold either in my opinion as different processes act at the surface (production) and below the surface (remineralization). Especially as the authors mentioned several times in the answers that long-term changes are the main point of the manuscript, the argument of the seasonal cycle is somehow empty. Averaging over 200 m can make it difficult to develop a process understanding. In case the 200 m should remain, I would at least like to see if things change drastically by choosing 50 m and 100 m to understand how sensitive are your results to the choice of the depth? I am especially thinking about section 3.2. You say that past changes are strongest in the Atlantic water and less strong in the Arctic waters. Maybe that is just because Arctic waters are much more stratified and by using 200 m depth, the depths from 100-200 m are much less affected. However, that may not be the case at the surface.

For the analysis of the discrete measurements, we made the choice to have a 200 m thick upper layer to minimize the number of seasonally affected layers, i.e., layers where seasonal

undersampling could have an effect on our results. If we would have used 0-100 m and then 100-500 then both would be seasonally affected (since the winter mixed layer goes down to approximately 200 m) with more uncertain trends. The point is that we prefer to have only one layer where the trends need to be interpreted with more caution due to seasonal undersampling.

We have added this explanation on lines 228-231.

For the model data, on the other hand , we want to show surface data so that it can easily be compared to other (future) modelling studies where it is standard to show the surface fields.

One may then ask the question: the aim of the manuscript is to put the observed changes into perspective to climate change. Is such a comparison valid if using the 200 upper meters for the observations, and only the surface for the model?
- The main comparison between the observational and model data is provided in figure 4, where we also use the 200 upper meters of the model. We refer to this when we put the different time periods in context to each other.
- The surface maps are then used to get an understanding of regional differences (which we do not touch upon when analyzing the observational data due to the different data coverages in the various regions)
- Furthermore, the past and future drops in pH are very close when comparing the mean changes over the 200 upper meters, and the changes in the surface, indicating that the upper 200 m are broadly representative of surface. Very shallow surface layers, as in the Arctic proper, hardly occur in the Nordic Seas, here, winter mixing is usually deeper than 200 m, except for in the East Greenland Current

We have added some text on this on lines 278-281, 380,  and 420, to make this clear.

Additionally, we realized that we have introduced some confusion by using "surface waters" both when discussing changes related to the discrete measurements and changes related to model projections. In sections 5.3 and 5.5, the results show the actual surface (i.e. 0m), while for the discrete measurements  (Section 5.4) it is the upper 200m (Please note that in figure 4 the model data have been averaged over the upper 200 m). To avoid this confusion, we now use "upper layer" when we talk about the 0-200m layer, and "surface" when we refer to 0m.

7) Please check the labeling of tables and figures. Sometimes numbers are missing or the S for supplement is missing.

Thank you for noticing this. We have corrected the incorrect labels.

8) Overall, a lot of discussion is already happening in the results section. It would be great to make a clear cut between discussion and results although I understand that it is difficult.

We have merged the sections "Results" and "Discussion" to one "Results and Discussion".

9) Did you subtract a piControl run to detrend your historical and future simulations? Especially in the deep ocean, the opposing trends in CT and AT maybe just a drift?

We checked if there is any drift in pH the pre-industrial control run to verify that the model is sufficiently spinned up. The change in pH in the pre-industrial control run is very small (more than one order of magnitude less) compared to the changes in the historical run and in the future projections, showing that model drift has a negligible impact on our results. We added some text on this on lines 173-176.

Minor comments

1) Line 2: This first sentence reads strange. The Nordic Seas are vulnerable to ocean acidification (1) at the surface because they already have low saturation states because of low temperatures and (2) below the surface because they have deep winter mixing and hence transport CT to the deeper ocean quickly in comparison to other parts of the ocean. The presence of cold-water coral reefs makes them not more vulnerable but gives one example of the impact of ocean acidification. Krill in the Southern Ocean or sea butterflies are also in danger because of ocean acidification. I suggest rewriting the first sentence.

We have rewritten the sentence after the suggestion of the reviewer.

2) Line 9: Suggest changing "relatively deep" to "below the surface".

We now write "In some regions, the pH decrease ca be detected down to 2000m"

3) Line 10: Is the significant decrease everywhere or on average or in subregions?

We do not anymore mention "significant" in the abstract, and only mention the range of observed rates over the different subregions.

4) Line 15: Suggest adding "all" before cold-water corals.

done

5) Line 22: Please add globally somewhere here. As your paper is about the Nordic Seas, the reader may think that you are already presenting local results.

done

6) Line 22: The decrease of 0.1 is until which year? 2020? 1981?

We now write from pre-industrial to present days. Because we write "approximately 0.1", we think that we do not need to be more precise.

7) Line 23: Suggest deleting "this" as there is only one ocean acidification.

We rephrased this sentence following the reviewer's suggestion (see below).

8) Lines 23-26: Suggest rephrasing this sentence: It is not the pH drop that causes the reduction in CaCO3. Maybe write: "Furthermore, the increasing CT also causes a reduction

in CaCO3 and hence poses a serious threat to marine organisms that have shells and skeletons consisting of CaCO3, such as pteropods and corals". Thus, you have old information (CT increase) before new (reduced CaCO3) and before the consequences (organisms).

Thank you for this suggestion, we used it in the text.

9) Line 37: All surface waters or only Arctic or Atlantic ones?

We rephrased to: "The surface water pCO 2 is generally lower than that of the atmosphere, making the Nordic Seas important sinks for atmospheric CO 2 ."

10) Line 40: Are the references for all three processes or can you add one reference for the primary production, one for cooling and one for the Arctic Ocean waters?

In some of the papers several processes are discussed, so we prefer to put all references at the end of the sentence. We noted that we had made one mistake and put Olsen et al., 2008 instead of Anderson and Olsen, 2002. We have corrected this.

11) Line 41: Do you have a reference for the deep-water formation in these regions?

We added references.

12) Line 47-49: Suggest moving this up in the paragraph as it is a general characteristic. You could write it as follows: The Nordic seas are particularly vulnerable due to low saturation states because of their low temperatures. Then, you continue with the Cant increase that worsens everything and you explain the increase in Cant with the circulation. I think that flows better.

Thank you for the suggestion, we have revised this paragraph.

13) Line 52: Acidification rate is not precise here as it includes saturation states and pH. Please say what this refers to.

Done

14) Line 66: It does not increase it, it creates it in the first place, right?

We have reformulated this part to avoid any confusion related to the choice of words.

15) Line 93: More than only two types of CaCO3 exist in seawater. These two are the most abundant ones.

We have re-written the sentence to mention that these two types are the most abundant ones.

16) Line 95: Not all CaCO3 dissolves. Maybe write: "When Omega of a CaCO3 mineral is less than one, the water is corrosive, and that mineral starts to dissolve"

Done

17) Lines 123/124: Is there a reason why you give the uncertainties sometimes in absolute numbers and sometimes in relative numbers?

Yes, this is what is given in the Olsen et al., 2019 paper as the consistency of the GLODAPv2 data. The values are based on the pre-defined minimum adjustment limits used in the secondary quality control routines and are therefore relative for those variables for which a potential bias is expected to vary with the concentration of the property measured and absolute for those variables where a potential bias is expected to remain constant over the range of concentrations encountered in the ocean.

18) Figure 1: Consider changing the colormap as it now suggest a hard cut at 2000 m.

We now use a blue-only colormap.

19) Line 172: Why is this behind the model data and not part of the observational data or at least behind it?

Thank you for this observation, it makes more sense to have it after the subsection on observational data. We moved it there.

20) Line 175: Suggest deleting: "It is important to mention that"

We removed this phase as there are certainly several regions with poor data coverage, and not only this one.

21) Line 184: Do you know the pre-industrial $CO_2$ level of the model?

We are now providing the modelled $CO_2$ concentrations in Table S1 in the supplementary material.

22) Line 207: How much is relatively small?

After the request of James Orr we now put figures showing change in H+ in the supplementary material. This sentence has been removed. We also included a new subsection (4.3) with figure 2 where we discuss the choice between H+ and pH.

23) Line 212: Can you just say why you used exactly these values?

We used these values because they are reasonable representative values of the Nordic Seas silicate and phosphate concentration (averaged over the whole dataset). We now describe it in the manuscript.

24) Line 239: Please precise what you mean by 'it' and please add a number to the equation.

Done

25) Line 276: Maybe consider subscripts that are understandable without having to look it up instead of 1 and 2.

We changed subscript 1 to 'point' (for discrete data) and 2 to 'field' (for mapped fields).

26) Line 290: Can you just briefly motivate the choice of these two periods please.

These periods were selected to include the productive season. We chose the longer period to include the spring bloom and the summer production, and additionally the summer period only because this is the time of lowest nutrients/DIC values. We added some text on this.

27) Line 300: Can you state why you think that they are of minor importance?

Here we put a reference to section 4.1, where this is discussed/shown.

28) Figure 2: In the legend you write present (1981-2019), but the tick in the figure suggest that you are showing 1980-2019. And it there a reason why you are showing the model data and not the change to 2002?

Thank you for spotting this error. We remade the figure so that the subplots exactly fits the time periods.
We show the model data because we want to show that the model performs reasonably well in the surface layer

29) Line 303: Suggest replacing "Before going into regional details of pH changes," by "First,"

We now write "Here we give an overview of the upper layer, taken to be the upper 200m for both model and observations, pH changes"

30) Line 303: Global or Nordic Seas?

We added "the Nordic Seas" to the sentence.

31) Line 306: These numbers are now calculated by adding the change to 2002 or it is still the absolute model output?

It is the modelled pH output, we added  a sentence to explain this.

32) Line 308: I think it is rather for a few locations than a few years. It is likely one cruise and if it is at the end of the range of pH it does not agree with the average.

We agree that this is probably the case. We removed this part. We think that the text on lines 392-395 describe this well.

33) Lines 310-312: Your samples at the beginning of the time period seem to be biased towards high pH regions. That could explain the difference in the trend. If you show the trend for the last 20 years, you will probably get a much better agreement. Maybe worth to think about mentioning that instead of writing that the data can be questioned. It is great data, no reason to make it look bad.

It was not our intention, indeed, to question the quality of the data, but rather their representativeness for the whole Nordic Seas. We concur with your example, and added it to the text.

34) Line 313: Suggest deleting "as expected"

Done

35) Line 315: Suggest addition "CMIP5" in front of model ensemble range.

This part has been revised. We now talk about inter-model spread instead of model ensemble range. We think that adding CMIP5 in front of "inter-model spread" would disturb the flow of the text. It is enough that the reader gets this information from section 3.3.

36) Lines 323-331: This paragraph is about the difference between Nordic Seas and global ocean pH. Starting it with 1850 gives the impression to the reader that we are now looking into the past but that is not the main point. I suggest starting the paragraph with what it is mainly about. That improves the flow. Just moving 'in 1850' to the end of the sentence may do the job.

This is a good point, we did as suggested.

37) Lines 333-334: How did you chose the salinity? Wekerle et al. (2016) have a regional definition? You show Arctic waters at the Norwegian coast. That makes not much sense… Did you include these waters as Arctic waters into the scatter plot?

The definition of polar waters /Atlantic Water varies slightly in the literature. We added two references using this specific definition (Malmberg and Desert 1999, and Nondal et al., 2009). It is right that when using a purely salinity-based definition the Norwegian coastal water are classified as polar waters. However, we prefer to keep this grouping of the water masses because the effect of the spatial varying salinity is the largest in the low salinity waters, including the Norwegian coastal water (compare figure 3 d and e, and 3g and h, in the revised manuscript). In the revised manuscript we refer to the watermasses with a salinity <34.5  as low-saline waters. We clarify that the low saline waters include the polar waters that are found in the northwestern part of the domain, and the Norwegian coastal waters that are confined to the Norwegian coast.

38) Line 339: This seems to be contra-intuitive. First you state cold water holds more CT and then you say polar waters hold less? It is because of low pCO2 after a long time after sea ice or because of the low alkalinity?

Both the low pCO2 and low alkalinity (salinity) results in a low CT. We have added a sentence to clarify this.

39) Line 342: Suggest replacing nonphysical by another word. If it happens in the real world, there must be a mechanism.

This paragraph has been removed because we think that it did not add anything to the manuscript. We think that figure 3 and Table 3 in the current manuscript is enough to illustrate the role of the different drivers in the spatial variability of pH.

40) Line 355: Do pH and pCO2 not always correlate, not only in the Nordic Seas?

This is true, we removed this part, we think that it does not add anything to the paper.

41) Line 365: Bring the salinity earlier when you describe the results and not after the discussion

Done

42) Lines 346-366: This paragraph is very verbose. Is it possible to condense the information?

This is very true, we removed quite a large part of the paragraph.

43) Line 371: Can you just explain why they reinforce each other?

We removed this part, we think that it did not give any added value.

44) Line 376: How can it be more important somewhere if everywhere all other contributions already explain 100%?

This is a result of the rounding, the salinity adds less than 1% to the explanation of the variability. We added a note on this.

45) Line 387: Why do you not just use the original data for the calculation? Thus, you do not need to make this statement.

In this section we prefer to use the depths obtained from interpolation in the figure, to be consistent with the figure. The original data (GLODAPv2) has a much coarser resolution.

46) Line 391: This should be figure 6, I think.

Yes, we had made a mistake with the labeling. This is now corrected.

47) Figure 6: Why are you still not using the whole range on the y-axis? Now, I cannot see the data well enough.

We want to keep the same ranges on the y-axes in all subplots to make it easier for the reader to compare the panels and to put the data in the different subplots in relation to each other.

48) Line 392: Maybe the Barents Sea trend isnot there because the water is too stratified? A surface trend may exist while waters below 50 or 100 m remain unaffected? The same happens probably in the Norwegian basin (lines 394-395)

Winter mixing is deeper that 200 m in both of these regions, so we do not expect that the upper 50 m trend should be different from the 50-200 m trend (we did verify this by calculating the trends also for the upper 50 m).
Regarding the comparison with Skjelvan et al., (2014), they also used the upper 200 m for the surface layer. We clarified this in the manuscript.

49) Line 393: between ±0.2 and ±0.8

Correct, we changed this.

50) Line 429: Maybe add that the largest decline in Atlantic waters is mostly due to the effect that polar waters are already close to zero and the decline in units of saturation state are becoming smaller and smaller.

We introduced a line where we say that the reasons behind this is discussed in Section 5.7.2. The reason to the larger decline of Aragonite saturation state in the Atlantic waters in esmRCP8.5 compared to esmRCP2.6 is that in esmRCP8.5 the warming is more uniform over the Nordic Seas compared to esmRCP2.6, where it mainly occurs in the Atlantic waters. The warming has a positive effect on the aragonite saturation, meaning that in esmRCP2.6 the spatially different warming reinforces the effect of the spatially varying CT increase.

51) Line 434: This shows that 200 m may be too deep.

Please see our discussion under your major point above.

52) Line 442: Different in which way?

We clarified this (lines 520-524).

53) Lines 500-507: Could the sampling also cause the trend in alkalinity as it did for pH above?

This is probably not the case as we also see the strong trend in the 200-500m layer. This is discussed at line 557-563 .

54) Lines 511-515: In addition to Shu et al. (2018), you should also cite Woosley et al (2020) for the effect of freshening on CT uptake (line 513) and Terhaar et al. (2021) for the effect of freshening on ocean acidification in polar waters (line 515, line 523, line 526).

Thank you for these references, we added them.

55) Figure S5: Could you show the pH change with respect to GLODAPv2 in 2002 for each model? That would allow to clearly see how large the trends are in comparison?

We have added a table (S6) showing the mean pH change in three depth layers obtained from the GLODAP-climatology in combination with the modelled DIC and AT change in the different ESMs.

56) Lines 600/601: Can you compare two different periods? The atm. CO2 increase is exponential, so one would expect the trend since 1991 to be even stronger. On the other hand, the Nordic Sea trend may be biased high because of sampling biases in high pH regions at the beginning of the observational period.

We removed the comparison with the global data.

References

Anderson, L. G., and Olsen, A., Air–sea flux of anthropogenic carbon dioxide in the North Atlantic, Geophys. Res. Lett., 29( 17), 1835, doi:10.1029/2002GL014820, 2002.

Terhaar, J., Orr, J. C., Gehlen, M., Ethé, C., and Bopp, L.: Model constraints on the anthropogenic carbon budget of the Arctic Ocean, Biogeosciences, 16, 2343–2367, https://doi.org/10.5194/bg-16-2343-2019, 2019.

Terhaar, J., Torres, O., Bourgeois, T., and Kwiatkowski, L.: Arctic Ocean acidification over the 21st century co-driven by anthropogenic carbon increases and freshening in the CMIP6 model ensemble, Biogeosciences, 18, 2221–2240, https://doi.org/10.5194/bg-18-2221-2021, 2021.

Wekerle, C., Wang, Q., Danilov, S., Schourup-Kristensen, V., von Appen, W.-J., and Jung, T. (2017), Atlantic Water in the Nordic Seas: Locally eddy-permitting ocean simulation in a global setup, J. Geophys. Res. Oceans, 122, 914– 940, doi:10.1002/2016JC012121.

Woosley, R.J. and Millero, F.J. (2020), Freshening of the western Arctic negates anthropogenic carbon uptake potential. Limnol Oceanogr, 65: 1834-1846. https://doi.org/10.1002/lno.11421
* * *
REFEREE #2

Overall the reviewers have addressed my comments on the previous version well, here I have just a few extra minor issues that should be taken care of before publication.

Line numbers refer to the manuscript without tracked changes.

We are grateful to reviewer 2 for re-reading the manuscript and identifying these issues. We have addressed all of them as described below.

**Abstract**

Line 2: consider 'Due to' instead of 'With...', because the latter doesn't imply causality.

Done

Line 2: Nordic Seas 'are', not 'is'.

Done

Line 3: and 'their' impact, not 'its' impact.

We decided to remove this part of the sentence to obtain a better flow.

Line 6: 'meter' => 'm' and add a space between '2000 m'.

Done

Line 6: consider 'well below' => 'still deeper than', for clarity.

This part of the abstract has been removed to obtain a better flow.

Line 7: add apostrophe after 'the Nordic Seas'

Also this part has been removed.

Line 8: by 'significant', do you mean 'statistically significant' or 'considerable/important'?

We mean statistically. We no longer mention significant in the abstract.

Line 9: 'until the year of 2100' => 'by the year 2100'.

This part has been removed.

Line 10: 'which are close' => 'which is close'.

Done

Line 12: 'undersaturated in aragonite' => 'undersaturated with respect to aragonite'.

Done

Line 13: remove apostrophe after Nordic Seas -or- add 'the' before Nordic Seas
Done

**Introduction**

Line 20: 'of which about 20% HAS been taken up'

Done

There are too many of these sort of language errors for me to continue to point them out, but the whole manuscript needs careful copy-editing.

We have carefully gone through the manuscript to identify language errors.

Eq. (4): this is the equation for Free scale pH, but you actually work with Total scale pH in the rest of the study. Same comment applies on line 239 for the dpH term.

We revised equation 4 to show pH on total scale. For the drivers we consider the contribution of sulphate to be negligible and do not include it in the equations. We added a sentence on this.

Line 83: AT is the difference between the sum of proton acceptors (weak bases) and the sum of proton donors (weak acids), not just the former.

The reviewer is correct. For simplicity, we now write that AT is mostly determined by bicarbonate and carbonate.

Line 93: CaCO3 might be virtually all aragonite and calcite, but there are other forms in seawater too.

We revised this line to write that they are the two most abundant forms.

**Methods**

Line 193: does CT/AT literally mean CT divided by AT or is it shorthand for something else?

We do not refer to the ratio, only that we include both CT and AT. We have clarified this by writing CT+AT.

Line 274: "including a correlation term would decrease the uncertainty". This statement is not true, or at least not certain. It depends on the sign of the uncertainty correlation term: if

positive, then the uncertainty in e.g. pCO2 would be reduced, but if negative it would be increased. For other variables, with different patterns of variability in CT-AT space, the opposite could apply.

The statement has been revised as following: "The correlation between uncertainties in AT, CT were set to 0. This is a reasonable assumption given that CT and AT are measured on different instruments using different analytical methodologies. In addition, including a positive correlation term would decrease the overall uncertainty and we prefer a potential overestimation"

**Discussion**

Lines 469, 471: missing figure number (11?)

There was a bug in our latex document that caused this, this has now been fixed.

**Results**

Line 323: still a rogue "units" after a pH value here.

we removed the remaining "units"

Figure 3: subplots (c) onwards should be equal aspect and might benefit from explicitly drawing the 1:1 line.

Thank you for this comment. We remade the scatter plots so that they have an equal aspect ratio. It makes the figure much better!

**Conclusion**

Line 628: final sentence does not make sense.

The conclusions has undergone a major revision, and this sentence has been removed.
* * *
REFEREE #3:

In my review of the original manuscript, I raised a number of concerns, which the authors have tried to address in their revised manuscript along with their response to the reviewers. They have dedicated much effort to this task, which is commendable. However, I still have some non-trivial concerns with the revised manuscript.

We are grateful to James Orr for this thorough review, and all the time he put into the manuscript.  The comments are very helpful and we think that working through them has led to a much improved manuscript.

We have addressed the comments as described below.

TWO MAJOR CONCERNS

In my original review, my concerns were listed in order of importance. My first concern was that the original manuscript did not consider potential misinterpretation of pH changes, which because of the log scale depend on the initial state of [H+] not just the change in [H+]. The authors response was to (1) mention H+ in the Introduction, (2) add a 4-line paragraph mentioning this concern, and (3) add Figure S17 and Table 7, which show trends in [H+]. In their response they also say "the pH variations in this study are relatively small, and our results do therefore not look significantly different if analyzing H+ instead". They also state that they "remade all our plots showing pH change to verify this", although they only show Figure S17. But in some of the pH maps in the revised manuscript, there are differences in pH across the Nordic Seas of about 0.1 for the preindustrial references state (Figure 4) and about 0.15 for the present-day reference state (Figure 8). These differences appear small, but they imply 30% to 40% differences in terms of the initial value of [H+]. These regional differences in the initial state of surface pH are much larger than the regional differences in the pH change between preindustrial and present (0.02, Fig. 4) and between present and future under RCP2.6 (0.04, Fig. 8); they are comparable to those between present and future under RCP8.5 (0.15, Fig. 10). Therefore differences in the pH reference state (preindustrial or present) shown in the maps of Figs. 4, 8, and 10 will have a substantial effect on the computed pH change. The authors have taken my previous comment too lightly. I asked for major revisions, but they did not comply. I did not ask that they not show pH in the revised manuscript, something they seem to imply in their response. This concern could be properly addressed if the authors would show changes in both H+ and pH on the same set of plots and if they would quantify how much of the regional differences in the pH change are due to differences in the initial state of [H+] and how much is due to the change in [H+].

We understand the concern of James Orr. To support our choice of working with pH only, we have done the following changes in the manuscript:

1) We added figure 2. and section 4.3 where the issue is discussed. In this figure we have plotted the change in pH vs the change in H+ concentration over a range of initial pH values, for six different increases in CT. The figure shows that for the initial pH values that are found in the Nordic Seas (in present climate), the relationship between the pH change and H+ change is approximately linear for the CT increases that we have in this study. This shows that the choice between H+ and pH does not have an important effect on the results in our study. The linear relationship breaks down for larger changes in CT, and/or of the initial pH spans lower pH values. In these cases it is more appropriate to present the results in H+.
2) We added maps for H+ in the Supplementary material (Figs. S17, S19 and S20), showing the H+ concentrations for the various periods of time, together with the change in the H+ concentration, to support the conclusions drawn from figure 2.

My second concern expressed was that the authors were only using 1 model to make projections. In response, the revised manuscript now compares results from multiple CMIP5 Earth system models (ESMs) for some of the analyses. To make this comparison, the authors rely results from those models under emissions-driven, not concentration-driven scenarios. Unfortunately, the emission-driven scenarios mentioned by the authors are not part of the core set of CMIP5 experiments (Taylor et al., 2012). The concentration-driven scenarios were designed for CMIP and IPCC to be the standard way of comparing models because that way the atmospheric CO2 forcing, both radiative and geochemical, remains identical between models. Otherwise, with an emissions-driven scenario the choice of a terrestrial carbon-cycle component of an ESM, may lead to quite unrealistic atmospheric CO2 levels and thus have an undue influence on ocean carbon. That is why the concentration-driven scenarios have been used in the most cited previous studies that compare CMIP5 as well as CMIP6 Earth system models (Bopp et al. 2013, Kwiatkowski et al., 2020). Yes, an emission-driven scenario can be used IN ADDITION to a concentration driven scenario to study carbon-cycle feedbacks, but that is not the focus of the authors in this study. Therefore the authors comparison of the CMIP5 ESMs using emission-driven scenarios alone is poor experimental design.

We thank James Orr for his comment. We agree that when combined, the emission-driven and concentration-driven experiments can be used to quantify the carbon cycle-climate feedback. Nevertheless, the emissions driven runs (at least historical and RCP8.5) are actually listed as CMIP5 core experiments in Taylor et al. (2012, Fig. 2). They are further described as "suitable either for comparison with observations or provide projections." Below we further elaborate our motivation for using the emission-driven simulation.

The authors further confuse the subject by calling their emission-driven scenarios "RCPs" (Relative Concentration Pathways). Yes, CMIP5 provided emissions for the emission-driven scenarios, such as "esmrcp85", the same set of emissions used to drive the chosen integrated assessment model to come up with the common concentration pathway used by all ESMs under RCP8.5. But the authors cannot refer to their simulations as those from relative concentration pathways (RCPs). The best way to remedy this problem would be for the authors to compare results from the ESMs forced under the actual relative concentration pathways (experiments rcp26, rcp45, rcp85), as has previous work, instead of the emission-driven scenarios (esmrcp26, esmrcp45, esmrcp85).

**Before we argue for the use of emission driven runs, we need to give some background on the choices we have done while working on this manuscript:**

From the beginning, the primary focus of our study has been to use historical observational data sets to synthesize acidification patterns and their drivers over the past 39 years in the Nordic Seas. The use of the historical run and the future projections from an Earth system model (ESM) is to put these observed changes and its drivers into the climate-change perspective. Initially, we chose to work with one ESM only to facilitate a deeper analysis of regional changes and their drivers in several future scenarios (i.e. to avoid a too long paper). We decided to employ the Norwegian Earth system model due to our in-house expertise with this model, and because we have all required output variables readily accessible. At the start of the analyses of this paper, we had both concentration- and emission- driven runs available. For the emission-driven runs, some model improvements had been made compared to the concentration driven ones, including i) an extended spin-up by a few hundred years and ii) a replacement of the local mass-conservation correction related to surface freshwater fluxes with a global mass correction. The latter leads to improved representations of surface tracer concentration and trends in the deep ocean. Indeed, in an analysis of the output of both runs demonstrated that, the emission-driven NorESM runs revealed a better representation of the carbon system parameters against observations, and therefore is more credible in its projections.

In the first review-round, the reviewers pointed out that we should show how our NorESM simulations perform in comparison to other ESMs, not necessarily in all figures in the paper, but at least where we assess the impact on deep water corals, and maybe also in a supplementary figure. We fully agreed with this, and added simulations from seven emission-driven ESMs to our paper. We chose emission-driven runs for the model ensemble to maintain consistency between our utilised model run and the other ESMs.

**We agree that the atmospheric CO2 forcing can vary between emission-driven ESMs. However, we disagree that using emission driven runs is poor experimental design for the following reasons:**

Emission-driven simulations are run with a fully coupled, interactive carbon cycle, and therefore include the carbon cycle feedback on the physical climate. Compared to concentration-driven simulations, the emission-driven ones therefore have a more realistic representation of interactions that take place within the climate system, under a given $CO_2$ emission scenario. We also acknowledge that for this reason, inter-model spread in emission-driven runs is larger than in concentration driven runs (e.g. Booth et al., 2013, Friedlingstein et al., 2014). The choice of multi-model emission-driven ensemble therefore includes a more comprehensive estimate of the effect of model-related uncertainties on climate projections, which is the main point here and which we therefore prefer.

Indeed, concentration-driven runs are useful to identify the sources of the inter-model spread in an individual model component as the carbon cycle climate feedback is excluded. This is what is done in many model-intercomparison studies prepared for CMIP and IPCC, where the scope partly is to understand the sources of the model-differences. But, the aim of our

study is not to understand and attribute the source of inter-model spread in the projected acidification, i.e. it was never designed to be a model-intercomparison study. Here, we use model simulations primarily to (i) project the rates of future acidification and assess potential impacts on cold-water corals under different scenarios (e.g., high- vs low-$CO_2$) and (ii) elucidate the mechanism governing these rates. Nevertheless, we agreed in the first review round to add results from other models as suggested by the reviewer "… to show at least where the NorESM1-ME model is situated relative to other Earth system models." We agree that it is useful to give the reader an idea of model-related uncertainties in the climate projections.

We would like to add that, despite many model-intercomparison studies prepared for CMIP and IPCC use concentration-driven runs, there are several biogeochemistry model-intercomparisons that do not focus on carbon-cycle feedbacks but still employ emission-driven runs that have been published (e.g., Zhao et al., 2014, Kessler and Tjiputra, 2016; Wang et al., 2016; Oschlies et al., 2017).

We are confident that our preference of emission-driven runs over concentration-driven runs is valid, and have decided to stay with this in the manuscript. However, since the reviewer has raised this key issue, we have decided to clarify and discuss the projection uncertainties associated with emission-driven runs, and we have therefore done the following changes in the revised manuscript:

1. We have replaced all mentions of RCP with esmRCP when talking about the emission-driven runs
2. We have estimated the impact of the modelled deviation in atmospheric CO2 in our emission driven NorESM-runs, from the concentration driven ones, by adding Table 1.
3. We added pH-timeseries from concentration-driven ESM's in figure S5.
4. We discuss the implications of emission vs concentration driven runs at lines 169-174, and 172-182 in the section about models, and on lines 386 and 405 in section 5.2.

**References:**

- Kessler, A. and Tjiputra, J.: The Southern Ocean as a constraint to reduce uncertainty in future ocean carbon sinks, Earth Syst. Dynam., 7, 295–312, https://doi.org/10.5194/esd-7-295-2016, 2016.
- Zhao, F. and Zeng, N.: Continued increase in atmospheric $CO_2$ seasonal amplitude in the 21st century projected by the CMIP5 Earth system models, Earth Syst. Dynam., 5, 423–439, https://doi.org/10.5194/esd-5-423-2014, 2014.

- Wang, L., Huang, J., Luo, Y. *et al.* Narrowing the spread in CMIP5 model projections of air-sea $CO_2$ fluxes. *Sci Rep* **6,** 37548 (2016). https://doi.org/10.1038/srep37548
- Oschlies Andreas, Duteil Olaf, Getzlaff Julia, Koeve Wolfgang, Landolfi Angela and Schmidtko Sunke, 2017: Patterns of deoxygenation: sensitivity to natural and anthropogenic drivers, Phil. Trans. R. Soc. A., 375.
- Booth, B. B. B., Bernie, D., McNeall, D., Hawkins, E., Caesar, J., Boulton, C., Friedlingstein, P., and Sexton, D. M. H.: Scenario and modelling uncertainty in global mean temperature change derived from emission-driven global climate models, Earth Syst. Dynam., 4, 95–108, https://doi.org/10.5194/esd-4-95-2013, 2013.
- Friedlingstein, P., Meinshausen, M., Arora, V. K., Jones, C. D., Anav, A., Liddicoat, S. K., & Knutti, R. (2014). Uncertainties in CMIP5 Climate Projections due to Carbon Cycle Feedbacks, *Journal of Climate*, *27*(2), 511-526. https://journals.ametsoc.org/view/journals/clim/27/2/jcli-d-12-00579.1.xml

Overall, it appears to me that this manuscript still would still require substantial revision, and thus I cannot recommend it for publication at this time.

OTHER CONCERNS (of moderate or minor importance, language suggestions)

Title: The title "Nordic Seas Acidification" does not quite work because "Nordic Seas acidifcation". For the same reason, we cannot say "Oceans Acidfication". Thus I suggest to change the title to "Acidification of the Nordic Seas".

Thank you for this remark. We have done as suggested.

ABSTRACT

L2: change "Nordic Seas is" to "Nordic Seas are"

Done

L3-5: The second sentence is hard to understand because of too many clauses and commas. It should be rewritten. A solution that could work would be simply to delete ", and its impact on cold-water corals,". Moreover, this paper does not investigate impacts on cold-water corals, only the aragonite saturation state of the waters that corals may be exposed to. In addition, cold-water corals are mentioned later in the abstract so there is no need to try to squeeze that in here.

We followed the suggestion to remove "and its impact on cold-water corals".

L6-7: The authors make contradictory statements about impacts of the shoaling ASH on cold-water corals (compare L6-7 to L14-15).

The abstract has been revised, these contradictory statements are no longer there.

L8: "significant" should not be used here (or elsewhere) unless it is replaced by "statistically significant" and the authors tell us with respect to what it is statistically significant.

We have rewritten the abstract so that we no longer use the word significant.

L10: same problem as L8

Solved as described above.

L14: change "to be lifted to" to "to shoal by"

The abstract has undergone a major revision, and we no longer mention the actual depths of the saturation horizon obtained with the model projections.

L16-17: delete the qualitative ending ", which to some extent is opposed by increasing alkalinity".

This qualitative statement is no longer in the abstract.

INTRODUCTION

The objectives at end of Introduction (just before subsection 1.1) should be refined. Is the justification for this study just that the authors want to look at rates of acidification in both models and observations? Would they expect those rates to be so different from previous estimates. If so, why would they differ substantially. Unfortunately, the authors fail to return to this so-called gap in the Discussion, i.e., by comparing their results to the previous estimates of Skogen et al. (2014, 2018).

Thank you for this remark. The justification of this study is to get an overview of the pH changes, and their drivers, from pre-industrial to 2100 for various scenarios, and to get a more in-depth understanding of how this varies within the Nordic Seas and over depth, by compiling different kinds of data-sources.

We revised the last part of the introduction by:
- describing more in detail the work of Skogen et al., 2014 and 2018
- refining the objectives to make it more clear what new this study brings.

We have also revised the conclusions and the discussion by putting our work in context to previous work.

L34: change "its" to "their"

Done

L35: It would be clearer if the authors changed "characterized" by "dominated"

Done

L39: change "comes as a result of" to "results from"

Done

L42: change "and consequently help" to ", helping"

Done

L44-45: Confusing. I think the authors should replace "would ultimately lead to early and relatively large detection" with "leads to higher".

We agree that this was confusing. We decided to remove the link to the acidification in this phrasee, and put it later.

L46: change the verbose "have negative impacts on" to "degrade"

We put the link to the cold-water corals in a separate phrase.

L52. "Acidification rates" is ambiguous. All readers will not get that you mean pH change. Please refer to pH change explicitly.

Done

SUBSECTION 1.1
- You cannot have a subsection 1.1 if you do not have a subsection 1.2

We made it to a separate section.

- This section seems to be a repetition of well-known $CO_2$ system chemistry in the ocean. It is largely textbook material.
Because it is not novel, please just to delete it and cite references where readers can go if they are unfamiliar with seawater $CO_2$ chemistry.

We do think that this section makes an important part of the manuscript because we refer to it several times later on in the manuscript. We therefore decided to keep it.

- Equation (4): I think that if the authors are going to actually give equations with [H+] or pH, they should also mention that in seawater oceanographers have different pH scales, while also pointing out the pH scale that they have adopted for this study (presumably the total scale).

We revised equation 4 to show pH on total scale.

L115: Is it a Norwegian or EU program. Please state.

It is Norwegian, we now state this in the manuscript.

L123-124: What does "considered consistent" mean? The last part of the sentence could be shortened to "4 umol kg-1 for both Ct and At"

Olsen et al.2016 mean consistent among cruises. We added this to the sentence. We revised the last part of the sentence as suggested.

L139: shorten last part of sentence as suggested just above.

Done

Section 2.1.2: see 2nd major comment

Please see our answer under the 2nd major comment.

Section 2.2.2
* * *
L204:
- change the verbose "It is important to keep in mind that" to "Because"

We have removed this paragraph. We now discuss the issue in section 4.3

- delete "and that"

See answer above.

L207: Why say "are relatively small". Cannot the authors be more quantitative?

See answer above.

Present
* * *
L209-210: Why mention temperature twice in the same sentence?

Thank you for this observation. This has been revised.

L217-218: suggest to change " (they are" with ", being" and to remove the final ")".

Done

L222: comma fault: remove the comma

Done

L227: change "generally is" to "are generally"

This part has been revised, and we do no longer mention the summer mixed layer depth.

L235 (Equation (7)): The authors need to explicitly state how they calculate the partial derivatives.

Done

L239: what is meant by "it"?

We refer to H+ in equation 10 (8 in the last version of the manuscript). This has been clarified.

L248-249: The rates of change of atmospheric xCO2 (ppm) and atmospheric pCO2 (uatm) will not be identical unless the correction factor between them is 1 (e.g., atmospheric pressure of 1 atm and no humidity). In the cold high latitudes the atmospheric pressures are substantially less than 1 on average. The authors could say though that the rates of change of those two variables are proportional.

Thank you for pointing this out, we adopted your suggestion.

Section 2.2.3
* * *
L270: in the "MATLAB version of" CO2SYS?

Added

L280-281: While systematic uncertainties would tend to cancel out when calculating differences, random uncertainties would not.

Thank you for this, we adopted the suggestion.

L296: Delete "We note that"

Done

RESULTS

line 315:
- suggest to change "model ensemble range" to "CMIP5 model range". The word ensemble as used by modelers is ambiguous (multiple simulations with the same model or multiple models).

We now write "inter-model spread".

- The CMIP5 model range in final pH is actually quite large. It would be much tighter if the authors would have compared results from the concentration driven pathways rather than the emission driven pathways.

Please see our answer to your second major concern above.

line 324: Delete "Note that". How much lower?
Done. It is about 0.1 lower, we added this.

line 325: change "the one" to "that"

Done

line 329: The reference to Dai et al. (2019, Nature Comm.) confuses the issue. Consistent with previous studies, Dai et al. define Arctic amplification in terms of surface air temperature, not surface ocean temperature. In the Arctic there has been little warming of surface sea surface temperature because of ice cover; conversely, there has been enhanced warming of surface air temperature, i.e., about twice the global average. But what about the Nordic Seas that are largely ice free even during winter? I would expect the authors to focus on the Nordic Seas, given the title of the paper. Is the historical SST change there higher than the global average?

We do see a faster warming in the Nordic Seas in the esmRCP26 and esmRCP45 scenarios, but not in the esmRCP8.5 nor in the historical (we realize that this was not described in detail enough in the manuscript). It is a good point that the simulated faster warming does not have to be related to Arctic amplification. It could also be related to differences in other physical characteristics (for example stratification) and how it changes with climate.
We realize that this subject is too complex to go into in this paper, and we therefore remove the text related to the faster warming. Instead we now write:

"This is partially a result of the colder waters of the Nordic Seas, which gives them a lower buffer capacity. Additionally, in esmRCP8.5, there is an increase in the $pCO_2$ undersaturation of the global ocean that increases the global average pH (Fig. S16). Other factors driving this decreasing pH difference between the global ocean and the Nordic Seas can be differential heating. A quantitative assessment of the drivers is beyond the scope of this paper."

Lines 327-330: To make this statement, the authors would need to demonstrate that the SST change in the Nordic Seas is actually higher than the global average. Or it could be documented with appropriate references.

Please see our answer to the previous comment.

line 334: delete "its"

Done

line 337: It would be simpler and clearer if the authors could write "from southeast to northwest".

Done

line 337-338: I do not see evidence in Figure S15 for the authors statement here that CT increases with decreasing T in the Atlantic water. Furthermore, CT is a conservative property with respect to changes in temperature, so the authors would need to be clearer about the processes that make surface CT increase with temperature.

We remade the figure with more contour levels to make the gradients clearer. We agree that the increasing CT with decreasing temperature is not very clear, and we therefore now write "Within the Atlantic water there is a *tendency* of increasing CT with decreasing temperature" We added a sentence about the underlying process.

lines 338-339: Please explain why the CT of polar waters is lower than that of Atlantic waters?

Done

lines 341-343: I have some problems with this statement:
* The statement that "pH decreases with increasing temperature, salinity, CT and AT (... Fig. S16)" is not clear in Fig S16. For Atlantic waters there seems little correlation, while for polar waters, there are vastly different relationships

From the feedback we got from the reviewers on this part, we understood that the text on lines 340-345 in the last version of the manuscript adds little value to the manuscript, and we decided to remove it. We moved table 3 to the supplementary material. We think that the text related to Table 4 and figure 3  (Table 3 and Figure 3 in the revised manuscript) demonstrates what we want to show here.

* The use of "strong" and "significant" in this sentence does not seem to be supported by Fig S16.

This text has been removed as described above.

* The correlations of pH vs salinity, pH vs CT, and pH vs AT look extremely similar, suggesting that the latter two are also driven by changes in salinity.

It does not have to be a direct effect of salinity, but can be a result of the contrasting properties of the polar waters and the Atlantic Water. This text has been removed as described above.

lines 343-345:

\* The authors seem to infer that if the drivers were perfectly orthogonal it would be possible to calculate the contributions of each driver from such correlation plots (Fig. S16). But even in this ideal case, I do not think that is possible.

This text has been removed as described above.

\* Is not the conclusion at the end of the sentence something we could not already say before this analysis? I think the authors could do better than make this weak statement.

We agree that the analysis on lines 340-345 does not add much to the story, and we therefore removed it. We think that the text related to Table 4 and figure 3 demonstrates what we want to show here.

Line 349:
- change "to the picture" to "contributions"

Done

- the parenthetical statement "(temperature, CT and AT explain all together 89%)" is redundant. It should be deleted.

Done

Line 361: remove the comma

This sentence has been removed after the suggestion of reviewer 1 to shorten the paragraph.

line 362: change "relation" to "relationship"

Also this sentence has been removed. Reviewer 1 pointed out that pH and pCO2 always correlate. We therefore thought that it does not bring anything new by going into these details.

line 371: delete "are counteracting, they"

This part has been removed to shorten the paragraph.

line 372: "solubility of OmegaAr"? Do the authors mean "solubility product"?

Yes, this has been corrected.

line 390: What is "Fig. 4.1"? The same problem is found in the title to Table 5.

Thank you for spotting this. It was related to a problem with the labelling in our latex-file. This has been fixed.

line 413:
* "this period of time"? Please be more specific.

We followed the suggestion and have now specified the time.

* "we detect trends in the uncertainties". Some detail is needed.

We added some numbers on the trends.

line 442: The authors refer to "The strong positive trend in pCO2", but it seems they actually mean to refer to Delta pCO2. Unclear.

Thank you for spotting this, we have clarified it.

line 469 and 471: missing Figure number

This was also related to the problem with the labelling in our latex-file. It has been fixed.

line 497: "salinity decomposition" is ambiguous. I would recommend to change that to ""more detailed freshwater decomposition".

We have put the freshwater decomposition figure in the main manuscript. This sentence has been removed.

lines 498-500: I disagree with the authors statement here that seems to say that the uncertainties of freshwater component compromise the assessment of the so-called biogeochemical component. The uncertainties concerning the freshwater component may be large but the freshwater contributions of AT and CT counterbalance one another, making their combined contribution negligible. The uncertainties will typically affect freshwater contributions from AT and CT in equal an opposite ways and thus they don't really matter for the BGC component.

Thank you for pointing this out. We have removed these lines.

line 516: What does "reduces/amplify" mean? Please use the word or words you mean instead of the slash, which is ambiguous.

Here we referred to the effect of increasing alkalinity and salinity. We agree that it was not clear. The paragraph has been revised to make it clearer.

line 535: change "4.5" to "RCP4.5"

It has been changed to esmRCP4.5.

line 549: It would be better to begin the sentence with "Assuming a Redfield ratio of O2:C = 132:106,"

We adopted the suggestion.

line 553: The errors in the way the authors approximate the partial derivatives (sensitivities) are not mentioned, but are probably as large or larger than the other factors mentioned.

Thank you for this remark, we added as one of the possible reasons behind the residuals.

line 589: ambiguous. What does upper bound mean? shallower or deeper?

We meant shallower, we edited the text to make this clear.

SUMMARY AND CONCLUSIONS

It is not clear in the Discussion nor in the Conclusions how the authors have filled the gap they pointed out in the Introduction, namely the advances made by this study relative to the previous work by Skogan et al. Indeed the findings of Skogan et al. were never really mentioned, so we don't have anything to compare this work to. The authors need to bring this up in the Conclusions. To what extent do the findings of this study confirm or contradict previous work. To what extent do they embark on completely new territory. How has our knowledge been incremented by this study. The Conclusions section mostly just reiterates what was said in previous sections and ends with a caveat, a detail about a minor part of the work. The big picture seems to be missing.

We agree that this is an important part that is lacking in the manuscript. We worked on this by revising the last two paragraphs of the introduction and the conclusions. Specifically, in the conclusions we now put our results into perspective to previous work.

line 611: What is meant by "careful estimates"? Do the authors mean "conservative estimates"? If so, I am surprised. Since NorESM1-ME tends to simulate an ASH that is too shallow, does that not imply a bias in the opposite direction, i.e., corals will be projected to be exposed too soon to undersaturated waters. This it seems not a conservative estimate but rather the opposite.

We agree that the formulation of this sentence was not very good. We reformulated it as follows:

Because NorESM1-ME tends to simulate relatively strong drops in pH and shallow saturation horizons in comparison to our ESM-ensemble for esmRCP8.5, our estimated aragonite saturation horizons for esmRCP2.6 and esmRCP4.5 can be considered to be in the shallow bound of possible future states.

line 614-615: The authors state "The effects of increasing CT is slightly opposed by increasing AT , which partly comes as a result of the increasing salinities". But I find this statement puzzling. Based on their freshwater Taylor expansion (Fig. S19), the change in salinity has an equal and opposite effect on CT and AT. These effects cancel out. The same figure shows that it is the changes in the biogeochemical AT term that partly

counterbalances the changes in the biogeochemical CT term. Salinity changes have nothing to do with that.

This is correct, thank you for pointing this out! . It cannot come as a result of increasing salinities, and must be a result of biogeochemical processes. We have revised the sentence.

line 616: I am confused by this phrase: "the impact of temperature in the surface is ambiguous, and even shows a cooling in some places". What is meant by "ambiguous"? Could that phrase be replaced by something simpler such as "at the surface, there is cooling in some places."

We now write that there is no clear temperature change in the upper 200m.

line 629: change the first "to" to "too"

Done

FIGURES

Figure 10: change "rates of" to "the". To the GLODAP estimate for the present, the authors added the change, not the rates of change.

Done

Fig. 11: I think that this figure and the simple Taylor expansion should NOT be included in this paper. It offers no added value and only makes the paper longer. Only the freshwater Taylor expansion should be shown and discussed. Yes the latter is slightly more complex, but it is better because it is able to show that the combined contribution of salinity-driven changes in AT and CT terms is negligible for pH. Focusing on the simple Taylor expansion will confuse readers (see my comment just above concerning lines 614-615). Thus Fig. 11 should be replaced with Fig. S19, and only on the freshwater Taylor expansion should be shown. There is no reason to show the simple Taylor expansion if the freshwater Taylor expansion is presented.

We agree that the freshwater decomposition gives additional information that is worth taking up in the main manuscript. Although the freshwater component is negligible for the pH changes, this decomposition helps us understand what lies behind the changes in CT and AT, and in particular it enlightens the regional differences in CT and AT changes obtained from the model simulations. We have therefore made the following changes in the manuscript:

- we replaced figure 11 with S19 to show the freshwater decomposition, as suggested
- we added one figure showing the decomposition of the modelled surface changes in CT and AT into their respective freshwater and biogeochemical parts. We thought that adding this in figure 12 would result in too many subplots. Because the freshwater forcing mainly is at the surface we did not produce a similar figure for the cross-sections.

- We moved equations 1-4 in the supplementary material to the main manuscript.

Thank you for taking up this point!

Fig. S14: What do the colors mean. There is no information about color codes in the caption nor the legend.

We added a description in the caption.

Fig. S16: in the legend, please change "single grid" to "one grid cell".

Done

TABLES

Table 4: delete the 2nd header line that is in the middle of the table

Done. We also simplified the table by just having one header-row for the "Drivers".

Tables in the Supplement should include an S before the number, but some do not do so

Done

REFERENCES

Taylor, K. E., Stouffer, R. J., & Meehl, G. A. (2012). An overview of CMIP5 and the experiment design. Bulletin of the American Meteorological Society, 93(4), 485-498.

---

## Author Response (AR3)

Dear Dr. Jean-Pierre Gattuso,

We are happy to submit a revised version of our manuscript "Acidification of the Nordic Seas". We want to thank you and reviewer 4 for this revision round, which has helped clarifying some unclear points of the manuscript related to the emission-driven runs. We have addressed all the comments of the reviewer as described on the following pages.

Sincerely,
Filippa Fransner and co-authors

I was asked by the editor to comment on the respective merits of emission- versus concentration-driven simulations to address the specific objective to document acidification of the Nordic Seas. Here are my comments, which also address additional points.

We are grateful for the reviewer's work and suggestions that identify weak and unclear points in the manuscript! We have addressed all the points as described below.

First, this study mainly uses model simulations from one single model (i.e., NorESM1-ME). I think this needs to be stated upfront in the abstract. As currently written, the abstract gives the impression that more than one model was used, but that is not the case.

Thank you for this comment! We fully agree with this and have clarified it in the abstract.

Second, the authors have decided to use emission-driven runs and not concentration-driven runs from the NorESM1-ME model. Both emission-driven and concentration-driven simulations have been used in the literature to analyze changes in the Earth system. For ocean acidification studies, however, concentration-driven simulations are usually used especially for studies analyzing past changes (although not exclusively). In my opinion, it is ok using emission driven simulations when analyzing future changes in acidity in the Nordic Seas. However, the choice of using emission driven simulations is a bit problematic when analyzing past changes (i.e., over the historical period). The atmospheric $CO_2$ over the past 170 years is well known but the simulated atmospheric $CO_2$ over the historical period in these emission driven runs deviates from the observed one (in this study by 14 ppm in year 2005 according to Table S1), and therefore also the historical changes in acidity. This is especially problematic for Figure 4a and part of Figure 4b, but also for Figure 5 and Figure 6, and should be addressed.

We agree with the reviewer that it is important to analyze the impact of this deviation in the atmospheric $CO_2$ on simulated past changes in pH. In our previous version of the manuscript, we did this briefly in section 5.2 (first paragraph), where we referred to Table S1. There, we estimated that the deviation in atmospheric $CO_2$ in NorESM-ME causes a drop in pH that is 0.01 stronger than expected during the period 1850-2005 under actual atmospheric $pCO_2$ change. This is, however, on the same order of magnitude as the uncertainty in the GLODAPv2 pre-industrial estimate, and it is one order of magnitude less than the actual pH change from 1850 to 2005 (approximately 0.1) Its impact on our results is therefore minor.

To make this analysis more prominent in the manuscript, we now discuss Table S1 earlier, in Section 3.3, where we introduce our choice of emission-driven models. We also mention the impact of this deviation in atmospheric $CO_2$ on pH in all instances where we illustrate and discuss the historical change: in the figure captions of Figure 4, 5 and 6, and at line 393-395 in section 5.2.

Third, there are only a subset of CMIP5 emission driven simulations available and only for esmRCP8.5. This 'caveat' is mentioned in the manuscript. However, the comparison between the NorESM1-ME simulations and the CMIP5 emission driven simulations is currently buried in the supplementary material. I think the paper would become more comprehensive when including the CMIP5 range also in the figures of the main text, for example as vertical bars in Figure 4 (e.g. in year 2100, but also in year 1850 and year 2005). The reader would immediately see where the NorESM1-ME stands in comparison with other comprehensive CMIP5 ESMs. In addition, the paper would benefit from a paragraph, where the comparison with CMIP5 ESMs is described. For example, when looking at Figure S5 and Table S6 it seems to me that the NorESM1-ME model simulates the largest decrease in pH of all CMIP5 models in the upper 2000m. This might be due to stronger simulated atmospheric $CO_2$ increases than in other models. When using the same forcing (concentration driven runs), the model seems to be more in line with other CMIP5 models. Is this true? In any case, this needs to be discussed in the main text, maybe even highlighted in the abstract.

Thank you for these suggestions! We included the CMIP5 model ranges for 1870, 2005 and 2099 in Figure 4 (We could not do it for 1850 and 2100 because the simulations with some models did not cover these years). We also expanded our text on the inter-model spread in section 5.2 (lines 410-415) where we now mention that NorESM1-ME simulates a stronger decrease in pH than the other models, which most likely is a result of a stronger increase in atmospheric $CO_2$. As the reviewer suggested, the larger spread in the emission driven runs compared to the concentration-driven ones is most likely a result of the inter-model spread in atmospheric $CO_2$, which we now mention. A full analysis of the differences between the emission-driven and concentration-driven runs is, however, beyond the scope of this paper.

Forth, why are the emission driven simulations so different than the concentration driven simulations below 3000m (CMIP5 as well as NorESM1-ME; Figure S5g-i)?

Thank you for noticing this! This spread was a result of a mistake in our data handling, which we apologise for. After correcting this, the inter-model spread in the deep is very similar in the emission-and concentration-driven runs. The corresponding figure has been revised accordingly.

---

## Author Response (AR4)

Dear Dr. Gattuso,

Thank you very much for all the work that you have done on the paper! We have finalized all the corrections. Please find our point-by-point answers below.

Sincerely,
Filippa Fransner and co-authors

—--------------------------------------------------------------------

- Fig. 2: it would be great to make this figure and its legend self-explanatory. Could you mention the units of CT and of H+ change, specifically mention in the legend that pHi is initial pH, also in the legend the pH scale?

We have revised the figure as suggested, and think that it is now much better!

- 221: at at

Corrected

- 378: Fig. 3d

corrected

- Fig. 4: it would be great to have the color code of the future scenarios shown in the figure rather than described in the legend; mention the fact that this is pH on the total scale (also in other figures even though it is mentioned once at the beginning.)

We added the color codes for the future scenarios in the figure, and now mention that pH is on total scale. We also revised the other figures accordingly.

- Table 4, S2, S10 and Fig. 7, 8, 13, S1, S18, S21: Indicate that the acronyms identify basins which are defined in Fig. 1 (definitions are often though not always given in the supplementary material but I suggest to refer to Fig. 1). That is the minimum to be done. It would be even better to provide the basin names in full, on two lines if needed.

We have provided the basin names in full in all related figure and table captions.

- Fig. 6 is perhaps a bit small. Make sure that the final one has the proper size and resolution.

Done

- 517: 2 °C

corrected

- Fig. 14 and : spell out "residuals" in full on the plot (it is not longer than "temperature")

Done

- 647: This work builds on Skjelvan et al. (2014)

corrected

- 650: climatology

corrected

- Data availability: please add the urls of the datasets stored in the NECI and NMDC databases.

Added

- Table S2 and S3, S4, S5: In most instances, it does not make sense to provide 2 decimal places considering the (relatively large) SEs.

We changed table S2 (for temperature), so that it displays one decimal place. For the other tables, however, we decided to stick with two. In these tables, in some cases the significance of the trend depends on the second decimal, and we prefer being consequent with the numbers of decimals within one table.

- Fig. S5: add space between a number and the unit (4000 m)

Done

- Fig. S16: there is no yellow. By the way, it would be a good idea to keep the blue here and replace the yellow colo, which is more difficult to read, in previous figures.

We corrected it so the time series figures have the same color coding in the main manuscript and the supplementary. We made the yellow darker, more orange, to make it more visible.